# Provably Efficient Learning Algorithms for Noisy Quantum State and Process Tomography

## Abstract

Characterizing noisy $n$-qubit states and processes is vital yet lacks scalability with conventional methods. Considering the circuit under unital or non-unital independent and identically distributed (i.i.d) single-qubit noise where each local gate follows the local 2-design assumption, we propose a structure-free learning algorithm that reconstructs any noisy process or state from measurement data. The proposed algorithm yields $\mathrm{poly}(n, 1/\epsilon)$ sample complexity and classical post-processing running time for target accuracy $\epsilon$ in the *average case* scenario over the random circuit ensemble. We numerically benchmark the algorithm on both unital and non-unital i.i.d single-qubit noise channels, and our results indicate that the algorithm remains highly effective and accurate even for specific quantum circuits, such as noisy Hamiltonian dynamics, suggesting its broader practical utility. This work offers a new approach to practical quantum-process learning, and suggests a potential path for scalable process characterization in near-term quantum devices.

## 1 Introduction

Quantum computers are entering regimes beyond the reach of classical computational power (Arute et al., 2019; Morvan et al., 2024; Zhong et al., 2020). Coherent manipulation of complex quantum states with hundreds of physical qubits has been demonstrated across multiple platforms, including trapped ions (Smith et al., 2016), neutral atom arrays (Evered et al., 2023), and superconducting qubit circuits (Arute et al., 2019; Morvan et al., 2024; Acharya et al., 2024). As quantum hardware continues to scale in size and complexity, the ability to characterize quantum states and quantum processes becomes critical for advancing quantum error correction code (Bravyi et al., 2024; Acharya et al., 2024), quantum error mitigation (Kim et al., 2023b; O'Brien et al., 2023), and quantum algorithms (Kim et al., 2023a; Morvan et al., 2024). This drive for advancing quantum utility is coupled with an increasing demand for verifiable results, as emphasized in recent literature arguing that the ultimate success of quantum systems hinges on robust certification and system validation (Babbush et al., 2025). Consequently, the comprehensive characterization of quantum states and processes is paramount to meet this demand. Among various approaches for characterizing quantum states and processes, quantum state tomography (QST) (Banaszek et al., 2013; Blume-Kohout, 2010; Eisert et al., 2020; Gross et al., 2010; Hradil, 1997; Mauro D'Ariano et al., 2003) and quantum process tomography (QPT) (Chuang & Nielsen, 1997; D'Ariano & Lo Presti, 2001; Mohseni et al., 2008) stand as fundamental methods to reconstruct target quantum processes (states) by leveraging quantum measurement results.

Due to the quantum process and quantum state being defined in exponentially high-dimensional Hilbert space, the challenge is fundamental for both QST and QPT tasks. It is proven that both approaches rely on extensive measurements of many observables and incur exponential resource scaling with system size *in the worst-case scenario* (Chen et al., 2022b; Haah et al., 2023; 2017; O'Donnell & Wright, 2016; Oufkir, 2023). However, the above "no-go" results do not rule out efficient algorithms for QPT (QST) tasks in the *average-case* scenario. Actually, assuming specific structures or relaxed learning objectives, QPT (QST) tasks would be efficient (Aaronson & Grewal, 2023; Anshu et al., 2020; Arunachalam et al., 2023; Bairey et al., 2019; Che et al., 2021; Chen et al., 2022a; Cramer et al., 2010; Flammia & Wallman, 2020; Flammia & O'Donnell, 2021; Gebhart et al., 2023; Granade et al., 2012; Grewal et al., 2024; 2023; Gross et al., 2021; Gu et al.,

2024; Haah et al., 2022; Hangleiter et al., 2024; Huang et al., 2023b; 2020; Lai & Cheng, 2022; Lanyon et al., 2017; Li et al., 2020; Montanaro, 2017; Rouzé & França, 2024; Stilck França et al., 2024; Van Den Berg et al., 2023; Yu et al., 2023; Zubida et al., 2021; Wu et al., 2025b) in sample or classical post-processing complexity. To the best of our knowledge, current results are not efficient when the target quantum process (state) is given by a general *Noisy Quantum Computer* which has a certain level of noise channels before and after each quantum gate, and the quantum noise could be either unital or non-unital channels.

On the other hand, given the power of classical artificial-intelligence methods, it is natural to consider their application to complex QPT and QST tasks, such as neural-network models (Melko et al., 2019; Acharya et al., 2019; Wanner et al., 2024; Tang et al., 2024), tensor networks (Torlai et al., 2023), diffusion models (Yehui et al., 2025), and other approaches (Wu et al., 2025a; Du et al., 2025). However, these heuristic methods generally lack theoretical guarantees or may not handle QPT and QST in a noisy environment. These advances, together with the fundamental limitations discussed above, naturally raise a question:

*"Can we efficiently learn a general noisy quantum process and quantum state when the underlying noise channel may be unital or non-unital?"*

In this paper, we answer this question by proposing a unified learning framework for both QPT and QST. The key idea relies on a unified representation of noisy quantum processes and states (Lemmas 2 and 3). Specifically, let $\mathcal{C}$ denote the target noisy quantum circuit. We show theoretically that any noisy quantum process $\mathrm{Tr}\,(O\mathcal{C}(\cdot))$ accompanied by an unknown measurement $O$, and quantum state $\rho = \mathcal{C}(|0^n\rangle\langle 0^n|)$, regardless of whether the underlying noise channel is unital or non-unital, their related tomography tasks can be reduced to learning an unknown observable with the decomposition $\mathcal{M} = \sum_{|P|\leq\mathcal{O}(1),P\in\{I,X,Y,Z\}^{\otimes n}} \alpha_P P$, where the coefficients $\alpha_P \in \mathbb{R}$. This observation reduces the learning space from $4^n$ to $\mathrm{poly}(n)$, yielding an efficient learning algorithm when the quantum circuit suffers from a constant-strength noise channel after each quantum gate. The fundamental idea is illustrated in Figure 1. Finally, we numerically benchmark our algorithm on noisy Hamiltonian dynamics driven by a two-dimensional lattice model (Kim et al., 2023a) and apply it to the quantum error mitigation with an agnostic input state. The results demonstrate high accuracy for both QST and QPT tasks.

## 2 PRELIMINARY KNOWLEDGE

To motivate and contextualize our contribution, we briefly review the requisite background on noisy quantum channels and circuits.

**Definition 1** (Single-Qubit Pauli Channel). *Let $\mathcal{E}_{\mathrm{Pauli}}$ denote the single-qubit Pauli channel, which is*

$$\mathcal{E}_{\mathrm{Pauli}}(\rho) = \gamma_1\rho + \gamma_2 X\rho X^\dagger + \gamma_3 Y\rho Y^\dagger + \gamma_4 Z\rho Z^\dagger, \tag{1}$$

*where real parameters $\gamma_1 + \gamma_2 + \gamma_3 + \gamma_4 = 1$, and $\gamma_i \in [0,1]$ for $i \in [4]$.*

As a standard unital quantum channel, the Pauli noise has the property $\mathcal{E}_{\mathrm{Pauli}}(I) = I$, $\mathcal{E}_{\mathrm{Pauli}}(X) = (1 - 2(\gamma_3 + \gamma_4))X$, $\mathcal{E}(Y) = (1 - 2(\gamma_2 + \gamma_4))Y$ and $\mathcal{E}_{\mathrm{Pauli}}(Z) = (1 - 2(\gamma_2 + \gamma_3))Z$. Note that if $\gamma_2 = \gamma_3 = \gamma_4$, $\mathcal{E}$ degenerates to an i.i.d single-qubit depolarizing noise, which is $\mathcal{E}_{\mathrm{depo}}(P) = (1 - \gamma)P$ for $P \in \{X,Y,Z\}$. Techniques like Pauli twirling are employed to transform complex unital channels into diagonal forms on the Pauli basis (Chen et al., 2023; Wallman & Emerson, 2016a). In the following, we utilize the Pauli noise channel to represent the unital channel.

Another widely studied class of quantum channels is the non-unital channel, which describes channels that do not map the identity operator to itself. This kind of noise often reflects complicated environmental disturbances on the quantum system, where a canonical example is the amplitude damping. Ref. (Angrisani et al., 2025) decompose the normal form of a non-unital single-qubit noise channel $\mathcal{E}$ as

$$\mathcal{E} = \mathcal{E}_{\mathrm{depo}}^\gamma \circ \mathcal{E}', \tag{2}$$

where $\mathcal{E}'$ is a suitable (non-physical) linear map and $\mathcal{E}_{\mathrm{depo}}^\gamma$ is a depolarizing noise with the effective depolarizing rate $\gamma$. Given this observation, we define a unified noise parameter across unital and

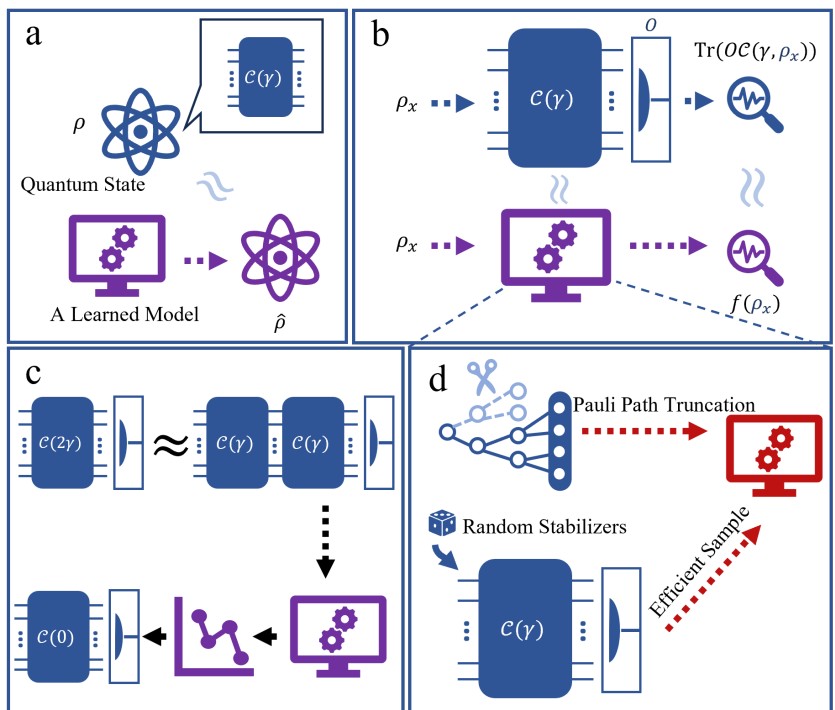

Figure 1: (a) Illustration of the noisy quantum state learning, wherein a trained model $\hat{\rho}$ is generated by leveraging the adaptive measurement result from the target noisy quantum state $\rho$. (b) Depiction of the noisy quantum process learning. Here, the noisy quantum process $\mathcal{C}(\gamma)$ represents a $d$-depth quantum circuit with noise strength $\gamma$, and $O$ represents an unknown local measurement operator. The task is to learn a function $f$ such that $|f(\cdot) - \mathrm{Tr}[O\mathcal{C}(\gamma, \cdot)]| \leq \epsilon$ for all input quantum states $\rho_x$. (c) The proposed learning algorithm can be applied to the quantum error mitigation task. (d) Outline of the fundamental principle underlying our learning algorithm.

non-unital noise channels:

$$\gamma = \begin{cases} 2(\gamma_i + \gamma_j) \ (i,j) \in \{2,3,4\}, & \mathcal{E} \text{ is unital} \\ 1 - \chi_{\mathcal{D}}(\mathcal{E}), & \mathcal{E} \text{ is non-unital} \end{cases} \tag{3}$$

where $\chi_{\mathcal{D}}(\mathcal{E})$ denotes the mean squared contraction coefficient of $\mathcal{E}$ with respect to the locally unbiased distribution $\mathcal{D}$. The details of the non-unital noise are in Appendix C.1.3.

**Definition 2** (Schatten $\tau-$Norm). *The Schatten $\tau-$norm of a matrix $A$ is defined as $\|A\|_\tau = (\sum_i \nu_i^\tau)^{\frac{1}{\tau}}$, where $\nu_i$ is the singular value of $A$ and $\tau$ is a positive integer. Note that $\|A\|_1 = \mathrm{Tr}[\sqrt{AA^\dagger}]$, and $\|A\|_2$ coincides with the Frobenius norm $\|A\|_F$.*

**Definition 3** (The Squared Normalized Frobenius Norm). *Suppose the matrix $A = \sum_P \alpha_P P$, with $P \in \{X/\sqrt{2}, Y/\sqrt{2}, Z/\sqrt{2}, I/\sqrt{2}\}^{\otimes n}$, its squared normalized Frobenius norm is defined by $\|A\|_F^2 = \sum_P \alpha_P^2$.*

**Definition 4** (Hamming Weight of Pauli Operators). *Suppose $P$ represents an $n$-qubit (normalized) Pauli operator, then its Hamming weight $|P|$ is defined as the number of qubits that are non-trivially acted by $P$.*

## 3  PROBLEM STATEMENT

Here, we consider an $n$-qubit noisy quantum process

$$\mathcal{C} = \mathcal{E}^{\otimes n} \mathcal{C}_d \mathcal{E}^{\otimes n} \mathcal{C}_{d-1} \cdots \mathcal{E}^{\otimes n} \mathcal{C}_1 \tag{4}$$

in which a $\gamma$-strength local noise channel $\mathcal{E}$ (unital or non-unital) is applied uniformly throughout the circuit. The quantum circuit depth is $d$, and each layer of the circuit consists of two-qubit gates acting between every pair of qubits, where each gate is uniformly sampled from a local 2-design unitary group. In other words,

This architecture, which interleafs high-fidelity unitary operations with i.i.d single-qubit noise, serves as a standard model for benchmarking computational hardness (e.g., in quantum supremacy (Arute et al., 2019)) and is the theoretical basis for characterizing device fidelity via Randomized Benchmarking (RB) (Magesan et al., 2011). For a complete and rigorous understanding of the model's topological structure, its graph-theoretic definitions, and its high generality, see Appendix B

Here, a natural step toward fully understanding the power of noisy quantum computation is to learn the behavior of the noise process. Specifically, we focus on learning the quantum mean value $\text{Tr}(O\mathcal{C}(\rho_x))$, where $O$ is an unknown observable and $\rho_x$ is the input state of $\mathcal{C}$. In the worst-case scenario, learning the output of a general quantum process is even quantumly hard; however, we argue that the noisy quantum process can be efficiently characterized when the noise parameter $\gamma = \Omega(1)$. We first consider a warm-up task that inspires us to design a highly efficient learning algorithm for general noisy quantum processes.

**Problem 1** (Noisy Quantum State Learning). *Let $\rho = \mathcal{C}(|0^n\rangle\langle 0^n|)$ be an unknown quantum state prepared by $\mathcal{C}$. The target is to learn an approximation $\hat{\rho}$, which is a classical representation of $\rho$. such that their trace distance $T(\rho, \hat{\rho}) \leq \epsilon$ for any $\epsilon \in (0, 1)$.*

Here, the trace distance $T(\sigma, \rho) = \frac{1}{2}\text{Tr}\left(\sqrt{(\sigma - \rho)^\dagger(\sigma - \rho)}\right)$ is used as the maximum bias derived by quantum states $\rho$ and $\sigma$. In the following section, we will demonstrate that learning a noisy quantum state representation may inspire a quantum process characterization learning algorithm.

**Problem 2** (Noisy Quantum Process learning). *Given an unknown noisy quantum process $\mathcal{C}$ and an unknown observable $O$, the task is to learn a classical function $f$, such that for any $\epsilon \in (0, 1)$ and input quantum state $\rho_x$, $|f(\rho_x) - \text{Tr}(O\mathcal{C}(\rho_x))| < \epsilon$ .*

## 4 QUANTUM LEARNING ALGORITHM FOR NOISY QUANTUM STATE

We first present an efficient method for learning a classical representation of a noisy quantum state.

**Lemma 1** (Unified Representation of Noisy Quantum State). *Let the noisy quantum state be $\rho = \mathcal{C}(|0^n\rangle\langle 0^n|)$, with $\mathcal{C} = \mathcal{E}^{\otimes n}\mathcal{C}_d\mathcal{E}^{\otimes n}\mathcal{C}_{d-1}\cdots\mathcal{E}^{\otimes n}\mathcal{C}_1$ represent a $d$-depth noisy quantum circuit, where $\mathcal{C}_i(\cdot) = C_i^\dagger(\cdot)C_i$ is a unitary channel consisting of a layer of two-qubit Haar random gates. The noisy quantum state*

$$\rho = \sum_{s \in \mathcal{P}_n^{\otimes(d+1)}} (1-\gamma)^{|s|}\Phi(\mathcal{C}, s)s_d, \tag{5}$$

*where the $n(d+1)$-qubit operator is $s = (s_0 s_1 \cdots s_d)$ and $\mathcal{P}_n = \{I/\sqrt{2}, X/\sqrt{2}, Y/\sqrt{2}, Z/\sqrt{2}\}^{\otimes n}$. The related coefficient*

$$\Phi(\mathcal{C}, s) = \begin{cases} \text{Tr}(s_d\mathcal{C}_d(s_{d-1}))\cdots\text{Tr}(s_1\mathcal{C}_1(s_0))\text{Tr}(s_0|0^n\rangle\langle 0^n|), & \mathcal{E} \text{ is unital,} \\ \text{Tr}(s_d\mathcal{E}'^{\otimes n}\mathcal{C}_d(s_{d-1}))\cdots\text{Tr}(s_1\mathcal{E}'^{\otimes n}\mathcal{C}_1(s_0))\text{Tr}(s_0|0^n\rangle\langle 0^n|), & \mathcal{E} \text{ is non-unital} \end{cases} \tag{6}$$

*where the channel $\mathcal{E}'$ is defined as Eq. 2.*

See Appendix C.1 for the proof. Although an arbitrary $n$-qubit density operator requires $4^n$ Pauli operators, our result theoretically demonstrates the intrinsic simplicity of a noisy quantum state. It is observed that most of Pauli paths $s_d$ will be exponentially decayed when the noise channel strength $\gamma = \mathcal{O}(1)$, which implies the low-weight Pauli paths dominate the noisy quantum state. This observation enables a much more compact approximation to the noisy quantum state.

**Lemma 2.** *Let the noisy quantum state $\rho = \mathcal{C}(|0^n\rangle\langle 0^n|)$ with $\mathcal{C}$ defined as Eq. 4 , where $\mathcal{C}_i$ is a layer of two-qubit Haar random quantum gates. With the success probability $\geq 1 - \delta_1$, there exists a density matrix $\hat{\rho} = \sum_{|s_d|\leq l', s_d \in \mathcal{P}_n} \alpha_{s_d}s_d$ such that*

$$||\rho - \hat{\rho}||_1 \leq \epsilon_1, \tag{7}$$

*where coefficients $\alpha_{s_d} \in \mathbb{R}$ and $l' = \mathcal{O}\left(\frac{1}{\gamma}\log\left(\frac{1}{\epsilon_1\delta_1}\right)\right)$.*

We note that Lemma 2 holds for both *unital* and *non-unital* noisy channels. Consider the trace distance $T(\rho, \hat{\rho}) = \frac{1}{2}\|\rho - \hat{\rho}\|_1$ and $T(\rho, \hat{\rho}) = \max_{P \leq I} |\text{Tr}[P(\rho - \hat{\rho})]|$, the foregoing approximation result immediately yields a tight upper bound on quantum expectation values.

*Proof Sketch.* The fundamental idea is to obtain an efficient representation of a noisy quantum state by leveraging Lemma 1. The contribution of each Pauli path $s_d$ is determined by a related pre-factor $(1 - \gamma)^{|s|}\Phi(\mathcal{C}, s)$, which decays exponentially with the Pauli-path weight $|s|$. Since $|s_d| \leq |s|$, we truncate the noisy-state representation in Lemma 1 to terms with $|s_d| \leq l'$. It therefore suffices to show that the rest of the average-case error $\mathbb{E}_\mathcal{C}[\sum_{|s_d|>l'} \Phi(\mathcal{C}, s)]^2$ is a constant. If $\mathcal{E}$ represents an unital noise channel, the quantum local random gate property enables us to bound the contribution of each truncated Pauli term (Aharonov et al., 2023). Specifically, if $|s_d| > l'$, then there are at least $|s|/4$ gates whose input and output are both non-identity Pauli operators, and consequently $\Phi(\mathcal{C}, s)$ can be upper bounded by $\frac{1}{15}^{|s|/4}$. For non-unital noise channel $\mathcal{E}$, the local 1-design quantum gate property enables $\mathbb{E}_\mathcal{C}[\sum_{|s_d|>l'} \Phi(\mathcal{C}, s)]^2$ upper bounded by the normalized Frobenius norm of the input quantum state, that is $|0^n\rangle\langle 0^n|$ in our case (Angrisani et al., 2025). In both cases, the average-case error has the upper bound $\epsilon_1$ with large probability by choosing $l' = \gamma^{-1} \log(\epsilon_1^{-1}\delta_1^{-1})$. This completes the proof. Details are provided in Appendix C.2. $\qquad\square$

The above observation implies that the number of non-trivial terms (those with $\alpha_{s_d} \neq 0$) is bounded by $N_s \leq 2^{\mathcal{O}(l')} = \mathcal{O}(1/\epsilon_1)$. Hence, when the required accuracy is $\epsilon_1 = 1/\text{poly}(n)$, all Pauli terms $s_d$ appearing in the ansatz $\hat{\rho}$ can be enumerated efficiently. Consequently, tomography of the noisy state $\rho$ is reduced to tomography of its approximation $\hat{\rho}$, determining the unknown coefficients $\alpha_{s_d}$ for $s_d$, then it suffices to perform the noisy-state tomography task. Since all 'low-weight' Pauli operators $s_d$ can be enumerated in advance, the classical shadow method (Huang et al., 2020) is a natural candidate for estimating the coefficients $\alpha_{s_d}$, yielding an $\mathcal{O}\left(\log(1/\epsilon_1)\epsilon_1^{-2}\right)$ sample-complexity guarantee.

Nevertheless, the classical shadow method may not extend directly to quantum process tomography tasks. To implement a 'unified' learning approach for both quantum noisy state and process tomography tasks, we provide another method for estimating coefficients $\alpha_{s_d}$ from the quantum randomized measurement results. We generate a dataset $\{|\psi_j\rangle = \otimes_{i=1}^n |\psi_{i,j}\rangle, v_j = \langle\psi_j|\rho|\psi_j\rangle\}_{j=1}^{N_{\text{data}}}$ by drawing each single-qubit stabilizer $|\psi_{i,j}\rangle$ uniformly sampled from the set $\text{Stab} = \{|0\rangle, |1\rangle, |+\rangle, |-\rangle, |y+\rangle, |y-\rangle\}$. Here, the quantum state overlap $v_j = \langle\psi_j|\rho|\psi_j\rangle$ can be efficiently obtained by using the SWAP-test method (Buhrman et al., 2001). Without loss of generality, we assume each single-qubit stabilizer state can be prepared by $|\psi_{i,j}\rangle = U_{i,j}|0\rangle_i$, where $U_{i,j}$ is a random single-qubit Clifford gate. By leveraging the orthogonal property of single-qubit Pauli operators $Q_i$ in the context of the Clifford ensemble, that is

$$\mathbb{E}_{U_{i,j}\sim\text{Cl}(2)}\left[U_{i,j}^{\dagger\otimes 2}(Q_i \otimes Q_i')U_{i,j}^{\otimes 2}\right] = \begin{cases} I^{\otimes 2}, \text{ if } Q_i = Q_i' = I, \\ \frac{1}{3}\sum_{Q_i\in\{X,Y,Z\}^{\otimes 2}}(Q_i \otimes Q_i), \text{ if } Q_i = Q_i' \neq I. \\ 0, \text{ if } Q_i \neq Q_i', \end{cases} \quad (8)$$

coefficients $\alpha_{s_d}$ are obtained as

$$\alpha_{s_d} = 3^{|s_d|}\mathbb{E}_{|\psi_j\rangle\sim\text{Stab}^{\otimes n}} v_j \langle\psi_j| s_d |\psi_j\rangle \approx \frac{3^{|s_d|}}{N_{\text{data}}}\sum_{j=1}^{N_{\text{data}}} v_j \langle\psi_j| s_d |\psi_j\rangle. \quad (9)$$

The in-depth explanation of the learning Algorithm is provided in Appendix C.3. We note that the above learning approach is efficient in both sample and computational complexity (classical post-processing).

**Theorem 1** (Noisy Quantum State Learning). *For any noisy quantum state $\rho$ prepared by a noisy quantum circuit $\mathcal{C}$ (Eq. 4), where $\mathcal{C}_i$ is a layer of two-qubit random haar quantum gates, there exists a learning algorithm that can efficiently solve Problem 1 with success probability $\geq 1 - \delta$. The learning algorithm requires sample complexity $N_{\text{data}} = 6^{\mathcal{O}(\gamma^{-1}\log(\epsilon^{-1}\delta^{-1}))}\log(1/\delta)\epsilon^{-2}$ and classical post-processing complexity $\mathcal{O}\left(n \cdot 24^{\mathcal{O}(\gamma^{-1}\log(\epsilon^{-1}\delta^{-1}))}\log(1/\delta)\epsilon^{-2}\right)$.*

When the required accuracy and failure probability $\epsilon, \delta = 1/\text{poly}(n)$, the proposed learning algorithm is highly efficient to construct a density matrix $\hat{\rho}$ such that (a) $T(\rho, \hat{\rho}) \leq \epsilon$ and (b) $|\text{Tr}(O\rho) - \text{Tr}(O\hat{\rho})| \leq \epsilon\|O\|$. In many noisy intermediate-scale quantum (NISQ) algorithms, one is often interested in the expectation values of Pauli operators. Theorem 1 supplies an efficient method for benchmarking the output of NISQ algorithms.

## 5  LEARNING A QUANTUM PROCESS CHARACTERIZATION

Compared with the noisy quantum state tomography, QPT is a more challenging task, which requires an exponential query complexity in the worst-case scenario (Haah et al., 2023), rendering it infeasible for large-scale systems. Inspired by the noisy quantum state tomography method, we proposed an efficient learning algorithm for the QPT task, particularly when the quantum process is given by a noisy quantum circuit $\mathcal{C}$ (Eq. 4) followed by an unknown quantum measurement $O$. Without loss of generality, we assume the $n$-qubit observable $O = \sum_{Q \in \{I,X,Y,Z\}^{\otimes n}} \text{Tr}[OQ]Q/2^n$ is the linear combinations of local operators, where each local Pauli operator $|Q| = \mathcal{O}(1)$. In other words, $O$ is considered as a sum of few-body observables, where each qubit is acted on by a constant number of the few-body observables.

Let the noisy quantum channel be given by the Kraus decomposition $\mathcal{C} = \sum_j K_j(\cdot)K_j^\dagger$. It is observed that

$$\text{Tr}\left[\mathcal{C}(\rho_x)O\right] = \text{Tr}\left[\sum_j K_j \rho_x K_j^\dagger O\right] = \text{Tr}\left[\sum_j \rho_x K_j^\dagger O K_j\right] = \text{Tr}\left[\rho_x \mathcal{C}^\dagger(O)\right]. \quad (10)$$

Consequently, the key step is to learn the 'dual' representation $\mathcal{C}^\dagger(O)$. We demonstrate that this dual operator also admits low-weight Pauli paths, allowing for a truncation-based approximation similar to that employed for noisy states.

**Lemma 3.** *Let the noisy quantum circuit $\mathcal{C} = \mathcal{E}^{\otimes n}\mathcal{C}_d\mathcal{E}^{\otimes n}\mathcal{C}_{d-1}\cdots\mathcal{E}^{\otimes n}\mathcal{C}_1$ represent a $d$-depth noisy quantum circuit, where $\mathcal{C}_i$ is a layer of two-qubit Haar random quantum gates and $\mathcal{E}$ represents an i.i.d single-qubit noisy channel (unital or non-unital). With success probability $\geq 1 - \delta_2$, there exists an operator $\mathcal{C}^{(l')\dagger}(O) = \sum_{|P| \leq l', P \in \mathcal{P}_n} \beta_P P$ such that*

$$\left\|\mathcal{C}^{(l')\dagger}(O) - \mathcal{C}^\dagger(O)\right\|_F \leq \epsilon_2, \quad (11)$$

*where coefficients $\beta_P \in \mathbb{R}$ and $l' = \mathcal{O}\left(\gamma^{-1}\log(1/(\delta_2\epsilon_2))\right)$.*

Similar to the noisy quantum state tomography task, reconstructing $\mathcal{C}^{(l')\dagger}(O)$ proceeds from the data set $\mathcal{D}_{\text{QPT}} = \{|\psi_j\rangle = \otimes_{i=1}^n |\psi_{i,j}\rangle, \phi_j = \text{Tr}\left[O\mathcal{C}(|\psi_j\rangle\langle\psi_j|)\right]\}_{j=1}^{N_{\text{data}}}$, where $|\psi_{i,j}\rangle$ is a single-qubit stabilizer randomly sampled from the set $\text{Stab}$, and $\phi_j$ denotes the output of the target quantum process. According to the Eq. 8, coefficients $\beta_P$ can be learned efficiently via

$$\beta_P = \frac{3^{|P|}}{N_{\text{data}}} \sum_{i=1}^{N_{\text{data}}} \phi_j \langle\psi_i| P |\psi_i\rangle. \quad (12)$$

The complete QPT procedure is summarized in Algorithm 1.

From the above algorithm, it can be observed that the computational overhead primarily stems from two sources: (1) the sampling complexity $N_{\text{data}}$, (2) and the complexity of classical post-processing. Both of these costs depend on the number of $s_d$ (Pauli operator $P$ in the algorithm), which in turn is governed by how many legal Pauli paths are retained, in other words, the number of Pauli operators $s_d$ with non-zero parameter $\beta_{s_d}$ contained in $\mathcal{C}^{(l')\dagger}(O)$. According to Lemma 3, the weight of $s_d$ is given by $l' = \mathcal{O}(\gamma^{-1}\log(\epsilon^{-1}\delta^{-1}))$. Therefore, a rough estimate of the number of legal paths is $\mathcal{O}(n^{l'})$. For $\epsilon = 1/n$, the number of legal paths becomes $\mathcal{O}(n^{\log n})$, incurring quasi-polynomial sampling and post-processing complexity.

However, we can tighten the bound to retain only $e^{\mathcal{O}(l')}$ legal paths. For unital noisy circuit, the lower bound of $\mathbb{E}_{\mathcal{C},|s|\leq l}\Phi(\mathcal{C}, s)^2$ is $\frac{1}{15}^{|s|}$, when the input and the output of the each gate are non-identity Pauli operators for a legal Pauli path. Consequently, the sum over all paths with weight up

---

**Algorithm 1** Quantum Process Learning Algorithm

---

**Input:** Data set $\mathcal{D}_{\mathrm{QPT}} = \{|\psi_j\rangle = \otimes_{i=1}^{n} |\psi_{i,j}\rangle\}_{j=1}^{N_{\mathrm{data}}}$ and accuracy parameter $\epsilon$;

**Output:** A $f(\cdot)$ such that $|f(\cdot) - \mathrm{Tr}\,[O\mathcal{C}(\cdot)]| \leq \epsilon$ with high success probability for all input quantum states;

Let $l' = [\log(1/\epsilon)]$, enumerate all the $P \in \mathcal{P}_n$ with $|P| \leq l'$;

**For** $j \in [N_{\mathrm{data}}]$:

   Take the input state $|\psi_j\rangle\langle\psi_j|$ into the target quantum process, and obtain the output $\phi_j = \mathrm{Tr}\,[O\mathcal{C}(|\psi_j\rangle\langle\psi_j|)]$;

**End For**

**For** each $P \in \mathcal{P}_n$ with $|P| \leq l'$:

   Compute $\beta_P = \frac{3^{|P|}}{N_{\mathrm{data}}} \sum_{j=1}^{N_{\mathrm{data}}} \phi_j \langle\psi_j| P |\psi_j\rangle$.

**End For**

**Output**: $f(\cdot) = \sum_{|P| \leq l'} \beta_P \mathrm{Tr}(P(\cdot))$

---

to $l'$ is given by $\mathcal{O}(1) = \sum_{|s_d| \leq l'} \mathbb{E}_{\mathcal{C}, |s| \leq l} \Phi(\mathcal{C}, s)^2 \geq N_s \frac{1}{15}^{l'}$, where $N_s$ is the number of legal Pauli paths. Thus $N_s = 15^{\mathcal{O}(l')} \in e^{\mathcal{O}(l')}$. For non-unital noisy circuits, $N_s$ is bounded by $\max_{Q \in O} |Q| e^{l'}$ for the enumeration that starts from a local term $Q \in O$ non-trivially acting on a constant number of qubits. We conclude the main results in the following Theorem. The proof is given in Appendix D.2.

**Theorem 2** (Noisy Quantum Process Learning). *For any noisy quantum process $\mathcal{C}$ defined as Eq. 4, where $\mathcal{C}_i$ is a layer of two-qubit Haar random quantum gates, and $n$-qubit observable $O = \sum_{Q \in \{I,X,Y,Z\}^{\otimes n}, |Q|=\mathcal{O}(1)} \mathrm{Tr}[OQ]Q/2^n$, there exists a learning algorithm that can efficiently solve Problem 2 with success probability $\geq 1 - \delta$. The learning algorithm requires sample complexity*

$$N_{\mathrm{data}} = \max_{Q \in O}(|Q|^2) 6^{\mathcal{O}\left(\gamma^{-1} \log\left(\|O\|_F \epsilon^{-1} \delta^{-1}\right)\right)} \log\left(\delta^{-1}\right) \epsilon^{-2}, \tag{13}$$

*and classical post-processing complexity* $\mathcal{O}\left(\frac{n \cdot \max_{Q \in O}(|Q|^3) 24^{\mathcal{O}\left(\gamma^{-1} \log\left(\|O\|_F \epsilon^{-1} \delta^{-1}\right)\right)} \log\left(\delta^{-1}\right)}{\epsilon^2}\right)$.

*Moreover, if the noise is unital, the sample complexity is* $6^{\mathcal{O}\left(\gamma^{-1} \log\left(\frac{\|O\|_F}{\epsilon\delta}\right)\right)} \log\left(\delta^{-1}\right) \epsilon^{-2}$ *and classical post-processing complexity is* $\mathcal{O}\left(n \cdot 24^{\mathcal{O}\left(\gamma^{-1} \log\left(\frac{\|O\|_F}{\epsilon\delta}\right)\right)} \log\left(\delta^{-1}\right) \epsilon^{-2}\right)$.

## 6 NUMERICAL EXPERIMENTS

In this section, we present numerical results that employ the proposed learning algorithms to perform noisy quantum process and state tomography, thereby substantiating the theoretical analysis. We further illustrate that the same pipeline can be harnessed for quantum error mitigation. Although our theoretical results rely on the randomness assumption, we numerically verify that our learning algorithm remains highly efficient for a broader class of circuits, including those with specific structure, such as noisy quantum dynamical processes. This demonstrates the broad practical applicability of our approach.

### 6.1 EXPERIMENT SETTING

Benchmarks are performed on the two-dimensional transverse-field Ising model described by the Hamiltonian

$$H = -J \sum_{\langle q,p \rangle} Z_q Z_p + h \sum_q X_q, \tag{14}$$

where the notation $\langle q, p \rangle$ restricts the interaction to nearest-neighbor pairs. The positive coupling strength $J$ and the transverse field $h$ fully parameterize the system. Evolving for total time $T$ via a first-order Trotter formula gives $U(T) \approx \left(\prod_{\langle qp \rangle} e^{i\delta t J Z_q Z_p} \prod_q e^{-i\delta t h X_q}\right)^{T/\delta t}$ with Trotter step

length $\delta t$. Consequently, the quantum circuit reduces to an alternating sequence of RZZ($\theta_J$) and RX($\theta_h$) gates whose rotation angles are fixed by the physical parameters through $\theta_J = -2J\delta t$ and $\theta_h = 2h\delta t$ (Kim et al., 2023a). To simplify the subsequent gate decomposition–specifically, to minimize the CNOT count required for each RZZ – we fix $\theta_J = -\frac{\pi}{2}$ and only change $\theta_h$. As reported in Kim et al. (2023a), current superconducting quantum computers have a certain level of noise within each quantum gate. During our simulation, we thus introduce a i.i.d single-qubit depolarizing channel after each quantum gate, with strength $2 \times 10^{-2}$ for unital cases in both quantum state and process tomography tasks. For more general non-unital noisy channel cases, we assume each quantum gate suffers from an i.i.d single-qubit depolarizing channel and a local amplitude damping channel. We simulate the noisy quantum state and process using the Qulacs Package (Suzuki et al., 2021). The quantum circuit is tested up to a $3 \times 5$-sized instance 20 layers, corresponding to a $2^{15} \times 2^{15}$ matrix, occupying 16 GB of RAM and taking approximately 17 hours for each shot.

### 6.2 EXPERIMENT RESULTS

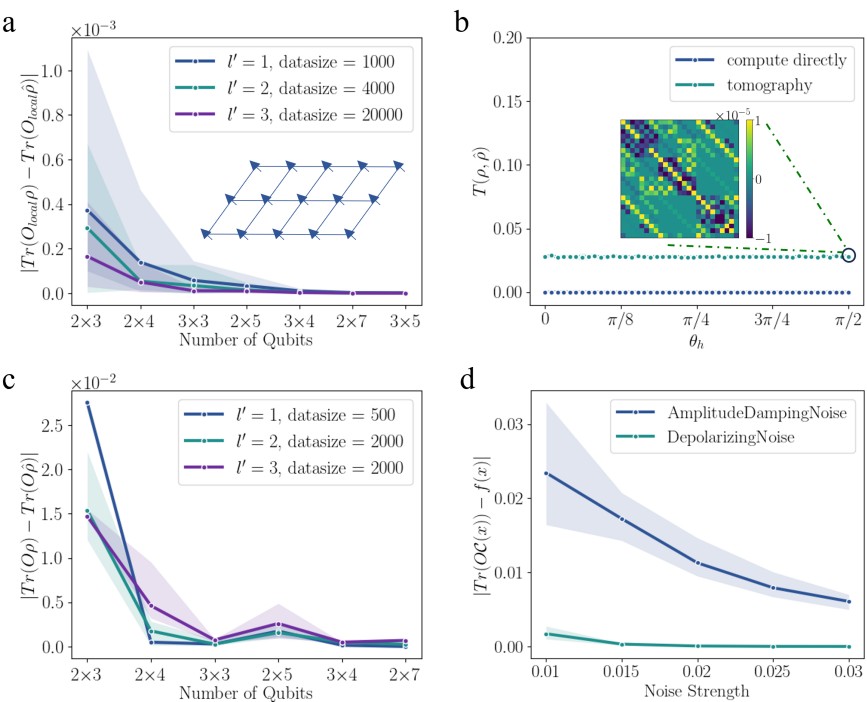

Figure 2: (a) QST results for various numbers of qubits and $l'$. Each circuit is 20 layers accompanied by depolarizing noise of strength 0.02 and fixed $\theta_h = \frac{\pi}{4}$. The grid illustrates the $3 \times 5$ $2D$ transverse field Ising model. (b). Learning of the $\rho$ generated by sweeping $\theta_h$ from 0 to $\frac{\pi}{2}$; the circuit size $2 \times 5$, 45 layers, depolarizing noise strength 0.02. The heat-map shows a $25 \times 25$ sub-matrix of the $\rho - \hat{\rho}$ at $\theta_h = \frac{\pi}{2}$ (the full matrix in Appendix F.2). (c). QPT results for different qubit numbers and $l'$, where the circuit depth is 5 layers and the depolarizing noise strength is 0.01. (d). QPT for the $2 \times 5$ system under 2 kinds of noise and other settings identical to c.

The visualizations of the results of 4 experiments are shown in Figure 2. In all panels, the shaded area indicates the range of experimental outcomes, with the upper bound representing the maximum value and the lower bound the minimum value; the solid line indicates the mean over 10 trials. Figure 2 a shows that the error decreases as the system size increases and stabilizes once the system is sufficiently large; the same behavior is observed in Figure 2 c, consistent with the theoretical findings Theorem 1 and 2. In Figure 2 b, the two dot lines respectively display the ideal outcomes and the outcomes from the tomography algorithm, which conveys that the trace distance exhibits minimal fluctuation as $\theta_h$ varies. Figure 2 d demonstrates that our protocol is capable of effectively learning depolarizing noise and also shows the capacity to learn non-unital noise. Interestingly, the

learning performance improves as the noise strength increases, demonstrating the strong robustness of the learning algorithm in terms of the noise strength. Moreover, our approach reduces the storage required for storing a quantum state: for example, a 14-qubit density matrix ($2 \times 7$ lattice) generally occupies 8 GB, whereas storing the coefficients of its Pauli decomposition requires only 1 KB, offering an efficient and compact representation of noisy quantum processes.

## 6.3 APPLICATION: QUANTUM ERROR MITIGATION

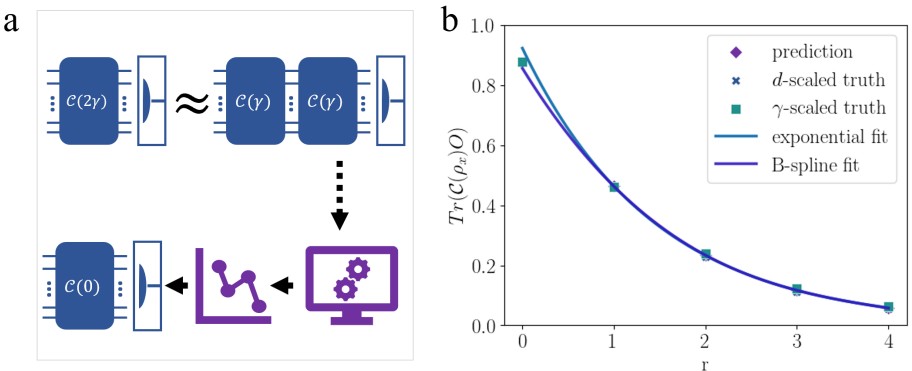

Figure 3: (a) The procedure of the ZNE. (b) The numerical result of the ZNE-QEM using the proposed learning algorithm.

We note that our learning algorithm can also be applied to solve the quantum error mitigation (QEM) task (Eisert et al., 2020). QEM comprises protocols that suppress stochastic errors on NISQ hardware by classical post-processing of measurement data, without introducing full quantum error-correcting codes. Whereas error correction aims to eliminate noise, QEM converts every hardware improvement into an immediate fidelity gain by suppressing residual errors. One QEM approach is zero-noise extrapolation (ZNE), which executes the circuit at several circuit fault rates $\lambda$, which measures the level of errors occurring in the overall circuit, and $\lambda \propto \gamma$ (Cai et al., 2023). Although the circuit output at $\lambda = 0$ cannot be measured directly, an empirical model $h(\lambda)$ linking $\lambda$ to the circuit output can be built from a set of different $\lambda$ values. This allows us to extrapolate the case of $\lambda = 0$, which corresponds to zero noise. Different $\lambda$ values can be generated by purse-stretching(Kandala et al., 2019; Kim et al., 2023b) or by inserting additional noise channels (Endo et al., 2018). For an i.i.d single-qubit noise, it is natural to set $\lambda$ proportional to the gate count, thus $\gamma \propto \lambda \propto$ Number of the gates. Here we vary $\lambda$ by controlling the depth of the circuit $d$.

Whereas the conventional ZNE must be tailored to each specific input, our protocol is input-agnostic. Similar to Lemma 1, $\mathrm{Tr}(O\mathcal{C}(\cdot)) = \sum_{|P| \leq l'} (1 - \gamma)^{|P|} \Phi(\mathcal{C}, P) \mathrm{Tr}(P \cdot)$. Considering the depolarizing noise strength $\gamma < 1$, $(1 - \gamma)^{|P|} = (1 - \gamma)^{\frac{|P|}{d}d} \approx (1 - \gamma d)^{\frac{|P|}{d}}$. In other words, one can obtain the characterizations of the same quantum processes with different noise strength by appending extra quantum circuit layers to the original process, this yields a sequence of learned values $\{f_r || f_r - \mathrm{Tr}[O\mathcal{C}_{rd}(|0^n\rangle\langle 0^n|)]| \leq \epsilon\}_{r \geq 1}$. One can utilize $\{f_r\}_{r \geq 1}$ to extrapolate $f_0$, which is considered as the characterization of $\mathcal{C}_d$ with zero noise.

The result of numerical experiments of application is shown in Figure 3, where we simulate a six-qubit 2D transverse field Ising model Eq. 14 with 5 layers. Two key observations emerge:

- Rescaling either the depth coefficient $d$ or the noise strength $\gamma$ perturbs the dynamics to a comparable extent, as seen from the nearly overlapping dots.

- The characterization obtained by learning the coefficients of $d$ can be extrapolated via curve fitting to estimate the noise-free system (i.e., when $\gamma = 0$) characterization. Exponential extrapolation yields an error $0.0446$; a cubic B-spline (piecewise polynomial) reduces it to $0.0222$.

## 7 DISCUSSION

Efficiently characterizing noisy quantum states and processes has stood as one of the most significant problems over the past decades. In this paper, we propose a provably efficient quantum learning algorithm that handles both unital and non-unital noisy channels, extending previous art from restricted input distributions to arbitrary input quantum states. A more detailed comparison with other works can be found in Appendix A. When the noise strength is a constant value, our learning algorithm is efficient in both sample and runtime complexity. These advances provide rigorous theoretical foundations for analyzing quantum machine learning models, verifying computational outcomes, and benchmarking noisy quantum processes in near-term quantum devices.

For noisy quantum state learning, we have proven that the learning algorithm can be efficient in the average case. We also provide a worst-case lower bound for the sample complexity in both noisy state and process tomography tasks. Specifically, when the noise strength is a constant and the noisy quantum circuit depth is $d = \text{poly}(\log n)$, the sample complexity lower bound for the worst-case scenario is quasi-polynomial. We emphasize that this result does not contradict Theorem 1 and 2: the former statement concerns the worst case, while the latter addresses the average case under the random-circuit assumption. The details are provided in Appendix E.

**Distinguishing Learning from Simulation**    The primary distinction of our approach lies in its informational requirements compared to classical simulation methods. From a practical perspective, simulating a quantum circuit typically requires prior knowledge of the noise strength(Shao et al., 2024; Schuster et al., 2024), and some noise strengths whose efficient characterization can be inherently challenging(Chen et al., 2023). Conversely, our learning algorithm merely requires the noise level to be constant, obviating the need for its exact strength to be known. From a theoretical standpoint, both our learning approach and the cited simulation methods(Aharonov et al., 2023; Angrisani et al., 2025) leverage the principle of Pauli-path integration, wherein exponential noise decay ensures that noisy processes are dominated by low-weight Pauli paths. The crucial difference resides in the informational premise: In our learning algorithm, the quantum gates, circuit architecture, and noise strength are all unknown; we exploit this property to engineer an efficient classical representation (ansatz) for tomography. Classical simulation algorithms, conversely, apply this same property to compute circuit outputs, yet their efficacy is predicated on requiring full knowledge of the involved quantum gates, architecture, and noise strengths. Consequently, our work may be viewed as a 'learning-theoretic dual' to the classical-simulation results(Gil-Fuster et al., 2025). These two paradigms operate in parallel, reflecting complementary perspectives on benchmarking noisy quantum processes—specifically, quantum tomography versus classical verification within benchmarking toolkits (Eisert et al., 2020)

**Future Directions and Open Problems**    Despite the high generality and efficiency of our theoretical framework in the current NISQ regime (assuming constant-level noise $\gamma$), several important open problems remain as we look toward applications in future fault-tolerant quantum computing:

> **Noise Strength Optimization and Inference ($l'$ Truncation):**Theoretically, the Pauli truncation length $l'$ depends on $\gamma^{-1}$. In practice, exploring the use of sparsity-promoting techniques (e.g., LASSO or OMP) to adaptively identify and learn only the most relevant Pauli terms based on data could potentially compress the numerical complexity and enhance precision beyond the existing theoretical bound.

> **Extension to Gate-Dependent Noise:** Our analysis is rigorously founded on the i.i.d. gate-independent noise model. We acknowledge that in larger, more complex quantum architectures, noise often exhibits gate-dependence. Extending our learning framework to enable effective tomography of these more physically challenging noise models remains a significant avenue for future research.

> **Optimal Scaling:** The polynomial dependence of our algorithm's complexity on $\gamma$ and $\epsilon$ raises a natural open question: can these scalings (currently $\mathcal{O}(\gamma^{-1})$ and $\mathcal{O}(\epsilon^{-2})$) be further optimized?

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

## A  COMPARISON WITH OTHER WORKS

The comparison provided in Table 1 and 2 highlights the significant differences in resource scaling across current quantum learning paradigms. Our work offers a novel, provably efficient regime, fundamentally separating its resource requirements from existing methods under the assumption of random noisy quantum circuit.

### A.1  GENERALIZATION AND NOISE SCOPE

The comparison provided in Table 1 highlights the differing constraints of each framework. Our model offers crucial extensions in two key dimensions relative to existing works.

First, the core theoretical advance of our framework is the ability to unify the treatment of both i.i.d single-qubit unital and non-unital noise channels. This contrasts with complexity results derived primarily for the Pauli channel model Raza et al. (2024), which is restricted to unitary noise. Our incorporation of non-unital CPTP maps is essential for modeling realistic hardware decoherence processes like amplitude damping.

Second, our QPT algorithm provides an input-agnostic characterization of the noisy quantum channel, operating without requiring the rotational symmetry of the input state distribution (a requirement for works like Huang et al. (2023a)) or adherence to arbitrary product distributions Chen et al. (2024). This ensures that our framework maintains predictive scalability for arbitrary input states, despite the structural limitation of our noise model to i.i.d. single-qubit errors.

Table 1: Comparison with Related Work on Condition.

| work | input distribution | Channel |
|---|---|---|
| Huang et al. (2023a) | at most polynomially far from a locally flat distribution | General CPTP map |
| Raza et al. (2024) | No restriction | $n$-qubit Pauli channel |
| Chen et al. (2024) | Product state, $\|\|\Gamma\|\|_{op} \leq 1 - \eta$ | Accessible |
| Our Work | No restriction (QPT) / $|0^n\rangle$ (QST) | arbitrary i.i.d. single-qubit noise |

### A.2  RESOURCE SCALING AND EFFICIENCY REGIME

Table 2: Comparison with Related Work on Complexity. $\eta \in (0, 1)$ is related to the input distribution; $M$ is the number of Observables. Let $L = \gamma^{-1} \log \left( \|\|O\|\|_F \epsilon^{-1} \delta^{-1} \right)$ denote the dominant term in the exponent.

| work | Classical Runtime | Sample Complexity ($N_{\text{data}}$) |
|---|---|---|
| Huang et al. (2023a) | $N_{\text{data}}$ | $2^{\mathcal{O}[\log \frac{1}{\epsilon} \log(n)]}$ |
| Raza et al. (2024) | $\mathcal{O}(N_{\text{data}} \cdot 4^n)$ | $\mathcal{O}\left( \frac{\sqrt{n} \log(M) \log^{\frac{3}{2}}\left(\frac{1}{(\epsilon\delta)}\right)}{\epsilon^3} \right)$ |
| Chen et al. (2024) | $N_{\text{data}}$ | $\min\left( \frac{2^{\mathcal{O}(n)}}{\epsilon^2}, n^{\mathcal{O}\left(\frac{\log \epsilon^{-1}}{\log \frac{1}{1-\eta}}\right)} \right) \cdot \log \frac{1}{\delta}$ |
| Our Work | $\mathcal{O}(n \cdot N_{\text{data}})$ | $\max_{Q \in O}(|Q|^2) 6^{\mathcal{O}(L)} \cdot \log\left(\delta^{-1}\right)\epsilon^{-2}$ |

The Table 2 shows a comparison between our theoretical work with other related works. For constant accuracy $\epsilon$, the sample complexity in our algorithm is proportional only to a factor exponential in $L$, which is independent of $n$. This achieves constant sample complexity for a constant noise strength $\gamma$. This contrasts sharply with the method of Raza et al. (2024), which suffers from a $\mathcal{O}(\sqrt{n})$ polynomial dependency on the system size $n$, and an intractable $\mathcal{O}(4^n)$ classical runtime bottleneck inherent to standard shadow tomography protocols when estimating general channels.

When considering high precision requirements, such as setting the accuracy $\epsilon$ to $\frac{1}{n}$, our complexity grows only polynomially with $n$. This provides a substantial advantage over the approaches of Huang et al. (2023a) and Chen et al. (2024), which incur a quasi-polynomial dependence ($n^{\mathcal{O}(\log n)}$ or $2^{\mathcal{O}(\log n \log(1/\epsilon))}$) due to their reliance on methods that scale with the logarithm of the system size in the exponent. Our ability to bypass this quasi-polynomial scaling stems from leveraging the physical constraint imposed by constant noise, which confines the learned system to a low-weight Pauli subspace, ensuring genuinely polynomial scaling in $n$ for practical precision levels.

## B  STRUCTURE AND APPLICABILITY OF THE NOISY CIRCUIT IN THIS WORK

We emphasize that the quantum process studied serves as a standard model with wide and practical applications, especially in the Near-Term Intermediate Scale Quantum (NISQ) era. This appendix details the topological definitions, generality, and practical relevance of the quantum circuit model investigated.

### B.1  TOPOLOGICAL DEFINITION AND MODEL GENERALITY

The studied noisy quantum process $\mathcal{C}$ adopts a layered structure, representing a large class of quantum circuits:

$$\mathcal{C} = \mathcal{C}_1 \mathcal{E}^{\otimes n} \cdots \mathcal{E}^{\otimes n} \mathcal{C}_d \tag{15}$$

in which a $\gamma$-strength local noise channel $\mathcal{E}$ (unital or non-unital) is applied uniformly throughout the circuit. The quantum circuit depth is $d$, and each layer of the circuit consists of two-qubit gates acting between every pair of qubits, where each gate is uniformly sampled from a local 2-design unitary group. The local 2-design assumption is an extremely weak condition, where quantum neural network models are typical cases(McClean et al., 2018; Cerezo et al., 2021), and even Clifford gates satisfy such an assumption (Zhu et al., 2016). We note that if an ensemble follows a $(t + 1)$-design, it must follow the $t$-design property (Mele, 2024). As a result, this assumption is very general and covers a large amount of NISQ algorithms related to 'randomly initial parameters' and 'classical optimizations'(McClean et al., 2018; Cerezo et al., 2021).

The circuit model is formally defined below using graph-theoretic definitions:

**Definition 5** (Architecture, restatement of Haferkamp et al. (2022)). An architecture is a directed acyclic graph that contains $R \in \mathbb{Z}_{>0}$ vertices (gates). Two edges (qubits) enter each vertex, and two edges exit. Two typical examples are listed below:

- A brickwork is the architecture of any circuit formed as follows. Apply a string of two-qubit gates: $U_{1,2} \otimes U_{3,4} \otimes \cdots \otimes U_{n-1,n}$.Then apply a staggered string of gates. Perform this pair of steps $T$ times in total, using possibly different gates each time.

- A staircase is the architecture of any circuit which apples a stepwise string of two-qubit gates: $U_{n,n-1}U_{n-2,n-1} \cdots U_{2,1}$. Repeat this process $T$ times, using possibly different gates each time.

Here, the quantum circuit layer $\mathcal{C}_i$ may adopt any architecture, and ***we note that our learning algorithm can be applied to any geometrical architecture,*** and thus covers a large class of noisy quantum circuits, especially for those used in NISQ algorithms.

**Definition 6** (Random Quantum Circuit, restatement of Haferkamp et al. (2022)). Let $G$ denote an arbitrary architecture. A probability distribution can be induced over the architecture-$G$ circuits as follows: for each vertex in $G$, draw a gate Haar-randomly from $SU(4)$. Then contract the unitaries along the edges of $G$. Each circuit so constructed is called a random quantum circuit.

**Definition 7** (Random noisy quantum circuit). Let $\widetilde{G}$ denote an arbitrary architecture. A probability distribution can be induced over the architecture-$\widetilde{G}$ circuits as follows: for each vertex in $\widetilde{G}$, draw a gate Haar-randomly from $SU(4)$ and an i.i.d single-qubit noisy channel. Then contract the unitaries along the edges of $\widetilde{G}$. Each circuit so constructed is called a random noisy quantum circuit.

The guarantee is a high-probability bound ($\geq 1 - \delta$) over random circuit ensemble defined in Definition. 7. Furthermore, we numerically demonstrate that our learning algorithm can successfully

handle a noisy Hamiltonian dynamics approach, where the underlying quantum circuit does not possess the locally random property.

## B.2 IMPORTANCE FOR QUANTUM BENCHMARKING AND LEARNING

To design powerful quantum algorithms, such as quantum neural network models and related states, a benchmarking algorithm is necessary (Arute et al., 2019; Babbush et al., 2025); otherwise, one may not verify and check the correctness of the implemented quantum algorithm. Following this logic, a large amount of quantum learning algorithms are proposed for quantum state (process) tomography, Hamiltonian learning(Haah et al., 2024) , shallow circuit learning(Huang et al., 2024), quantum gate tomography, and other quantum benchmarking algorithms. ***To the best of our knowledge, this is the first efficient learning algorithm for noisy state and process tomography***, providing an efficient tool for verifying the output of the implemented quantum algorithms on NISQ devices.

## B.3 THE GATE-INDEPENDENT NOISE MODEL

We utilize the gate-independent noise model, which posits that the detrimental effects impacting quantum operations are uniform across all fundamental gates, irrespective of their specific type or physical implementation. This simplifying assumption is widely adopted due to several key factors:

- **Theoretical Tractability:** Adopting a gate-independent noise assumption allows researchers to advance the development and analysis of error correction protocols and fault-tolerant methodologies** without needing to incorporate the intricate details of gate-specific noise characteristics(Knill et al., 2008; Helsen et al., 2019; Chen et al., 2021). This uniformity facilitates the derivation of universal results and theoretical performance bounds (Nielsen & Chuang, 2001).

- **Practical Approximations:** In particular quantum systems—especially those featuring highly calibrated gates acting on the same number of qubits and employing standardized control mechanisms—the variability of noise across different gates can be negligible(Shor, 1996; Arute et al., 2019). In these instances, the gate-independent noise model serves as a tenable approximation, streamlining analysis without substantially compromising precision.

- **Alignment with Noise Conversion Methods (Twirling):** Techniques like Pauli twirling are routinely applied to convert complicated physical noise channels into simpler, diagonal forms in the Pauli basis(Wallman & Emerson, 2016b; Chen et al., 2023). The resulting channel can often be effectively approximated as gate-independent, thereby conforming to the model's postulates.

The gate-independent noise model thus furnishes a foundational framework for comprehending error propagation and engineering correction strategies. We identify the robust depiction of gate-dependent noise, which typically manifests in larger, more intricate quantum architectures, as a significant avenue for future exploration.

## C LEARNING A QUANTUM STATE

The proofs of Lemma 1 and Lemma 2 are presented in this section, together with further implementation details of the QST algorithm.

### C.1 PROOF OF LEMMA 1

In this section, we will prove Lemma 1.

**Lemma 4** (Unified Representation of Noisy Quantum State, Lemma 1). *Let the noisy quantum state $\rho = \mathcal{C}(|0^n\rangle\langle 0^n|)$ with $\mathcal{C} = \mathcal{E}^{\otimes n}\mathcal{C}_d\mathcal{E}^{\otimes n}\mathcal{C}_{d-1}\cdots\mathcal{E}^{\otimes n}\mathcal{C}_1$ representing a $d$-depth noisy quantum circuit, where $\mathcal{C}_i(\cdot) = C_i^\dagger(\cdot)C_i$ is a unitary channel consisting of a layer of two-qubit gates, and $\mathcal{E}$ is a general single-qubit noise channel with strength parameter $\gamma$. Then the noisy quantum state $\rho$*

*can be represented by the Pauli path integral, that is*

$$\rho = \sum_{s \in \mathcal{P}_n^{\otimes(d+1)}} (1-\gamma)^{|s|} \Phi(\mathcal{C}, s) s_d, \tag{16}$$

*where the $n(d+1)$-qubit operator $s = s_0 s_1 \cdots s_d$, $\mathcal{P}_n = \{I/\sqrt{2}, X/\sqrt{2}, Y/\sqrt{2}, Z/\sqrt{2}\}^{\otimes n}$. The Pauli weight $|s|$ represents the number of non-identity operators in $s \in \mathcal{P}_n^{\otimes(d+1)}$. The coefficient*

$$\Phi(\mathcal{C}, s) = \begin{cases} \mathrm{Tr}(s_d \mathcal{C}_d(s_{d-1})) \cdots \mathrm{Tr}(s_1 \mathcal{C}_1(s_0)) \langle 0^n | s_0 | 0^n \rangle, & \text{unital,} \\ \mathrm{Tr}(s_d \mathcal{E}'^{\otimes n} \mathcal{C}_d(s_{d-1})) \cdots \mathrm{Tr}(s_1 \mathcal{E}'^{\otimes n} \mathcal{C}_1(s_0)) \langle 0^n | s_0 | 0^n \rangle, & \text{non-unital} \end{cases} \tag{17}$$

*where the channel $\mathcal{E}'$ is defined as follows.*

We prove it by describing three types of noisy channels, which are depolarizing noise, single-qubit Pauli noise, and non-unital noise.

### C.1.1 DEPOLARIZING NOISE

The property of depolarizing noise $\mathcal{E}_{\mathrm{depo}}$ is that

$$\begin{aligned} \mathcal{E}_{\mathrm{depo}}(I) &= I, \\ \mathcal{E}_{\mathrm{depo}}(X) &= (1-\gamma)X, \\ \mathcal{E}_{\mathrm{depo}}(Y) &= (1-\gamma)Y, \\ \mathcal{E}_{\mathrm{depo}}(Z) &= (1-\gamma)Z, \end{aligned} \tag{18}$$

so that

$$\begin{aligned} \rho &= \sum_{s \in P_n^{d+1}} s_d \mathrm{Tr}(s_d \mathcal{E}_{\mathrm{depo}}^{\otimes n} \mathcal{C}_d(s_{d-1})) \cdots \mathrm{Tr}(s_1 \mathcal{E}_{\mathrm{depo}}^{\otimes n} \mathcal{C}_1(s_0)) \langle 0^n | s_0 | 0^n \rangle \\ &= \sum_{s \in P_n^{d+1}} (1-\gamma)^{|s|} s_d \Phi(C, s), \end{aligned} \tag{19}$$

where $|s|$ means the number of non-identities in s.

### C.1.2 PAULI NOISE

We use $\mathcal{E}$ to denote the noise function. Pauli noise $\mathcal{E}_{Pauli}$ is

$$\mathcal{E}_{\mathrm{Pauli}}(\rho) = \gamma_1 \rho + \gamma_2 X \rho X^\dagger + \gamma_3 Y \rho Y^\dagger + \gamma_4 Z \rho Z^\dagger, \tag{20}$$

where $\gamma_1 + \gamma_2 + \gamma_3 + \gamma_4 = 1$. The Pauli noise has the property that

$$\begin{aligned} \mathcal{E}_{\mathrm{Pauli}}(I) &= (\gamma_1 + \gamma_2 + \gamma_3 + \gamma_4)I = I, \\ \mathcal{E}_{\mathrm{Pauli}}(X) &= (\gamma_1 + \gamma_2 - \gamma_3 - \gamma_4)X = (1 - 2(\gamma_3 + \gamma_4))X, \\ \mathcal{E}_{\mathrm{Pauli}}(Y) &= (\gamma_1 - \gamma_2 + \gamma_3 - \gamma_4)Y = (1 - 2(\gamma_2 + \gamma_4))Y, \\ \mathcal{E}_{\mathrm{Pauli}}(Z) &= (\gamma_1 - \gamma_2 - \gamma_3 + \gamma_4)Z = (1 - 2(\gamma_2 + \gamma_3))Z. \end{aligned} \tag{21}$$

So the Pauli Channel can be written as

$$\begin{aligned} \rho &= \sum_{s \in P_n^{d+1}} s_d \mathrm{Tr}(s_d \mathcal{E}_{\mathrm{Pauli}}^{\otimes n} \mathcal{C}_d(s_{d-1})) \cdots \mathrm{Tr}(s_1 \mathcal{E}_{\mathrm{Pauli}}^{\otimes n} \mathcal{C}_1(s_0)) \mathrm{Tr}(s_0 | 0^n \rangle \langle 0^n |) \\ &= \sum_{s \in \mathcal{P}_n^{d+1}} (1 - 2(\gamma_3 + \gamma_4))^{|s|_X} (1 - 2(\gamma_2 + \gamma_4))^{|s|_Y} (1 - 2(\gamma_2 + \gamma_3))^{|s|_Z} s_d \Phi(C, s) \\ &\leq \sum_{s \in \mathcal{P}_n^{d+1}} (1 - 2\gamma)^{|s|} s_d \Phi(C, s), \end{aligned} \tag{22}$$

where $|s|_P$ denotes the number of $P$ in $s$. $\gamma = \min\{(\gamma_2 + \gamma_3), (\gamma_2 + \gamma_4), (\gamma_3 + \gamma_4)\}$. $\gamma$ still satisfying $0 < \gamma \leq 1$.

### C.1.3 NON-UNITAL NOISE

Angrisani et al. (2025) gives a way of simulating arbitrary noise by Pauli propagation. The normal form of a non-unital noise single-qubit channel $\mathcal{E}$ is decomposed as

$$\mathcal{E} = \mathcal{E}_{depo}^{\gamma} \circ \mathcal{E}', \tag{23}$$

where $\mathcal{E}'$ is a suitable (non-physical) linear map and $\mathcal{E}_{depo}^{\gamma}$ is a depolarizing noise with the effective depolarizing rate $\gamma = 1 - \chi_{\mathcal{D}}(\mathcal{E})$:

$$\chi_{\mathcal{D}}^2(\mathcal{E}) := \max_{A \subseteq [n]} \max_{x_A \neq 0 \text{ supp}(\rho_x^A) = A} \left( \mathbb{E}_{U \sim \mathcal{D}^{\otimes n}} \left[ \frac{\|\mathcal{E}^{\dagger \otimes n}(U^{\dagger} \rho_x^A U)\|_F^2}{\|\rho_x^A\|_F^2} \right] \right)^{1/|A|} \tag{24}$$

is the mean squared contraction coefficient of $\mathcal{E}$ in terms of the locally unbiased distribution $\mathcal{D}$. The input of the $\mathcal{C}$ is decomposed as $\rho_x = \sum_{P \in \mathcal{P}_n} \alpha_P P$. $\rho_x^A$ retains those Pauli terms whose support is exactly $A$: nontrivial on $A$ and identity elsewhere, which is different from the reduced density matrix. Define the squared normalized Frobenius norm $\|\rho\|_F^2 = \sum_{P \in \mathcal{P}_n} \alpha_P^2$, $|A|$ is the size of the $\text{supp}(\rho_x^A)$.

In that case, the output of a non-unital noisy channel is

$$\rho = \sum_{s \in \mathcal{P}_n^{d+1}} \text{Tr}(s_d \mathcal{E}_{\text{depo}}^{\gamma \otimes n} \mathcal{E}'^{\otimes n} \mathcal{C}_d(s_{d-1})) \cdots \text{Tr}(s_1 \mathcal{E}_{\text{depo}}^{\gamma \otimes n} \mathcal{E}'^{\otimes n} \mathcal{C}_1(s_0)) \text{Tr}(s_0 |0^n\rangle\langle 0^n|)$$

$$= \sum_{s \in P_n^{d+1}} (1-\gamma)^{|s|} s_d \Phi(\mathcal{C}, s). \tag{25}$$

Thus, complete the proof of Lemma 1.

### C.2 PROOF OF LEMMA 2

Aharonov et al. (2023) proved that sampling from a depolarizing channel reduces to fitting a constant number $l$ of Pauli paths. We generalize this observation to single-qubit Pauli noise and, further, to any i.i.d single-qubit non-unital noise that admits a sparse Pauli-path expansion.

According to Lemma 1, for an arbitrary i.i.d single-qubit noise, the output state is approximated by

$$\hat{\rho} = \sum_{|s_d| \leq l', s_d \in \mathcal{P}_n} \alpha_{s_d} s_d = \sum_{s \in \mathcal{P}_n^{d+1}, |s| \leq l} (1-\gamma)^{|s|} s_d \Phi(\mathcal{C}, s). \tag{26}$$

In other words, we can learn the finite number of the legal Pauli paths to get a $\hat{\rho}$ satisfying $\|\rho - \hat{\rho}\|_1 < \epsilon$, where $\|A\|_1$ is the Schatten $1-$norm of $A$. The formal statement and proof are given in Lemma 5.

**Lemma 5** (Restatement of Lemma 2). *Let the noisy quantum state $\rho = \mathcal{C}(|0^n\rangle\langle 0^n|)$ with $\mathcal{C} = \mathcal{E}^{\otimes n} \mathcal{C}_d \mathcal{E}^{\otimes n} \mathcal{C}_{d-1} \cdots \mathcal{E}^{\otimes n} \mathcal{C}_1$ representing a d-depth noisy quantum circuit, where $\mathcal{C}_i$ is a layer of two-qubit Haar random quantum gates. With nearly unit success probability, there exists a density matrix $\hat{\rho} = \sum_{|s_d| \leq l', s_d \in \mathcal{P}_n} \alpha_{s_d} s_d$ such that*

$$\|\rho - \hat{\rho}\|_1 < \epsilon_1, \tag{27}$$

*where coefficients $\alpha_{s_d} \in \mathbb{R}$ and $l' = \mathcal{O}(\log(1/(\epsilon_1 \delta_1))$ with the success probability $\geq 1 - \delta_1$.*

*Proof.*

$$\Delta := \|\rho - \hat{\rho}\|_F$$

$$= \sqrt{\text{Tr}\left( \left( \sum_{|s|>l} (1-\gamma)^{|s|} s_d \Phi(\mathcal{C}, s) \right) \left( \sum_{|s|>l} (1-\gamma)^{|s|} s_d \Phi(\mathcal{C}, s) \right)^{\dagger} \right)}$$

$$= \sqrt{\text{Tr}\left( \sum_{|s|>l} \sum_{|s'|>l} (1-\gamma)^{|s'|+|s|} s_d s_d'^{\dagger} \Phi(\mathcal{C}, s) \Phi(\mathcal{C}, s') \right)} \tag{28}$$

$$= \sqrt{\sum_{|s|>l} \sum_{|s'|>l} (1-\gamma)^{|s'|+|s|} \Phi(\mathcal{C}, s) \Phi(\mathcal{C}, s') \text{Tr}\left( s_d s_d'^{\dagger} \right)}$$

The above equation illustrates the constraint that the error of $\hat{\rho}$ receives from the sum of a series of constants. The orthogonality of the circuit, which is

$$\mathbb{E}[\Phi(\mathcal{C}, s)\Phi(\mathcal{C}, s')] = 0. \tag{29}$$

Furthermore, we have

$$
\begin{aligned}
\mathbb{E}_{\mathcal{C}}(\Delta^2) &= \mathbb{E}_{\mathcal{C}}\left(\sum_{|s|>l}\sum_{|s'|>l}(1-\gamma)^{|s'|+|s|}\Phi(\mathcal{C}, s)\Phi(\mathcal{C}, s')\mathrm{Tr}\left(s_d s_d'^{\dagger}\right)\right) \\
&= \mathbb{E}_{\mathcal{C}}\left(\sum_{|s|>l}(1-\gamma)^{2|s|}\Phi(\mathcal{C}, s)^2\mathrm{Tr}\left(s_d s_d^{\dagger}\right)\right) \\
&= \mathbb{E}_{\mathcal{C}}\left(\sum_{|s|>l}(1-\gamma)^{2|s|}\Phi(\mathcal{C}, s)^2\right) \\
&= \sum_{k>l}(1-\gamma)^{2k}W_k
\end{aligned} \tag{30}
$$

The second line is obtained via the orthogonality mentioned above. The third line uses the property that $\mathrm{Tr}\left(s_d s_d^{\dagger}\right) = 1$ where $s_d$ is the combination of the normalized Pauli operators. The last line denotes $W_k = \mathbb{E}_{\mathcal{C}, |s|=k}\Phi(\mathcal{C}, s)^2$. For $W_k$, it can be written as follows

$$
\begin{aligned}
W_k &= \mathbb{E}_{\mathcal{C}, |s|=k}\Phi(\mathcal{C}, s)^2 \\
&= \mathbb{E}_{\mathcal{C}, |s|=k}\left(\mathrm{Tr}(s_d\mathcal{C}_d(s_{d-1}))\cdots\mathrm{Tr}(s_1\mathcal{C}_1(s_0))\mathrm{Tr}(s_0|0^n\rangle\langle 0^n|)\right)^2 \\
&= 2^{-n}\mathbb{E}_{C_d}\left(\mathrm{Tr}(s_d\mathcal{C}_d(s_{d-1}))\cdots\mathrm{Tr}(s_1\mathcal{C}_1(s_0))\right)^2.
\end{aligned} \tag{31}
$$

The third line assumes that every $C_i$ is independent respectably, and $\mathrm{Tr}(s_0|0^n\rangle\langle 0^n|) = \frac{1}{\sqrt{2^n}}$.

For unital noises, using the equation

$$
\mathbb{E}_{U\sim\mathbb{SU}(4)}\mathrm{Tr}(xUyU^{\dagger})^2 = \begin{cases} 1, & x = y = I^{\otimes 2}/2, \\ 0, & x = I^{\otimes 2}/2, y \neq I^{\otimes 2}/2, \\ 0, & x \neq I^{\otimes 2}/2, y = I^{\otimes 2}/2, \\ \frac{1}{15}, & \text{else,} \end{cases} \tag{32}
$$

We observe that certain Pauli paths contribute 0 to the circuit; these are termed illegal Pauli paths.

For $k = 0$, $W_k = 1$, where the Pauli path $s$ consists of identity operators.

For $k \in (0, d]$, $W_k = 0$.

For $k \geq d + 1$, we can bound $W_k$ by focusing on every term, which is in the form of $\mathbb{E}_{C_i}\mathrm{Tr}(s_i\mathcal{C}_i(s_{i-1}))^2$.

Noting that each $C_i$ is a layer of two-qubit gates, $C_i$ is equal to the multiplication of $C_i^{(j)}$, where $j$ indexes the two-qubit gates in the layer and $N_g$ is the total number of such gates in a layer. So

$$
\begin{aligned}
\mathbb{E}_{C_i}\mathrm{Tr}(s_i\mathcal{C}_i(s_{i-1}))^2 &= \bigotimes_j^{N_g}\mathbb{E}_{C_i^{(j)}}\left(\mathrm{Tr}(s_i^{(j)}s_i^{(j+1)}C_i^{(j)}s_{i-1}^{(j)}s_{i-1}^{(j+1)}C_i^{(j)\dagger})\right)^2 \\
&\leq \left(\frac{1}{15}\right)^{\frac{|s_i|}{2}}.
\end{aligned} \tag{33}
$$

The last line is due to that one gate introduces at most 2 non-identity Pauli operators to the path when $k \geq d + 1$. Since any single gate can be accountable for at most four non-identity entries (two incoming and two outgoing), the number of two-qubit gates that actually contribute to the suppression factor $1/15$ is at least $\frac{|s|}{4}$.

In that case, Eq. 31 is bounded by:

$$W_k \leq \left(\frac{1}{15}\right)^{\frac{k}{4}} \left(\frac{1}{2}\right)^n \tag{34}$$

Since the $\left(\frac{1}{15}\right)^{\frac{k}{4}} < \left(\frac{1}{2}\right)^{\frac{3k}{4}}$ is a decreasing sequence, we can get $\sum_k \left(\frac{1}{2}\right)^{\frac{3k}{4}} \leq \mathcal{O}(1)$, which results in $\sum_{k>l} \left(\frac{1}{2}\right)^{\frac{3k}{4}} \leq \mathcal{O}(1)$.

For non-unital noise, the $\mathcal{E}'$ is not a CPTP map, so the preceding argument does not apply directly. Lemma 10 of Angrisani et al. (2025) shows that for $\gamma = 1 - \chi_{\mathcal{D}}(\mathcal{E})$, the (non-physical) linear map $\mathcal{E}'$ does not increase the Frobenius norm on average.

**Lemma 6** (Non-unital Noise, Lemma 10 of Angrisani et al. (2025)). *Let $\mathcal{D}$ be a 1-design over $\mathbb{SU}(2)$ and let $\gamma = 1 - \chi_{\mathcal{D}}(\mathcal{E})$. For all observables $O$, we have*

$$\mathbb{E}_{V \sim \mathcal{D}^{\otimes n}} \left\| \mathcal{E}'^{\dagger \otimes n}(V^{\dagger}OV) \right\|_{\mathrm{F}}^2 \leqslant \|O\|_{\mathrm{F}}^2, \tag{35}$$

*which shows the linear map $\mathcal{E}'$ does not increase the Frobenius norm in expectation over a randomly sampled $V$.*

For an $i$-th layer of $\mathcal{C}$, $\mathcal{C}_i = V_i \circ G_i$, where $V_i \sim \mathcal{D}^{\otimes n}$ and $G_i$ acts on $\mathcal{O}(1)$ qubits.

Consequently, $\sum_{k>l} W_k = 2^{-n} \mathcal{O}(1) \| \left| 0 \right\rangle^n \left\langle 0 \right|^n \|_F = 2^{-n} \mathcal{O}(1)$. Because the Frobenius norm of $\Phi(\mathcal{C}, s)$ remains bounded, the Pauli-path expansion of a non-unital noisy circuit can be truncated at finite weight.

By inequality between Schatten $\tau$-norms, we have

$$||\rho||_1 \leq 2^{n/2} ||\rho||_F \tag{36}$$

So the $\mathbb{E}_{\mathcal{C}}(||\rho - \hat{\rho}||_1^2)$ is bounded as:

$$\begin{aligned}
\mathbb{E}_{\mathcal{C}}(||\rho - \hat{\rho}||_1^2) &\leq 2^n \mathbb{E}_{\mathcal{C}}(\Delta^2) \\
&= 2^n \sum_{k>l} (1-\gamma)^{2k} W_k \\
&\leq 2^n \sum_{k>l} (1-\gamma)^{2l} W_k \\
&\leq 2^n (1-\gamma)^{2l} 2^{-n} \mathcal{O}(1) \\
&\leq e^{-2\gamma l} \mathcal{O}(1).
\end{aligned} \tag{37}$$

By Markov's inequality,

$$\mathbb{P}(||\rho - \hat{\rho}||_1 \geq \epsilon_1) \leq \frac{\mathbb{E}(||\rho - \hat{\rho}||_1)}{\epsilon_1} = \delta_1. \tag{38}$$

Hence, choosing $l \approx \mathcal{O}(\frac{1}{\gamma} \log \frac{1}{\epsilon_1 \delta_1})$, yields $\Delta \leq \epsilon_1$ with success probability $\geq 1 - \delta$. $\qquad\square$

## C.3 Algorithm of Learning a Quantum State

For the first problem, there are several ways to get the $\hat{\rho}$. The sections following introduce 2 methods, including computing directly by classical shadow (Huang et al., 2020), and a way of learning alpha based on Huang et al. (2024)

### C.3.1 Compute Directly

As shown before, $\rho = \sum_{s_d \in P_n} \alpha_{s_d} s_d$, where $s_d \in \mathcal{P}_n$. In that case,

$$\begin{aligned}
\alpha_{s_d} &= \mathrm{Tr}(\rho s_d) \\
&= \mathrm{Tr}(\sum_{s_d' \in \mathcal{P}_n} \alpha_{s_d'} s_d' s_d) \\
&= \sum_{s_d' \in \mathcal{P}_n} \alpha_{s_d'} \mathrm{Tr}(s_d' s_d) \\
&= \alpha_{s_d}.
\end{aligned} \tag{39}$$

The fourth line uses

$$\mathrm{Tr}(s_d s_d') = \begin{cases} 0, & \text{if } s_d \neq s_d', \\ 1, & \text{if } s_d = s_d'. \end{cases} \tag{40}$$

Thus, $\alpha_{s_d}$ is obtained by evaluating $\mathrm{Tr}(\rho s_d)$, where $\rho$ is estimated via classical shadows. Using a set of POVMs (Positive Operator-Valued Measures) such as the random Pauli basis that measures each qubit and yields outcomes $|b\rangle \in \{0,1\}^n$, the classical shadow is constructed as $\tilde{\rho} = \otimes_{j=1}^n \left( 3 P_j^\dagger |b_j\rangle \langle b_j| P_j - I \right)$, immediately gives $\alpha_{s_d} = \mathrm{Tr}(\tilde{\rho} s_d)$.

### C.3.2 QUANTUM STATE TOMOGRAPHY

---

**Algorithm 2** Quantum State Learning Algorithm

---

**Input:** Data set $\mathcal{D}_{\mathrm{QST}} = \{|\psi_j\rangle = \otimes_{i=1}^n |\psi_{j,i}\rangle\}_{j=1}^{N_{\mathrm{data}}}$ and accuracy parameter $\epsilon$.
**Output:** $\hat{\rho}$ such that $T(\rho, \hat{\rho}) \leq \epsilon$.
Let $l' = [\log(1/\epsilon)]$, enumerate all the legal $s_d \in \mathcal{P}_n$ with $|s_d| \leq l'$.
**For** $j \in [N_{\mathrm{data}}]$:
  Using the SWAP-test to obtain the overlap $v_i$ of $\rho$ and $|\psi_j\rangle$.
**End For**
**For** each legal $s_d$:
  Compute $\alpha_{s_d} = \frac{3^{|s_d|}}{N_{\mathrm{data}}} \sum_{j=1}^{N_{\mathrm{data}}} v_j \langle \psi_j| s_d |\psi_j\rangle$,
**End For**
**Output**: $\hat{\rho} = \sum_{|s_d| \leq l'} \alpha_{s_d} s_d$
**End**

---

This section is mainly about a way of learning $\alpha$ based on Huang et al. (2024), which introduces a classical dataset to reconstruct the channel's output. Our results are given below.

**Theorem 3** (Noisy Quantum State Learning). *For any noisy quantum state $\rho$ prepared by a noisy quantum circuit $\mathcal{C}$ (Eq. 4), there exists a learning algorithm that can efficiently solve Problem 1 with success probability $\geq 1 - \delta$. The learning algorithm requires sample complexity $N_{\mathrm{data}} = 6^{\mathcal{O}\left(\gamma^{-1} \log\left(\epsilon^{-1} \delta^{-1}\right)\right)} \log(1/\delta) \epsilon^{-2}$ and classical post-processing complexity $24^{\mathcal{O}\left(\gamma^{-1} \log\left(\epsilon^{-1} \delta^{-1}\right)\right)} \log(1/\delta) \epsilon^{-2}$.*

Details of our method are as follows.

Let $\mathrm{Stab}$ be a list of single-qubit stabilizers:

$$\mathrm{Stab} = \{|0\rangle, |1\rangle, |+\rangle, |-\rangle, |y+\rangle, |y-\rangle\}. \tag{41}$$

Let $\{|\psi_j\rangle = \otimes_{i=1}^n |\psi_{i,j}\rangle\}_{j=1}^{N_{\mathrm{data}}}$, where $|\psi_{i,j}\rangle \in \mathrm{Stab}$.

$$\begin{aligned}
&\mathbb{E}_{|\psi_j\rangle \sim \mathrm{Stab}^{\otimes n}} \langle \psi_j| \mathcal{C}(|0^n\rangle \langle 0^n|) |\psi_j\rangle \langle \psi_j| s_d |\psi_j\rangle \\
&= \sum_{|s_d| \leq l_s} \alpha_{s_d} \mathbb{E}_{|\psi_j\rangle \sim \mathrm{Stab}^{\otimes n}} \langle \psi_j| s_d |\psi_j\rangle \langle \psi_j| s_d |\psi_j\rangle \\
&= \sum_{|s_d| \leq l_s} \alpha_{s_d} \mathbb{E}_{U \sim U(2)} \bigotimes_{i=1}^n \langle 0| U_{i,j}^\dagger s_d U_{i,j} |0\rangle \langle 0| U_{i.j}^\dagger s_d U_{i,j} |0\rangle \\
&= \frac{\alpha_{s_d}}{3^{|s_d|}} \bigotimes_{i=1}^n \sum_{Q \in \{X,Y,Z\}} \langle 0^2| Q \otimes Q |0^2\rangle \\
&= \frac{\alpha_{s_d}}{3^{|s_d|}}.
\end{aligned} \tag{42}$$

The third line employs $|\psi_j\rangle = \otimes_{i=1}^n |\psi_{i,j}\rangle = \otimes_{i=1}^n U_{i,j} |0\rangle$, where $U_{i,j} \sim \mathrm{Cl}(2)$. The fourth line uses

$$\mathbb{E}_{U_{i,j}\sim\mathrm{Cl}(2)}\left[U_{i,j}^{\dagger\otimes 2}(Q_i \otimes Q_i')U_{i,j}^{\otimes 2}\right] = \begin{cases} I^{\otimes 2}, \text{ if } Q_i = Q_i' = I, \\ \dfrac{1}{3}\displaystyle\sum_{Q_i\in\{X,Y,Z\}^{\otimes 2}}(Q_i \otimes Q_i)\,, \text{ if } Q_i = Q_i' \neq I. \\ 0, \text{ if } Q_i \neq Q_i', \end{cases} \tag{43}$$

Therefore, $\alpha_{s_d}$ can be calculated by

$$\alpha_{s_d} = 3^{|s_d|}\mathbb{E}_{|\psi_j\rangle\sim\mathrm{Stab}^{\otimes n}}\langle\psi_j|\rho|\psi_j\rangle\langle\psi_j|s_d|\psi_j\rangle$$
$$\approx \frac{3^{|s_d|}}{N_{\mathrm{data}}}\sum_{j=1}^{N_{\mathrm{data}}}\langle\psi_j|\rho|\psi_j\rangle\langle\psi_j|s_d|\psi_j\rangle. \tag{44}$$

The first part of the summation term (of the form $\langle\psi_i|\rho|\psi_i\rangle$) can be obtained by using the SWAP-test method, while the latter part can be derived through classical post-processing. The data complexity $N_{\mathrm{data}}$ is $6^{\mathcal{O}(\gamma^{-1}\log\frac{1}{\epsilon\delta})}\epsilon^{-2}\log\frac{1}{\delta}$, with failure probability $\delta$. The details of the proof are in Appendix D.2 The quantum state learning procedure is presented as Algorithm 2.

# D    LEARNING A QUANTUM PROCESS CHARACTERIZATION

## D.1    PROOF OF LEMMA 3

This section is to give a proof of Lemma 3, which is

**Lemma 7** (Restatement of Lemma 3). *Let the noisy quantum circuit $\mathcal{C} = \mathcal{E}^{\otimes n}\mathcal{C}_d\mathcal{E}^{\otimes n}\mathcal{C}_{d-1}\cdots\mathcal{E}^{\otimes n}\mathcal{C}_1$ represent a $d$-depth noisy quantum circuit, where $\mathcal{C}_i$ is a layer of two-qubit Haar random quantum gates and $\mathcal{E}$ represents an i.i.d single-qubit noisy channel (unital or non-unital). With nearly unit success probability $\geq 1 - \delta_2$, there exists an operator $\mathcal{C}^{(l')\dagger}(O) = \sum_{|P|\leq l',P\in\mathcal{P}_n}\beta_P P$ such that*

$$\left\|\mathcal{C}^{(l')\dagger}(O) - \mathcal{C}^\dagger(O)\right\|_F \leq \epsilon_2, \tag{45}$$

*where coefficients $\beta_P \in \mathbb{R}$ and $l' = \mathcal{O}\left(\gamma^{-1}\log(1/(\delta_2\epsilon_2))\right)$.*

*Proof.* Given $O = \sum_{P\in\mathcal{P}_n}\alpha_P P$, we have

$$\mathcal{C}^\dagger(O) = \sum_{s\in\mathcal{P}_n^d}(1-\gamma)^{|s|}\Phi(\mathcal{C},s)s_0, \tag{46}$$

where

$$\Phi(\mathcal{C},s) = \begin{cases} \mathrm{Tr}(s_1\mathcal{C}_1(s_0))\cdots\mathrm{Tr}(s_d\mathcal{C}_d(s_{d-1}))\mathrm{Tr}(s_dO), & \text{unital,} \\ \mathrm{Tr}(s_1\mathcal{E}'^{\otimes n}\mathcal{C}_1(s_0))\cdots\mathrm{Tr}(s_d\mathcal{E}'^{\otimes n}\mathcal{C}_d(s_{d-1}))\mathrm{Tr}(s_dO), & \text{non-unital} \end{cases} \tag{47}$$

Considering the unital noise, let

$$\mathbb{E}(\Delta)^2 := \|\mathcal{C}^{(k)\dagger}(O) - \mathcal{C}^\dagger(O)\|^2$$
$$= \sum_{k>l}(1-\gamma)^{2k}W_k, \tag{48}$$

where $W_k$ is

$$W_k = \mathbb{E}_{\mathcal{C},|s|=k}\Phi(\mathcal{C},s)^2$$
$$= \mathbb{E}_{\mathcal{C},|s|=k}\left(\mathrm{Tr}(s_1\mathcal{C}_1(s_0))\cdots\mathrm{Tr}(s_d\mathcal{C}_d(s_{d-1}))\mathrm{Tr}(s_dO)\right)^2$$
$$= 2^{-n}\alpha_{s_d}^2\mathbb{E}_{C_1}(\mathrm{Tr}(s_1\mathcal{C}_1(s_0)))^2...\mathbb{E}_{C_d}(\mathrm{Tr}(s_d\mathcal{C}_d(s_{d-1})))^2 \tag{49}$$
$$\leq 2^{-n}\alpha_{s_d}^2\left(\frac{1}{15}\right)^{\frac{k}{4}}$$

The third line is due to that $\mathrm{Tr}(s_d O) = 2^{-n}\alpha_{s_d}^2$. In that case, there is

$$\mathbb{E}(\Delta^2)$$

$$\leq \sum_{k>l} 2^{-n}(1-\gamma)^{2k}\alpha_{s_d}^2 \left(\frac{1}{15}\right)^{\frac{k}{4}}$$

$$\leq \sum_{k>l} (1-\gamma)^{2l} \left(\frac{1}{2}\right)^{\frac{3k}{4}} \|O\|_F \tag{50}$$

$$\leq e^{-2\gamma l} \|O\|_F \mathcal{O}(1).$$

Considering the non-unital noise, Angrisani et al. (2025) has shown that the non-unital noisy circuit can be truncated by the low-weight Pauli integral because of the Theorem 5 in Angrisani et al. (2025) shown below.

**Lemma 8** (Non-unital Noisy Circuit Path Truncation, Theorem 5 in Angrisani et al. (2025)). *Let $\mathcal{D}_{circ}$ be an $d$-layered locally unbiased distribution over noisy circuits, and let $\gamma$ be the effective depolarizing rate of $\mathcal{D}_{circ}$. We have*

$$\mathbb{E}_{\mathcal{C}\sim\mathcal{D}_{circ}} \left| \mathrm{Tr}\left[ \left(\mathcal{C}^\dagger(O) - \mathcal{C}^{(l')\dagger}(O)\right)\rho_x \right] \right|^2 \leq (1-\gamma)^{2l'}\|O\|_F^2. \tag{51}$$

It conveys that the non-unital noisy process can be simulated by a low-Pauli weight. For our problem, the gate of the circuit is the random two-qubit gate, which belongs to $\mathcal{D}_{circ}$. The last proof is similar to the Appendix C.2. Therefore when $l' = \mathcal{O}(l) = \mathcal{O}\left(\gamma^{-1}\log\left(\frac{\|O\|_F}{\epsilon_2\delta_2}\right)\right)$, $\Delta \leq \epsilon_2$ is satisfied with the success probability $\geq 1 - \delta_2$. $\qquad\square$

### D.2 PROOF OF THEOREM 2

In this section, we will prove the main result of our learning algorithm.

**Theorem 4** (Noisy Quantum Process Learning). *For any noisy quantum process $\mathcal{C}$ defined as Eq. 4, where $\mathcal{C}_i$ is a layer of two-qubit Haar random quantum gates, and $n$-qubit observable $O = \sum_{Q\in\{I,X,Y,Z\}^{\otimes n}, |Q|=\mathcal{O}(1)} \mathrm{Tr}[OQ]Q/2^n$, there exists a learning algorithm that can efficiently solve Problem 2 with success probability $\geq 1-\delta$. The learning algorithm requires sample complexity*

$$N_{\mathrm{data}} = \max_{Q\in O}(|Q|^2)6^{\mathcal{O}\left(\gamma^{-1}\log\left(\|O\|_F\epsilon^{-1}\delta^{-1}\right)\right)}\log\left(\delta^{-1}\right)\epsilon^{-2}, \tag{52}$$

*and classical post-processing complexity* $\mathcal{O}\left(\frac{n\cdot\max_{Q\in O}(|Q|^3)24^{\mathcal{O}\left(\gamma^{-1}\log\left(\|O\|_F\epsilon^{-1}\delta^{-1}\right)\right)}\log\left(\delta^{-1}\right)}{\epsilon^2}\right)$.

*Moreover, if the noise is unital, the sample complexity is* $6^{\mathcal{O}\left(\gamma^{-1}\log\left(\frac{\|O\|_F}{\epsilon\delta}\right)\right)}\log\left(\delta^{-1}\right)\epsilon^{-2}$ *and classical post-processing complexity is* $\mathcal{O}\left(n\cdot 24^{\mathcal{O}\left(\gamma^{-1}\log\left(\frac{\|O\|_F}{\epsilon\delta}\right)\right)}\log\left(\delta^{-1}\right)\epsilon^{-2}\right)$.

*Proof.* The discrepancy between the algorithm's learned outcome and the true value, quantified via absolute value, encompasses two types of errors: truncation error and learning error.

$$|f(\rho_x) - \mathrm{Tr}(\mathcal{C}(\rho_x)O| = \left| \sum_{|P|\leq l'} \hat{\beta}_P \mathrm{Tr}(\rho_x P) - \mathrm{Tr}(\mathcal{C}(\rho_x)O) \right|$$

$$\leq \left| \sum_{|P|\leq l'} \beta_P \mathrm{Tr}(\rho_x P) - \mathrm{Tr}(\mathcal{C}(\rho_x)O)) \right| + \left| \sum_{|P|\leq l'} \hat{\beta}_P \mathrm{Tr}(\rho_x P) - \sum_{|P|\leq l'} \beta_P \mathrm{Tr}(\rho_x P) \right|. \tag{53}$$

The inequality is derived through the application of the triangle inequality, where the first term on the right-hand side of the inequality represents the truncation error, and the second term represents the learning error. $\hat{\beta}_P$ denotes the learned value of $\beta_P$.

The proof for the truncation error can be analogously extended from that in Appendix D.1, demonstrating that when $l' = \mathcal{O}(\gamma^{-1} \log \frac{1}{\epsilon_2 \delta_2})$, $\left| \sum_{|P| \leq l'} \beta_P \mathrm{Tr}(\rho_x P) - \mathrm{Tr}(\mathcal{C}(\rho_x) O) \right| \leq \epsilon_2$.

The learning error is bounded by

$$
\begin{aligned}
&\left| \left( \sum_{|P| \leq l'} \hat{\beta}_P - \sum_{|P| \leq l'} \beta_P \right) \mathrm{Tr}(\rho_x P) \right| \\
&\leq \left| \sum_{|P| \leq l'} \hat{\beta}_P - \sum_{|P| \leq l'} \beta_P \right| \\
&= \sum_{|P| \leq l'} \left| \hat{\beta}_P - \beta_P \right| \\
&\leq N_s |\hat{\beta}_P - \beta_P| \\
&\leq \epsilon_3.
\end{aligned}
\tag{54}
$$

Combining the equation with Hoeffding's inequality, we can derive that given a dataset of size $N_{\text{data}} = \frac{3^{\mathcal{O}(l')} N_s^2}{\epsilon_3^2} \log \frac{1}{\delta}$ with probability at least $1 - \delta$, Eq. 54 is valid.

**Lemma 9** (Number of the Legal Pauli Paths). *For any noisy quantum process $\mathcal{C}$ defined as Eq. 4 and $n$-qubit observable $O = \sum_{Q \in \{I,X,Y,Z\}^{\otimes n}, |Q| = \mathcal{O}(1)} \mathrm{Tr}[OQ] Q / 2^n$, the number of the legal Pauli paths, denoted as $N_s$ is $\max_{Q \in O}(|Q|) 2^{\mathcal{O}(l')}$. When the noise is unital, $N_s = 2^{\mathcal{O}(l')}$.*

The proof of Lemma 9 is provided in the next section. Considering Lemma 9, for an arbitrary i.i.d single-qubit noise, given $\epsilon_2 = \epsilon_3$, the sample complexity can be further expressed as

$$
N_{\text{data}} = \max_{Q \in O}(|Q|^2) 6^{\mathcal{O}\left( \gamma^{-1} \log\left( ||O||_F \epsilon^{-1} \delta^{-1} \right) \right)} \log\left( \delta^{-1} \right) \epsilon^{-2}.
\tag{55}
$$

For the runtime complexity of classical post-processing, the calculation is derived directly from Algorithm 1. The dominant factor in the runtime is the computation of the coefficients $\beta_P$, which involves nested iterations over $N_{\text{data}}$ input samples and $N_s$ Pauli strings. The internal calculation of the expectation value $\langle \psi_j | P | \psi_j \rangle$ has a cost of $\mathcal{O}(n)$, because the input state $|\psi_j\rangle$ is a product state and $P$ is a Pauli string, allowing the expectation value to be computed via the product of $n$ single-qubit terms.
Thus, the total complexity is:

$$
\mathcal{O}\left( n \cdot N_{\text{data}} \cdot N_s \right) = \mathcal{O}\left( n \cdot \max_{Q \in O}(|Q|^3) 24^{\mathcal{O}\left( \gamma^{-1} \log\left( \frac{||O||_F}{\epsilon \delta} \right) \right)} \log\left( \delta^{-1} \right) \epsilon^{-2} \right),
\tag{56}
$$

where the factor $n$ accounts for the linear cost of evaluating the $n$ single-qubit terms that constitute $\langle \psi_j | P | \psi_j \rangle$ for each pair of sample $|\psi_j\rangle$ and Pauli string $P$.
Consequently, the runtime complexity scales as $\mathcal{O}(n \cdot \mathrm{poly}(1/\epsilon, 1/\gamma))$. Specifically, if the noise is unital, the sample complexity is $6^{\mathcal{O}\left( \gamma^{-1} \log\left( \frac{||O||_F}{\epsilon \delta} \right) \right)} \log\left( \delta^{-1} \right) \epsilon^{-2}$ and classical post-processing complexity is $\mathcal{O}\left( n \cdot 24^{\mathcal{O}\left( \gamma^{-1} \log\left( \frac{||O||_F}{\epsilon \delta} \right) \right)} \log\left( \delta^{-1} \right) \epsilon^{-2} \right)$.

$\square$

### D.3 NUMBER OF THE LEGAL PAULI PATHS

Focusing on the number of the legal Pauli paths, denoted $N_s$, the basic idea is to enumerate all combinations that satisfy the rule. Once the non-identity positions in one layer are fixed, those in the next layer are also fixed because a legal Pauli path requires the input and the output of every gate to be either both identities or both non-identities. Starting from the first layer, the positions and count of non-identities therefore match those of the input. For a local term $Q \in O$ acting non-trivially on a constant number of qubits, $N_s$ is bound by $\max_{Q \in O} |Q| 2^{\mathcal{O}(l')}$.

Specially, in QST, since the input $|0^n\rangle\langle 0^n| = \frac{1}{2^n}\sum_{P\in\{I,Z\}}\max_{Q\in O}|Q| \neq \mathcal{O}(1)$, so the previous bound cannot be used directly. Instead, we bound $N_s$ by showing that $\tilde{\mathcal{C}}_1^{(l_0)}(|0^n\rangle\langle 0^n|) = \sum_{s_1,s_0\in\mathcal{P}_n}\text{Tr}(s_1\mathcal{E}'^{\otimes n}\mathcal{C}_1(s_0))\langle 0^n|s_0|0^n\rangle s_1$ is sparse after an $l_0$-cutoff. Concretely, we prove

$$||\tilde{\mathcal{C}}_1(|0^n\rangle\langle 0^n|) - \tilde{\mathcal{C}}_1^{(l_0)}(|0^n\rangle\langle 0^n|)||_1 \leq \epsilon, \tag{57}$$

so that $|Q| = \mathcal{O}(1)$ for $Q \in \tilde{\mathcal{C}}_1^{(l_0)}(|0^n\rangle\langle 0^n|)|$.

When $d = 1$, Eq. 57 suffices by Lemma 2. Hence, the number of legal Pauli paths in QST is $\mathcal{O}(1)2^{\mathcal{O}(l')} \approx 2^{\mathcal{O}(l')}$.

For unital noise, a tighter bound is available. $W_k$ exists a lower bound when $l \geq d + 1$, which is $W_k \geq \left(\frac{1}{15}\right)^k$.

Since $W_k \leq (\frac{1}{2})^{3\lceil\frac{k}{2}\rceil}$, $\sum_{k=d+1}^l W_k = \mathcal{O}(1)$. Furthermore,

$$\begin{aligned}
\mathcal{O}(1) &= \sum_{k=d+1}^l W_k + W_0 \\
&\geq \sum_{k=d+1}^l (\frac{1}{15})^k + 1 \\
&\geq \sum_{k=d+1}^l (\frac{1}{15})^l + 1 \\
&= (\frac{1}{15})^l N_{|s|\in[d+1,l]} + 1.
\end{aligned} \tag{58}$$

where $N_{|s|\in[d+1,l]}$ denotes the legal Pauli paths except all identity one. The number of Pauli paths needed is

$$N_s = N_{|s|\in[d+1,l]} + 1 = \mathcal{O}(1)15^l = 2^{\mathcal{O}(l)}. \tag{59}$$

Here we focus on the learning algorithm, so only the number of $s_d$ is concerned. Since different $s$ may contain the same $s_d$, the number of Pauli paths is no less than the number of combinations of Pauli operators in $s_d$. We denote by $N_s$ an upper bound on the quantity $s_d$ that is independent of the system size, and by $l'$ the maximum hamming weight of $s_d$, with $l' = l - d = \mathcal{O}(l)$ due to the enumeration strategy in Aharonov et al. (2023).

# E    SAMPLE COMPLEXITY LOWER BOUND FOR THE WORST-CASE SCENARIO

The main manuscript essentially considers learning an efficient classical representation of noisy quantum states and processes in the average-case scenario. As we claimed in Theorems 1 and 2, the tasks of learning noisy quantum states and performing tomography are highly efficient in the average-case setting. However, this does not rule out intrinsic hardness in the worst case. Here we theoretically demonstrate that learning noisy quantum states prepared by quantum circuits subject to constant-strength noise channels is quantum-hard in the worst-case scenario.

The fundamental idea relies on constructing a polynomial reduction to the quantum state discrimination problem.

**Task 1.** *Consider two pure quantum states $\rho_0$ and $\rho_1$, and a noisy quantum circuit $\mathcal{C}$ with depth $d$, where Each quantum circuit is affected by by $\gamma$-strength Pauli channel in each layer. Suppose that a distinguisher is given access to copies of the quantum states $\mathcal{C}(\rho_0)$ and $\mathcal{C}(\rho_1)$, then what is the fewest number of copies sufficing to identify these two noisy quatum states with high probability?*

Obviously, if one can perform quantum state tomography on these noisy states, then efficient classical representations of the noisy states are obtained. Using these classical representations, one can easily distinguish the noisy states $\mathcal{C}(\rho_0)$ from $\mathcal{C}(\rho_1)$ easily. As a result, Task 1 can be used to benchmark the sample-complexity lower bound for the noisy quantum state tomography problem. We state the result below.

**Theorem 5.** *Given an unknown noisy quantum state $\rho$ prepared by a $d$-depth quantum circuit affected by $\gamma$-strength local Pauli noise channels, then any algorithm designed to learn an efficient representation to $\rho$ requires at least $m$ samplings in the* worst-case scenario*, where*

$$m = \frac{(1-\gamma)^{-2cd}(1-\eta)^2}{2n},$$

*where $c = 1/(2\ln 2)$ and constant $\eta \in \mathcal{O}(1)$.*

When the noise strength $\gamma = \mathcal{O}(1)$, and quantum circuit depth $d \geq \text{poly}\log(n)$, the sample complexity required for quantum state tomography grows at least quasi-polynomially with the system size in the worst-case scenario. We emphasize that this result does not contradict Theorem 1 and 2: the former statement concerns the worst case, while the latter addresses the average case under the random-circuit assumption.

In the quantum process tomography task, when $O \geq 0$, the target is to learn a classical representation to $\mathcal{C}^\dagger[O]$ which can be easily reduced to a density matrix learning task by setting $\rho = \mathcal{C}^\dagger[O]/\text{Tr}[\mathcal{C}^\dagger[O]]$. This justifies the statement that noisy process tomography (for this observable $O$) is no easier than state tomography.

To support the proof of our result, we require the following lemmas.

**Lemma 10** (Lemma 6 in Wang et al. (2021)). *Consider a single instanoise channel $\mathcal{N} = \mathcal{N}_1 \otimes \cdots \otimes \mathcal{N}_n$ where each local noise channel $\{\mathcal{N}_j\}_{j=1}^n$ is a Pauli noise channel that satisfies $\mathcal{N}_j(\sigma) = q_\sigma \sigma$ for $\sigma \in \{X, Y, Z\}$ and $q_\sigma$ be the Pauli strength. Then we have*

$$D_2\left(\mathcal{N}(\rho)\|\frac{I^{\otimes n}}{2^n}\right) \leq q^{2c} D_2\left(\rho\|\frac{I^{\otimes n}}{2^n}\right), \tag{60}$$

*where $D_2(\cdot\|\cdot)$ represents the 2-Renyi relative entropy, $q = \max_\sigma q_\sigma$ and $c = 1/(2\ln 2)$.*

**Lemma 11.** *Given an arbitrary $n$-qubit density matrix and maximally mixed state $I^{\otimes n}/2^n$, we have*

$$D\left(\rho\|I^{\otimes n}/2^n\right) \leq D_2\left(\rho\|I^{\otimes n}/2^n\right), \tag{61}$$

*where $D(\cdot\|\cdot)$ denotes the relative entropy and $D_2(\cdot\|\cdot)$ denotes the 2-Renyi relative entropy.*

*Proof:* Given quantum states $\rho$ and $\sigma$, the quantum 2-Renyi entropy

$$D_2(\rho\|\sigma) = \log \text{Tr}\left[\left(\sigma^{-1/4}\rho\sigma^{-1/4}\right)^2\right]. \tag{62}$$

When $\sigma = I^{\otimes n}/2^n$, we have $D_2(\rho\|I^{\otimes n}/2^n) = \log \text{Tr}\left[\left((I^{\otimes n}/2^n)^{-1}\rho^2\right)\right] = n + \log \text{Tr}[\rho^2]$. Noting that the function $y = x^2 - x\log x \geq 0$ when $x \in [0,1]$, and this implies $\text{Tr}(\rho^2) \geq \text{Tr}(\rho\log\rho)$. Finally, we have

$$D\left(\rho\|I^{\otimes n}/2^n\right) = n + \text{Tr}\left[\rho\log\rho\right] + n \leq \text{Tr}\left[\rho^2\right] + n = D_2\left(\rho\|I^{\otimes n}/2^n\right). \tag{63}$$

*Proof of Theorem 5:* Now we prove the sample complexity lower bound to the noisy quantum state tomography task. We consider the sample complexity $m$ in distinguishing quantum states $\mathcal{C}(\rho_0)$ and $\mathcal{C}(\rho_1)$. When their trace distance is quite large, let $\eta \in (0,1)$ and we have

$$1 - \eta \leq \frac{1}{2}\left\|\mathcal{C}(\rho_0)^{\otimes m} - \mathcal{C}(\rho_1)^{\otimes m}\right\|_1$$

$$\leq \frac{1}{2}\left(\left\|\mathcal{C}(\rho_0)^{\otimes m} - (I_n/2^n)^{\otimes m}\right\|_1 + \left\|\mathcal{C}(\rho_1)^{\otimes m} - (I_n/2^n)^{\otimes m}\right\|_1\right) \tag{64}$$

$$\leq \frac{1}{\sqrt{2}}\left(D^{1/2}\left(\mathcal{C}(\rho_0)^{\otimes m}\|(I_n/2^n)^{\otimes m}\right) + D^{1/2}\left(\mathcal{C}(\rho_1)^{\otimes m}\|(I_n/2^n)^{\otimes m}\right)\right),$$

where the second line comes from the triangle inequality and the third line comes from the Pinsker's inequality. Using Lemmas 10 and 11, we have

$$1 - \eta \leq \frac{1}{\sqrt{2}}\left(D_2^{1/2}\left(\mathcal{C}^{\otimes m}(\rho_0)\|(I_n/2^n)^{\otimes m}\right) + D_2^{1/2}\left(\mathcal{C}^{\otimes m}(\rho_1)\|(I_n/2^n)^{\otimes m}\right)\right)$$

$$\leq \frac{\sqrt{nm}}{\sqrt{2}}((1-\gamma)^{cd} + (1-\gamma)^{cd}) \tag{65}$$

$$\leq \sqrt{2nm}(1-\gamma)^{cd},$$

As a result we have

$$m \geq \frac{(1-\gamma)^{-2cd}(1-\eta)^2}{2n}. \tag{66}$$

## F EXPERIMENT RESULT

### F.1 NUMERICAL EXPERIMENT FOR HIGHLY ENTANGLED INPUT STATE

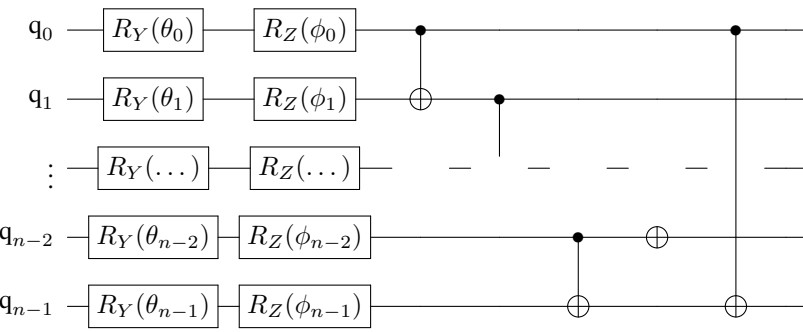

Figure 4: The demonstration of a layer of the state preparation circuit for $n$ qubits. The structure consists of parameterized single-qubit rotations followed by a cyclic CNOT entangling layer.

To further underscore the novelty of our **input-agnostic regime**, we constructed a Quantum Process Tomography (QPT) experiment using an input state that is **highly entangled** and falls outside the distribution set addressed by previous work (Huang et al., 2023a). The method detailed in (Huang et al., 2023a) requires the input distribution to be at most polynomially far from a **locally flat distribution**.

As demonstrated by (Huang et al., 2023a), locally flat distributions encompass: Random product states, ground and thermal states of random local Hamiltonians and any state generated by a circuit whose final layer consists of random single-qubit gates.

Here, we intentionally generated the input state $\rho_x$ using a two-layer parameterized circuit to ensure high entanglement. Each layer of this state preparation circuit is structured as depicted in Fig. 4.

The experimental result of performing QPT with this highly entangled input state is presented in Fig. 5. The outcome clearly demonstrates that **our algorithm's performance is not constrained by the entanglement level or specific structure of the input state**, thus validating its input-agnostic nature.

### F.2 FULL MATRIX FIGURE

## G LARGE LANGUAGE MODELS USAGE STATEMENT

During the preparation of this manuscript, the authors used large language models (LLMs) for language polishing and grammar checking. The LLM assistance was limited to improving the clarity and readability of the text; all scientific content, technical derivations, and numerical results were conceived, verified, and approved solely by the authors. No LLM was used for generating figures, tables, or novel scientific ideas. The authors remain fully responsible for the final content of the paper.

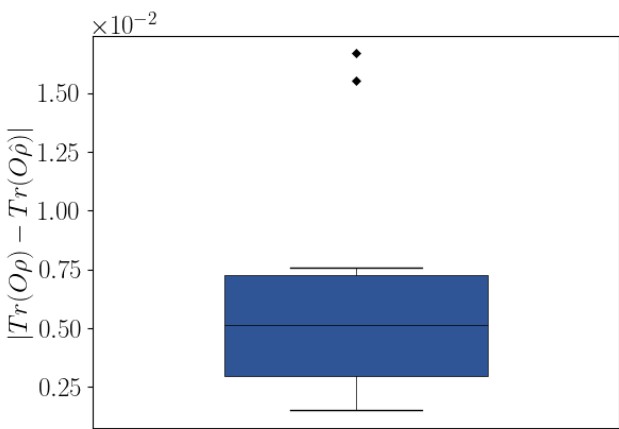

Figure 5: The experiment result of QPT where the input state is generated from the form as Fig 4. Set $l' = 2$ and the process Eq. 14 with 5 layers is at the depolarizing noise strength $0.01$.

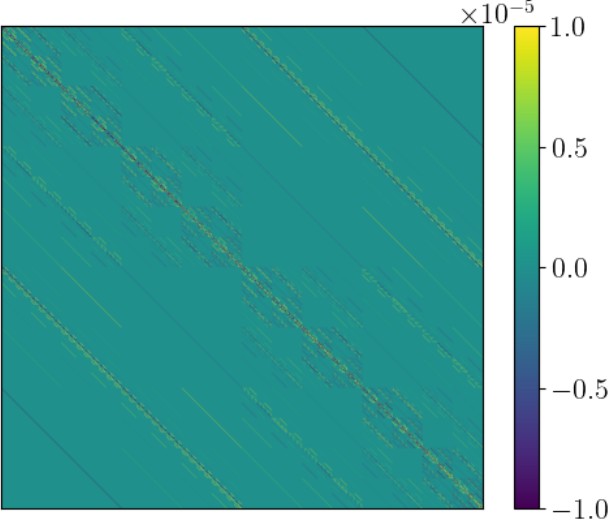

Figure 6: The heatmap visualizes the full matrix of the $\tilde{\rho} - \hat{\rho}$ in Figure 2 c, when $\theta_h = \frac{\pi}{2}$.

