# OpenReview forum: "Provably Efficient Learning Algorithms for Noisy Quantum State and Process Tomography"
_ICLR.cc/2026/Conference — Submitted to ICLR 2026_

### Official Review · Reviewer_NjPR · 2025-10-24

**Soundness:** 3
**Presentation:** 3
**Contribution:** 2
**Rating:** 4
**Confidence:** 4

**Summary:**

This paper proposes an efficient learning algorithm that would be able learn noisy quantum states and quantum channels with noise that has constant level strength by decomposing the problem to learning a polynomial number of Paulis. Compared to prior work, this algorithm has the capacity to handle both unital and non-unital noise as well as arbitrary input noise.

**Strengths:**

1. The theoretical work and proofs provided in this work is solid, and as well as simulation results that support theoretical results.
2. Apart from solving the problem, the authors provide an application for quantum error mitigation.

**Weaknesses:**

1. I have reservations about the usefulness and hardness of this protocol. The efficiency of the algorithm seems to be a result of the simplicity of the problem given its high noise setting instead of the algorithm design. I am also skeptical of the applicability of learning noisy systems as a lot of highly noisy systems end up classical simulatable anyways (e.g. arXiv:2306.05400).
2. The hardness of the ZNE protocol is the hardness of accurate prediction/interpolation under low-noise settings. The learning algorithm learns predictions of points in high-noise settings, yes, but I have trouble to see contributions to ease the hardness of ZNE itself.

**Questions:**

1. Given that the noise in the quantum system is constant level, it would stand to reason that the signal-to-noise ratio would be low.  Although the learning algorithm is efficient, how much useful information can be extracted as opposed to learning the effects of the noise channel itself? To put it another way, how much of the learnt results in noise and how much the underlying signal?
2. To clarify the case for ZNE, does the algorithm learn the cases for different $\lambda$ via different runs of the learning algorithm for each $\lambda$? Or does the algorithm have the capacity to learn different $\lambda$-s with a single run?

---

> ### Author Response · Authors · 2025-11-20
> **Addressing "Questions/Weaknesses in Official Review of 9252 by Reviewer NjPR" (part 1)**
>
> We thank the reviewer for their constructive review and comments. We have revised the paper according to the reviewer's feedback, with clarifications highlighted in blue for easy reference. Our detailed, point-by-point response to the reviewer's comments is provided below.
>
> **Comment:** *I have reservations about the usefulness and hardness of this protocol. The efficiency of the algorithm seems to be a result of the simplicity of the problem given its high noise setting instead of the algorithm design.*
>
> **Response:**  We thank the referee for this question. This is, in effect, a “which came first, the chicken or the egg?” question. It is known that state and process tomography generally require exponentially many samples in the worst case [O’Donnell et al. STOC. 2016], so these tasks are hard for any learning algorithm. Only quantum states and processes that possess special structure can be expected to be learned efficiently. In this paper we show that quantum states and processes with a constant level of noise admit a compact classical representation, which enables an efficient quantum algorithm for their tomography. ***To the best of our knowledge, this is the first efficient solution to the noisy state and process tomography problem.***
>
> Reference:
>
> O’Donnell, R.; Wright, J. Efficient Quantum Tomography. In Proceedings of the forty-eighth annual ACM symposium on Theory of Computing (STOC); ACM: Cambridge MA USA, 2016; pp 899–912.
>
> **Comment:** Given that the noise in the quantum system is constant level, it would stand to reason that the signal-to-noise ratio would be low. Although the learning algorithm is efficient, how much useful information can be extracted as opposed to learning the effects of the noise channel itself? To put it another way, how much of the learnt results in noise and how much the underlying signal?
>
> **Response:** Our learning objective is to capture the full characterization of the noisy process $\mathcal{C}$, rather than map $\mathcal{C}$ to the noise-free regime with a provable efficient guarantee. Nevertheless, to demonstrate the usefulness of the learning algorithm, our numerical results show that it can extrapolate from noisy data to the noise-free regime by leveraging a zero-noise extrapolation (ZNE)–like error-mitigation technique.
>
> Specifically, by utilizing the learned channel representation $\hat{\mathcal{C}}^\dagger_\lambda(O)$ at different noise scales $\lambda$, we successfully implement a ZNE technique (Section 5.3, Figure 3). **The numerical results confirm that the extrapolation accurately predicts the target noise-free expectation value (approaching the ideal $\gamma=0$ line) with a minimal final error of $0.0222$ using a cubic B-spline fit.** This quantitative result confirms the practical usefulness of the learned representation for downstream error mitigation tasks, demonstrating its power in recovering the intended clean circuit dynamics.
>
> **Comment:** *I am also skeptical of the applicability of learning noisy systems as a lot of highly noisy systems end up classical simulatable anyways (e.g. arXiv:2306.05400).*
>
> **Response:** We thank the referee for this question. Below we highlight the main differences between the proposed learning approach and the classical simulation methods cited.
>
> From a practical perspective, simulating a quantum circuit typically requires prior knowledge of the noise strength [Shao. et al. PRL, 2024; Schuster. et al. PRX, 2025], and some noise strengths are difficult to characterize efficiently [Chen et al.,Nat.Comm. 2023]. By contrast, our learning algorithm only assumes that the noise level is constant; its exact strength need not be known.
>
>
>
> Reference:
>
> Shao, Y.; Wei, F.; Cheng, S.; Liu, Z. Simulating Noisy Variational Quantum Algorithms: A Polynomial Approach. Phys. Rev. Lett. 2024, 133 (12), 120603.
>
> Chen, S. et al. The learnability of pauli noise. Nature Communications 14, 52 (2023)..
>
> Schuster, T.; Yin, C.; Gao, X.; Yao, N. Y. A Polynomial-Time Classical Algorithm for Noisy Quantum Circuits. Phys. Rev. X 2025, 15 (4), 041018.

---

> > ### Author Response · Authors · 2025-11-20
> > **Addressing "Questions/Weaknesses in Official Review of 9252 by Reviewer NjPR" (part 2)**
> >
> > From a theoretical perspective, both the learning algorithm and the cited simulation methods rely on Pauli-path integration, where most paths decay exponentially due to noise. As a consequence, noisy quantum processes and states are dominated by low-weight Pauli paths. ***In our learning algorithm, the quantum gates, circuit architecture, and noise strength are all unknown:*** we exploit the above property to design an efficient representation (ansatz) for tomography. Classical simulation algorithms, on the other hand, exploit the same property to compute circuit outputs, but it requires the full knowledge of involved quantum gates, circuit architecture and noise strengths.
> >
> > Finally, our work can be regarded as a 'learning-theoretic dual' of the classical-simulation results cited above.
> > This perspective is inspired by the recent paper [Gil-Fuster.et al. ICLR, 2025]. These two approaches run in parallel and reflect complementary perspectives on benchmarking noisy quantum processes — quantum tomography versus classical verification in benchmarking toolkits [Eisert. et al. Nat Rev Phys. 2020]. **The discussion above has been revised and incorporated into page 10 of the manuscript.**
> >
> > Reference:
> >
> > Gil-Fuster, E.; Gyurik, C.; Perez-Salinas, A.; Dunjko, V. On the Relation between Trainability and Dequantization of Variational Quantum Learning Models. In International conference on representation learning (ICLR); 2025; Vol. 2025, pp 24069–24093
> >
> > Eisert, J.; Hangleiter, D.; Walk, N.; Roth, I.; Markham, D.; Parekh, R.; Chabaud, U.; Kashefi, E. Quantum Certification and Benchmarking. Nat Rev Phys 2020, 2 (7), 382–390.
> >
> > **Comment:** *The hardness of the ZNE protocol is the hardness of accurate prediction/interpolation under low-noise settings. The learning algorithm learns predictions of points in high-noise settings, but does not contribute to ease the hardness of ZNE itself.\\To clarify the case for ZNE, does the algorithm learn the cases for different $\lambda$ via different runs of the learning algorithm for each $\lambda$? Or does the algorithm have the capacity to learn different $\lambda$-s with a single run?*
> >
> > **Response:** We thank the referee for this interesting question. We clarify that our learning algorithm follows the same assumptions as standard ZNE protocols: the underlying noise strength is treated as a constant-level parameter that does not decrease with the system size. As in ZNE, our approach can only increase the fault rate within quantum gates. It is generally not possible to decrease the noise rate; otherwise, one could directly access the zero-noise value without requiring error mitigation. Therefore, for a noisy quantum circuit with constant noise strength, the inability to access data from a lower-noise process is not a limitation of our algorithm, but rather an intrinsic constraint of ZNE-based error-mitigation methods [Cai et al., Rev. Mod. Phys., 2023; Kandala et al., Nature, 2019; Temme et al., PRL. 2017].
> >
> >
> >
> > Reference:
> >
> > Cai, Z.; Babbush, R.; Benjamin, S. C.; Endo, S.; Huggins, W. J.; Li, Y.; McClean, J. R.; O’Brien, T. E. Quantum Error Mitigation. Rev. Mod. Phys. 2023, 95 (4), 045005.
> >
> >
> > Kandala, A.; Temme, K.; Córcoles, A. D.; Mezzacapo, A.; Chow, J. M.; Gambetta, J. M. Error Mitigation Extends the Computational Reach of a Noisy Quantum Processor. Nature 2019, 567 (7749), 491–495.
> >
> >
> > Temme, K.; Bravyi, S.; Gambetta, J. M. Error Mitigation for Short-Depth Quantum Circuits. Phys. Rev. Lett. 2017, 119 (18), 180509.

---

> > > ### Author Response · Authors · 2025-11-20
> > > **Addressing "Questions/Weaknesses in Official Review of 9252 by Reviewer NjPR" (part 3)**
> > >
> > > On the other hand, we appreciate the referee for pointing out the hardness results for ZNE, where an exponential number of samples is required in the worst case for constant-level noise [Quek et al.,Nat. Phys., 2024]. We acknowledge that our learning approach may not circumvent this worst-case barrier. However, it offers additional benefits to ZNE-based methods. Specifically, the key advantage lies in providing an input-agnostic channel characterization method. We explain this more clearly as follows.
> > >
> > > Given a set of input states $\{\rho_x\}$, the goal is to predict the quantum expectation values $\{{\rm Tr}[\mathcal{C}(\rho_x)O]\}$ in the noise-free regime.In standard ZNE, one must perform extrapolation separately for each distinct input $\rho_x$, i.e., collect data at multiple scaling parameters $\lambda$'s for each $\langle O \rangle_{\rho_x, \lambda}$). In contrast, our learning algorithm directly learns a channel representation ($\hat{\mathcal{C}}^\dagger_\lambda(O)$) for each $\lambda$, which can be applied uniformly to all input states $\{\rho_x\}$ making it ***input-agnostic.*** This reuse significantly reduces the experimental overhead when characterizing the process across many input states. Crucially, the resulting family of channels $\{\hat{\mathcal{C}}^\dagger_\lambda(O)\}$ can then be classically extrapolated to the zero-noise regime for any input state.
> > >
> > > Reference:
> > >
> > > Quek, Y.; Stilck França, D.; Khatri, S.; Meyer, J. J.; Eisert, J. Exponentially Tighter Bounds on Limitations of Quantum Error Mitigation. Nat. Phys. 2024, 20 (10), 1648–1658.
> > >
> > > **Summary:** We thank the reviewer for these critical questionsthat have helped us clarify our contribution. . Our algorithm's efficiency stems from exploiting the compact representation enabled by constant noise—a nontrivial algorithmic contribution rather than problem trivialization. ZNE results (error $0.0222$) validate utility by successfully extracting the underlying signal. We distinguish our work from classical simulation, positioning it as a learning-theoretic dual that requires no knowledge of circuit geometry or noise parameters. Finally, our input-agnostic representation provides exponential savings in ZNE overhead by learning once per noise scale and reusing across all inputs. We hope these clarifications demonstrate both the theoretical significance and practical value of our approach, and respectfully ask the reviewer to consider them when re-evaluating the score.  Thank you for your insightful questions

---

> > > > ### Author Response · Authors · 2025-11-22
> > > >
> > > > Dear Reviewer NjPR,
> > > >
> > > > Thank you again for your constructive suggestions. We have made our best effort to address your comments and revised the manuscript accordingly. We hope our point-by-point responses provide satisfactory clarification and would be most grateful for your feedback.
> > > >
> > > > Best regards!

---

> > > > > ### Author Response · Authors · 2025-11-24
> > > > >
> > > > > Hi Reviewer NjPR, just wanted to gently follow up on our responses above. We appreciate your time and look forward to your feedback. Thanks!

---

> > > > > > ### Comment · Reviewer_NjPR · 2025-11-27
> > > > > >
> > > > > > I thank the authors for their timely feedback and response.
> > > > > >
> > > > > > I agree with the authors that tomography is a difficult problem in general, and special properties should be exploited to provide efficient learning. However, given that the protocol only addresses constant level noise, I do not feel that the paper provides substantial material for acceptance.
> > > > > >
> > > > > > For the ZNE application, I also agree with the authors that that their learning algorithm can indeed help with pinpointing noisy circuits, and that the extrapolation of the points to the zero noise setting is an intrinsic hardness of the ZNE protocol itself. However, I do not believe this protocol helps solve any substantial problem encountered in ZNE.
> > > > > >
> > > > > > Given the reasons above, I will be retaining my current score.

---

> > > > > > > ### Author Response · Authors · 2025-11-27
> > > > > > > **Reply to official comment by Reviewer NjPR**
> > > > > > >
> > > > > > > Thank you very much for your response. Although you stated you are unwilling to raise your score, we would still like to highlight the motivation behind our work.
> > > > > > >
> > > > > > > As you noted, learning quantum circuits is important both to quantum computational complexity theory and to learning theory. To date, progress on learning quantum circuits subject to 1/poly(n)-strength noise has been very limited, primarily because the task is exceptionally challenging. Efficient quantum measurements typically introduce errors of that order, which makes tomography (learning) of circuits with 1/poly(n)-strength noise approximately equivalent to the noiseless learning problem. So far, only a few classes of systems—such as constant-depth quantum circuits [H. Huang et al. 1343 - 1351, STOC 2024] or constant-time quantum dynamics [Y. Wu et al., arXiv:2503.24171, 2025]—are known to be learnable efficiently. These two methods utilize a special 'unitary-dual' identity, while it cannot be generalize from unitary to quantum channels, given the fact that quantum channel may not have its inverse. These works thus list as an open problem how to extend noise tolerance to constant-strength noise.
> > > > > > >
> > > > > > > When the noise strength is O(1/n), the study in [Y. Wu et al., ICLR 2025] used heuristic quantum algorithms to analyze the complexity of the associated quantum states; nevertheless, these schemes do not support tomography of quantum circuits when the noise strength is constant. On the other hand, for practical quantum circuits the noise amplitude is typically constant and does not decrease polynomially as the system size grows—if it did, fault-tolerant error correction would be unnecessary. This makes the assumption of constant-strength noise practically meaningful.
> > > > > > >
> > > > > > > We understand the reviewer’s concern that the actual noise level in experiments may be small (e.g., gamma\in (10^{-1}-10^{-2})), but from an asymptotic perspective such noise still does not diminish with increasing circuit size. Therefore, we maintain that our research question is of practical significance, and that our quantum algorithm advances prior works [Huang 2024 STOC, Wu 2025 ICLR, Wu 2025 arXiv: 2503.24171] in a manner consistent with ICLR acceptance criteria.

---

### Official Review · Reviewer_VPA8 · 2025-10-26

**Soundness:** 3
**Presentation:** 3
**Contribution:** 3
**Rating:** 6
**Confidence:** 2

**Summary:**

The paper proposes provably efficient algorithms for learning (i) noisy quantum states (QST) and (ii) noisy quantum processes (QPT) generated by depth-d circuits with local noise after every gate, including both unital and non-unital channels.
The key idea is that with constant local noise, learning a noisy state or process reduces to estimating an observable that is a linear combination of low-weight Pauli terms, making the problem solvable in polynomial time. The authors prove theoretical bounds on the sample and time complexity of their algorithm and also give numerical evidence on 2D transverse-field Ising circuits (with depolarizing and amplitude-damping noise) and a demonstration of zero-noise extrapolation powered by their learned models.

**Strengths:**

I support consideration of the paper for ICLR based on the following strengths:


- The paper introduces an algorithm for learning noisy quantum states with provable efficiency guarantees, directly addressing a pressing characterization task for today’s hardware.


- It leverages the structure of low-weight Pauli paths to compactly describe noisy quantum processes, resulting in a clear and elegant framework


- The authors present convincing experimental results that support their theoretical claims.

**Weaknesses:**

- As the authors note, the efficiency guarantees hold only when the per-layer noise remains at a constant level. This limits the theoretical reach as devices evolve toward lower per-gate noise and deeper circuits.

- Another minor weakness is that the analysis assumes a specific interleaved model C = E C_d E C_{d-1} … E\,C_1​. While this covers many practical cases, it does not encompass all possible noise behaviors.

**Questions:**

- Rather than estimating all Pauli coefficients <l′, could you use a sparsity-promoting, adaptive scheme (e.g., LASSO or OMP) to identify and learn only the most relevant terms?

---

> ### Author Response · Authors · 2025-11-20
> **Addressing "Questions/Weaknesses in Official Review of 9252 by Reviewer VPA8" (part 1)**
>
> We sincerely thank the reviewer for their careful review and insightful comments. In response to the reviewer's feedback, we have revised the paper, and all clarifying text is highlighted in blue for ease of inspection. Detailed responses to the reviewer's points are provided below.
>
> **Comment:** *As the authors note, the efficiency guarantees hold only when the per-layer noise remains at a constant level. This limits the theoretical reach as devices evolve toward lower per-gate noise and deeper circuits.*
>
> **Response:** We thank the referee for the opportunity to further clarify the scope of our assumptions.
>
> The assumption of a constant-level noise strength ($\gamma = \Omega(1)$) implies that the noise strength does not diminish as the system size ($n$) increases. This model is fundamentally characteristic of the current Noisy Intermediate-Scale Quantum (NISQ) era, where fixed physical imperfections (e.g., gate infidelity) dominate performance, **irrespective of the number of qubits** [Arute.et al. Nature, 2019; Kim. et al.Nature, 2023]. Therefore, our theoretical framework addresses a highly relevant and ubiquitous scenario in present-day quantum computing.
>
> Furthermore, we wish to emphasize that our theoretical guarantees are **independent of the circuit depth ($d$)**. While the required number of Pauli terms for accurate reconstruction, $l'$, depends polynomially on $\gamma$ and $\epsilon$ (i.e., $l' \propto \gamma^{-1} \log(1/\epsilon)$), it does not scale with $d$. This theoretical resilience to depth is empirically supported by our numerical results, which confirm the efficiency and scalability of our algorithm across a wide range of circuit depths, specifically testing circuits with 5, 20, and 45 layers. **Finally, for completeness, we have added an additional page to Appendix B with a comprehensive discussion of the model's topological structure and full applicability.**
>
> Reference:
>
> Arute, F.; Arya, K.; Babbush, R.; Bacon, D.; Bardin, J. C.; Barends, R.; Biswas, R.; Boixo, S.; Brandao, F. G.; Buell, D. A.; others. Quantum Supremacy Using a Programmable Superconducting Processor. Nature 2019, 574 (7779), 505–510.
>
> Kim, Y.; Eddins, A.; Anand, S.; Wei, K.; Berg, E. van den; Rosenblatt, S.; Nayfeh, H.; Wu, Y.; Zaletel, M.; Temme, K.; Kandala, A.
> Evidence for the Utility of Quantum Computing before Fault Tolerance. Nature 2023, 618, 500–505.
>
> **Comment:**  *Another minor weakness is that the analysis assumes a specific interleaved model $C = E C_d E C_{d-1} … E,C_1$. While this covers many practical cases, it does not encompass all possible noise behaviors.*
>
> **Response:** In quantum computing research, the gate-independent noise model assumes that the noise affecting quantum operations is uniform across all gates, regardless of their type or implementation. This simplification is widely adopted for several reasons:
>
> * **Theoretical Simplification:**  Assuming gate-independent noise allows researchers to develop and analyze error correction protocols and fault-tolerant methods without delving into the complexities introduced by gate-specific noise characteristics[Knill. et al. PRA, 2008; Helsen. et al. npj Quantum Information, 2019; Chen. et al. PRX Quantum, 2021]. This uniformity facilitates the derivation of general results and theoretical bounds[Nielsen. et al, 2001].
> * **Practical Approximations:** In certain quantum systems, particularly those with well-calibrated gates operating on the same number of qubits and utilizing uniform control mechanisms, noise variations across different gates can be negligible[Shor. FOCS. 1996; Arute.et al. Nature. 2019 ]. In such cases, the gate-independent noise model serves as a reasonable approximation, streamlining analysis without significantly compromising accuracy.
>
>
>
> Reference:
> Knill, E.; Leibfried, D.; Reichle, R.; Britton, J.; Blakestad, R. B.; Jost, J. D.; Langer, C.; Ozeri, R.; Seidelin, S.; Wineland, D. J. Randomized Benchmarking of Quantum Gates. Physical Review A 2008, 77 (1), 012307.
>
> Helsen, J.; Xue, X.; Vandersypen, L. M.; Wehner, S. A New Class of Efficient Randomized Benchmarking Protocols. npj Quantum Information 2019, 5 (1), 1–9.
>
> Chen, S.; Yu, W.; Zeng, P.; Flammia, S. T. Robust Shadow Estimation. PRX Quantum 2021, 2 (3), 030348. https://doi.org/10.1103/PRXQuantum.2.030348.
>
> Nielsen, M. A.; Chuang, I. L. Quantum Computation and Quantum Information; Cambridge university press Cambridge, 2001; Vol. 2.
>
> Shor, P. W. Fault-Tolerant Quantum Computation. In Proceedings of 37th conference on foundations of computer science(FOCS); IEEE, 1996; pp 56–65.
>
> Arute, F.; Arya, K.; Babbush, R.; Bacon, D.; Bardin, J. C.; Barends, R.; Biswas, R.; Boixo, S.; Brandao, F. G.; Buell, D. A.; others. Quantum Supremacy Using a Programmable Superconducting Processor. Nature 2019, 574 (7779), 505–510.

---

> ### Author Response · Authors · 2025-11-20
> **Addressing "Questions/Weaknesses in Official Review of 9252 by Reviewer VPA8" (part 2)**
>
> * **Alignment with Twirled Noise Models:**
> Techniques like Pauli twirling are employed to transform complex noise channels into diagonal forms on the Pauli basis[Wallman.et al. PRA. 2016; Chen.et al. Nat Commun. 2023]. While twirling simplifies the noise structure, it does not inherently eliminate gate dependence. However, in many scenarios, the resulting noise can be approximated as gate-independent, aligning with the assumptions of the model.
>
> The gate-independent noise model provides a foundational framework for understanding error propagation and developing correction strategies, which is a useful abstraction for theoretical exploration and the initial development of error correction methods. We leave an open question of how to depict the gate-dependent noise, which usually happens in larger, more complex quantum architectures.
>
> Reference:
>
> Wallman, J. J.; Emerson, J. Noise Tailoring for Scalable Quantum Computation via Randomized Compiling. Physical Review A 2016, 94 (5), 052325.
>
> Chen, S.; Liu, Y.; Otten, M.; Seif, A.; Fefferman, B.; Jiang, L. The Learnability of Pauli Noise. Nature Communications 2023, 14 (1), 52.
>
> **Comment:** *Rather than estimating all Pauli coefficients $\leq l'$, could you use a sparsity-promoting, adaptive scheme (e.g., LASSO or OMP) to identify and learn only the most relevant terms?*
>
> **Response:** We sincerely appreciate the reviewer's interesting suggestion.Indeed, our theoretical results already establish a bound of $l' \le \log(1/\epsilon)$, which provides a strict theoretical guarantee for the learned quantum states and processes.We hypothesize that utilizing sparsity-inducing techniques such as LASSO to further compress the number of Pauli terms could lead to highly accurate numerical results regarding the coefficient characteristics of the quantum process. However, this conjecture currently lacks a rigorous theoretical guarantee. We will definitely keep this valuable suggestion for exploration and verification in our future work. We have added a corresponding bullet point to the open problems on the page 10 and plan to solve it in subsequent research.
>
> **Summary:** We appreciate the reviewer's points regarding the scope of our assumptions. We clarified that the constant-level noise assumption is characteristic of the NISQ era and highlighted the strength of our theoretical guarantees: sample complexity is independent of the circuit depth ($d$) and empirically verified across deep circuits. We justified the interleaved noise model as a foundational theoretical framework that aligns with noise twirling techniques. Finally, we acknowledged the valuable suggestion regarding sparsity-promoting techniques (like LASSO) as a promising direction and included it in the open problems section (page 10) for future rigorous investigation. Thank you for your constructive feedback.

---

> > ### Author Response · Authors · 2025-11-22
> >
> > Dear Reviewer VPA8,
> >
> > Thank you again for your constructive suggestions. We have made our best effort to address your comments and revised the manuscript accordingly. We hope our point-by-point responses provide satisfactory clarification and would be most grateful for your feedback.
> >
> > Best regards!

---

> > > ### Author Response · Authors · 2025-11-24
> > >
> > > Hi Reviewer VPA8, just wanted to gently follow up on our responses above. We appreciate your time and look forward to your feedback. Thanks!

---

> > > > ### Comment · Reviewer_VPA8 · 2025-11-27
> > > >
> > > > I thank the authors for their reply, which indeed clarifies many aspects of their work. I now understand that the circuit and noise model studied is well-motivated and widely used in the NISQ regime, and I find the paper’s theoretical guarantees in this setting non-trivial and valuable, so I support acceptance at ICLR. However, I do not feel justified in further raising my score, since as hardware improves it becomes increasingly important to extend the analysis beyond constant noise (for instance to \gamma=1/poly(n), as noted by the authors in the discussion section) in order to substantially broaden the theory’s impact.

---

> > > > > ### Author Response · Authors · 2025-11-27
> > > > > **Reply to Reviewer VPA8**
> > > > >
> > > > > We are very grateful that you believe our paper should be accepted to ICLR. Thank you again for your valuable comments on our manuscript — they have been extremely helpful in improving the significance and practicality of our work. The study of circuit and state tomography under 1/poly(n) noise strength remains an important open direction for future work.

---

### Official Review · Reviewer_khkR · 2025-10-30

**Soundness:** 3
**Presentation:** 2
**Contribution:** 2
**Rating:** 4
**Confidence:** 3

**Summary:**

The paper presents a structure-free learning approach for a class of noisy quantum states and processes. The core mechanism for the efficiency of the algorithm is that depolarizing noise suppresses (exponentially) contributions with high Pauli weight. Through that, the authors show, they can well approximate states and processes by only learning contributions with low Pauli weight. The authors show the performance of their algorithm for an Ising model and discuss applications to quantum error mitigation.

**Strengths:**

The paper presents mathematical rigorous performance bounds for the learning algorithm, targeted for realistic experimental setups with noisy quantum computers. A key strength is the unified treatment of both unital and non-unital noise as well as of states and processes. Furthermore, the explicit treatment of realistic noise models compared to learning general CPTP maps is interesting.

**Weaknesses:**

The authors make strong claims about the significance of their contributions which do not seem justified based on numerous related literature and results. (e.g. "This work establishes a theoretical foundation for practical quantum-process learning...", "Efficiently characterizing noisy quantum states and processes has stood as one of the most significant problems over the past decades. In this paper, we resolve this problem..." ).

The limitations of this work are not sufficiently described and the comparison with related work (in particular the cited Huang et al 2023a and results on simulating noisy circuits) falls too short. (Details below)

**Questions:**

- How general is is the noisy quantum process given by eq 4?
- In the proofs, it says that channel $C_i$ is a layer of two-qubit random gates. How big of a restriction is this assumption? Do the results therefore only hold on average for circuits with that property? What is the behaviour in the worst-case (assuming $C_i$ is unitary but highly structured)?
-  Relation to Huang et al 2023a: that this algorithm is valid for an arbitrary CTPT map without any specified structure is not sufficiently described. Furthermore, the authors (of Huang et al 2023a) state that their work also holds if the distribution over $\rho$ is only polynomially far away from locally flat. Their work can predict many observables for the same training data, whereas the presented work here only predicts a single observable? It would be insightful to see numerical results how the the presented algorithm relates to the Huang work and potentially outperforms it in a regime beyond Huangs assumptions.
- Novelty of the analysis and proofs: The exponentially-low contribution of Pauli terms with high weight is used extensively in previous work (e.g. Huang  et al 2023,  Simulating quantum circuits with arbitrary local noise using Pauli Propagation ( https://arxiv.org/pdf/2501.13101  ) with all of the listed references in related work,  A polynomial-time classical algorithm for noisy random circuit sampling. (https://arxiv.org/pdf/2211.03999) The presented theorems and lemmas do not seem to sufficiently acknowledge this. Is it a fair assessment that those results on efficient simulation of noisy processes are re-interpreted and carried over to the task of learning noisy quantum processes?

Minor edits:
- In lemma 1, it says $C_i= C_i^{\dagger}C_i$
- please include eq 13 in table 2 in Appendix A for explicit comparison

---

> ### Author Response · Authors · 2025-11-20
> **Addressing "Questions/Weaknesses in Official Review of 9252 by Reviewer khkR" (part 1)**
>
> We thank the reviewer for their careful review and constructive comments on our manuscript. We have revised the paper according to the reviewer's feedback, and the clarifying text is highlighted in blue for easy inspection. We provide a detailed response addressing each of the reviewer's points below.
>
> **Comment:** *How general is is the noisy quantum process given by eq 4?*
>
> **Response:** We thank the referee for the opportunity to clarify the generality and physical relevance of the noisy quantum process defined by Eq. 4. The model is highly general and covers a broad, practically relevant class of quantum circuits encountered in the NISQ era.
>
> The process, $$\mathcal{C}=\mathcal{C}_1\circ \mathcal{E}^{\otimes n}\circ\cdots\circ \mathcal{E}^{\otimes n} \circ\mathcal{C}_d$$ (Eq. 4), encompasses a large variety of circuit architectures:
>
> **Algorithmic Scope:** Each layer $\mathcal{C}_i$ can represent standard variational quantum algorithms (VQAs) such as QAOA, VQE, and general Quantum Neural Networks (QNNs) [Zhou.et al. PRX. 2020; Kandala. et al.Nature, 2017; Grimsley. et al.Nat Commun, 2019].
>
> **Weak Structural Assumption:**  Our reliance on the local $2$-design property is an extremely weak and general assumption. This condition is satisfied by common structures like Clifford gates [Zhu. et al. arxiv; 2016] and is characteristic of algorithms utilizing 'randomly initialized parameters' and 'classical optimizations' [McClean.et al. Nat Commun, 2018; Cerezo. et al. Nat Commun, 2021].
>
>
>
> Reference:
>
> Zhou, L.; Wang, S.-T.; Choi, S.; Pichler, H.; Lukin, M. D. Quantum Approximate Optimization Algorithm: Performance, Mechanism, and Implementation on near-Term Devices. Phys. Rev. X 2020, 10 (2), 021067.
>
> Kandala, A.; Mezzacapo, A.; Temme, K.; Takita, M.; Brink, M.; Chow, J. M.; Gambetta, J. M. Hardware-Efficient Variational Quantum Eigensolver for Small Molecules and Quantum Magnets. Nature 2017, 549 (7671), 242–246.
>
> Grimsley, H. R.; Economou, S. E.; Barnes, E.; Mayhall, N. J. An Adaptive Variational Algorithm for Exact Molecular Simulations on a Quantum Computer. Nat Commun 2019, 10 (1), 3007.
>
> McClean, J. R.; Boixo, S.; Smelyanskiy, V. N.; Babbush, R.; Neven, H. Barren Plateaus in Quantum Neural Network Training Landscapes. Nature Communications 2018, 9 (1), 4812.
>
> Cerezo, M.; Sone, A.; Volkoff, T.; Cincio, L.; Coles, P. J. Cost Function Dependent Barren Plateaus in Shallow Parametrized Quantum Circuits. Nature Communications 2021, 12 (1), 1–12.

---

> > ### Author Response · Authors · 2025-11-20
> > **Addressing "Questions/Weaknesses in Official Review of 9252 by Reviewer khkR" (part 2)**
> >
> > In quantum computing research, the gate-independent noise model assumes that the noise affecting quantum operations is uniform across all gates, regardless of their type or implementation. This simplification is widely adopted for several reasons:
> >
> > **Foundational Modeling:** Assuming gate-independent noise simplifies the development and analysis of error correction protocols [Knill. et al. PRA, 2008; Shor. FOCS. 1996]. This uniformity facilitates the derivation of general theoretical results and bounds on error propagation [Nielsen. et al, 2001; Arute.et al. Nature. 2019].
> >
> > **Noise Twirling Alignment:** Techniques like Pauli twirling are employed to transform complex, gate-dependent physical noise channels into simpler, diagonal forms on the Pauli basis [Wallman.et al. PRA. 2016; Chen.et al. Nat Commun. 2023]. In many experimental scenarios, the resulting effective noise can be well-approximated as gate-independent, aligning with our model's assumptions.
> >
> > While our rigorous proofs rely on the statistical properties of the local random circuit ensemble, the practical applicability of the model extends further. Our numerical results successfully demonstrate the efficiency of our learning algorithm on a noisy Hamiltonian dynamics simulation (transverse-field Ising model), a scenario where the underlying circuit does not possess the locally random property. This empirical evidence suggests our algorithm can be applied to a broader range of noisy quantum processes beyond the strictly random model. We acknowledge that depicting true, gate-dependent noise in larger architectures remains an open research question. For a complete and rigorous understanding of the assumed model’s topological structure and its full range of applicability, **we have added a section introducing the circuit model we consider (see Appendix B).**
> >
> > Reference:
> >
> > Zhu, H.; Kueng, R.; Grassl, M.; Gross, D. The Clifford Group Fails Gracefully to Be a Unitary 4-Design. arXiv September 26, 2016.
> >
> > Mele, A. A. Introduction to Haar Measure Tools in Quantum Information: A Beginner’s Tutorial. Quantum 8, 1340 (2024).
> >
> > Knill, E.; Leibfried, D.; Reichle, R.; Britton, J.; Blakestad, R. B.; Jost, J. D.; Langer, C.; Ozeri, R.; Seidelin, S.; Wineland, D. J. Randomized Benchmarking of Quantum Gates. Physical Review A 2008, 77 (1), 012307.
> >
> > Helsen, J.; Xue, X.; Vandersypen, L. M.; Wehner, S. A New Class of Efficient Randomized Benchmarking Protocols. npj Quantum Information 2019, 5 (1), 1–9.
> >
> > Chen, S.; Yu, W.; Zeng, P.; Flammia, S. T. Robust Shadow Estimation. PRX Quantum 2021, 2 (3), 030348.
> >
> > Nielsen, M. A.; Chuang, I. L. Quantum Computation and Quantum Information; Cambridge university press Cambridge, 2001; Vol. 2.
> >
> > Shor, P. W. Fault-Tolerant Quantum Computation. In Proceedings of 37th conference on foundations of computer science(FOCS); IEEE, 1996; pp 56–65.
> >
> > Arute, F.; Arya, K.; Babbush, R.; Bacon, D.; Bardin, J. C.; Barends, R.; Biswas, R.; Boixo, S.; Brandao, F. G.; Buell, D. A.; others. Quantum Supremacy Using a Programmable Superconducting Processor. Nature 2019, 574 (7779), 505–510.
> >
> > Wallman, J. J.; Emerson, J. Noise Tailoring for Scalable Quantum Computation via Randomized Compiling. Physical Review A 2016, 94 (5), 052325.
> >
> > Chen, S.; Liu, Y.; Otten, M.; Seif, A.; Fefferman, B.; Jiang, L. The Learnability of Pauli Noise. Nature Communications 2023, 14 (1), 52.

---

> ### Author Response · Authors · 2025-11-20
> **Addressing "Questions/Weaknesses in Official Review of 9252 by Reviewer khkR" (part 3)**
>
> **Comment:** *In the proofs, it says that channel $\mathcal{C}_i$
> is a layer of two-qubit random gates. How big of a restriction is this assumption? Do the results therefore only hold on average for circuits with that property? What is the behaviour in the worst-case (assuming $\mathcal{C}_i$
> is unitary but highly structured)?*
>
> **Reponse:** Here we emphasize that the local 2-design assumption of the gates in $\mathcal{C}_i$ is an extremely weak condition; quantum neural network models are typical cases [McClean.et al. Nat Commun, 2018; Cerezo. et al. Nat Commun, 2021], and even Clifford gates satisfy such an assumption [Zhu.et al. arxiv, 2016]. We note that if an ensemble follows a $(t+1)$-design, it must follow the $t$-design property [Mele. et al.Quantum, 2024]. As a result, this assumption is very general and covers a wide range of NISQ algorithms related to 'randomly initial parameters' and 'classical optimizations' [McClean.et al. Nat Commun, 2018; Cerezo. et al. Nat Commun, 2021].
>
> **To clarify our work, we have added a section on pages 17-18 introducing the circuit model we consider.** Our method does not require knowledge of the circuit architecture, and the results hold for a given channe $\mathcal{C}$ with high success probability ($\geq 1-\delta$). We do not rule out the possibility that our learning algorithm fails for some worst-case quantum channels; however, the failure probability may be polynomially small and tends to zero as the system size increases.
>
>
> Reference:
>
> McClean, J. R.; Boixo, S.; Smelyanskiy, V. N.; Babbush, R.; Neven, H. Barren Plateaus in Quantum Neural Network Training Landscapes. Nature Communications 2018, 9 (1), 4812.
>
> Cerezo, M.; Sone, A.; Volkoff, T.; Cincio, L.; Coles, P. J. Cost Function Dependent Barren Plateaus in Shallow Parametrized Quantum Circuits. Nature Communications 2021, 12 (1), 1–12.
>
> Zhu, H.; Kueng, R.; Grassl, M.; Gross, D. The Clifford Group Fails Gracefully to Be a Unitary 4-Design. arXiv September 26, 2016.
>
> Mele, A. A. Introduction to Haar Measure Tools in Quantum Information: A Beginners Tutorial. Quantum 2024, 8, 1340.
>
> **Comment:** *Relation to Huang et al 2023a: that this algorithm is valid for an arbitrary CTPT map without any specified structure is not sufficiently described. Furthermore, the authors (of Huang et al 2023a) state that their work also holds if the distribution over $\rho$ is only polynomially far away from locally flat. Their work can predict many observables for the same training data, whereas the presented work here only predicts a single observable? It would be insightful to see numerical results how the the presented algorithm relates to the Huang work and potentially outperforms it in a regime beyond Huangs assumptions.*
>
> **Reponse:** From a theoretical perspective, the fundamental difference lies in the source of the sparse representation of the learning target $\mathcal{C}^{\dagger}(O)$. In our work, the sparsity originates from the noise signal via Pauli-path integration, which holds for all input states. By contrast, the efficiency in Huang et al. (2023a) is guaranteed by a constraint on the input distribution: the distribution over input states $\rho$ must be at most polynomially far from a locally flat distribution. Crucially, this distinction affects practical applicability: our QPT framework is designed for noisy quantum processes where no restriction is placed on the input-state distribution ($\rho$). Consequently, our algorithm is inherently input-agnostic across the entire Hilbert space, thereby overcoming the input-distribution constraint required by related simulation-based learning methods. **An additional numerical experiment is reported in Appendix E.1 (page 28-29) : even for a highly entangled input state, our algorithm still performs well.**
>
>
> Regarding the observable $O$, we only assume the observable can be written as a sum of few-body observables, where each qubit is acted on by a constant number of the few-body observables, which is same to the observable used in Huang etal 2023.
>
> **Comment:** *Novelty of the analysis and proofs: The exponentially-low contribution of Pauli terms with high weight is used extensively in previous work (e.g. Huang et al 2023, Simulating quantum circuits with arbitrary local noise using Pauli Propagation ( https://arxiv.org/pdf/2501.13101 ) with all of the listed references in related work, A polynomial-time classical algorithm for noisy random circuit sampling. (https://arxiv.org/pdf/2211.03999) The presented theorems and lemmas do not seem to sufficiently acknowledge this. Is it a fair assessment that those results on efficient simulation of noisy processes are re-interpreted and carried over to the task of learning noisy quantum processes?*
>
> **Reponse:** We thank the referee to point out this question. Below we highlight the main differences between the proposed learning approach and the classical simulation methods cited.

---

> ### Author Response · Authors · 2025-11-20
> **Addressing "Questions/Weaknesses in Official Review of 9252 by Reviewer khkR" (part 3)**
>
> From a practical perspective, simulating a quantum circuit typically requires prior knowledge of the noise strength [Shao. et al. PRL, 2024; Schuster. et al. PRX, 2025], and some noise strengths are difficult to characterize efficiently [Chen et al.,Nat.Comm. 2023]. By contrast, our learning algorithm only assumes that the noise level is constant; its exact strength need not be known.
>
> From a theoretical perspective, both the learning algorithm and the cited simulation methods rely on Pauli-path integration, where most paths decay exponentially due to noise. As a consequence, noisy quantum processes and states are dominated by low-weight Pauli paths.   ***In our learning algorithm, the quantum gates, circuit architecture, and noise strength are all unknown*** ; we exploit the above property to design an efficient representation (ansatz) for tomography. Classical simulation algorithms, on the other hand, exploit the same property to compute circuit outputs, but it requires the full knowledge of involved quantum gates, circuit architecture and noise strengths.
>
> Finally, our work can be regarded as a 'learning-theoretic dual' of the classical-simulation results cited above.
> This perspective is inspired by the recent paper( Gil-Fuster.et al. ICLR, 2025). These two approaches run in parallel and reflect complementary perspectives on benchmarking noisy quantum processes — quantum tomography versus classical verification in benchmarking toolkits [Eisert. et al. Nat Rev Phys. 2020].
> **The discussion above has been revised and incorporated into page 10 of the manuscript.**
>
> Reference:
>
> Shao, Y.; Wei, F.; Cheng, S.; Liu, Z. Simulating Noisy Variational Quantum Algorithms: A Polynomial Approach. Phys. Rev. Lett. 2024, 133 (12), 120603.
>
>
> Chen, S. et al. The learnability of pauli noise. Nature Communications 14, 52 (2023).
>
> Gil-Fuster, E.; Gyurik, C.; Perez-Salinas, A.; Dunjko, V. On the Relation between Trainability and Dequantization of Variational Quantum Learning Models. In International conference on representation learning (ICLR); 2025; Vol. 2025, pp 24069–24093.
>
> Schuster, T.; Yin, C.; Gao, X.; Yao, N. Y. A Polynomial-Time Classical Algorithm for Noisy Quantum Circuits. Phys. Rev. X 2025, 15 (4), 041018.
>
> Eisert, J.; Hangleiter, D.; Walk, N.; Roth, I.; Markham, D.; Parekh, R.; Chabaud, U.; Kashefi, E. Quantum Certification and Benchmarking. Nat Rev Phys 2020, 2 (7), 382–390.
>
> **Comment:** *In lemma 1, it says $C_i = C_i^{\dagger}C_i$*
>
> **Response:** We appreciate the careful check. There appears to be a minor confusion between the unitary gate $C_i$ and the channel symbol $\mathcal{C}_i$. Our text correctly states that the channel is defined as $\mathcal{C}_i(\cdot) = C_i^{\dagger}(\cdot)C_i$, where the underlying $C_i$ is the unitary gate itself (satisfying $C_i C_i^{\dagger} = I$). We confirm the original expression defining the unitary channel is correct.
>
> **Comment:** *Please include eq 13 in table 2 in Appendix A for explicit comparison*
>
> **Response:** Thanks for this comment. We have updated Table 2 in Appendix A to include the relevant information from Equation (13) for a more complete comparison.
>
> **Summary:** We believe these comprehensive clarifications accurately delineate the scope and novelty of our learning framework in the context of existing simulation literature, establishing its distinct contribution to noisy quantum device benchmarking. We confirmed the model's generality **(Appendix B)** and clarified that our method's efficiency is input-agnostic, overcoming the input constraints found in related works **( Appendix E.1 (page 28-29))** . We respectfully request the reviewer to consider these thorough revisions and the resulting clarity when re-evaluating and raising the score. Thank you for your time and expertise.

---

> > ### Author Response · Authors · 2025-11-22
> >
> > Dear Reviewer khkR,
> >
> > Thank you again for your constructive suggestions. We have made our best effort to address your comments and revised the manuscript accordingly. We hope our point-by-point responses provide satisfactory clarification and would be most grateful for your feedback.
> >
> > Best regards!

---

> > > ### Author Response · Authors · 2025-11-24
> > >
> > > Hi Reviewer khkR, just wanted to gently follow up on our responses above. We appreciate your time and look forward to your feedback. Thanks!

---

### Official Review · Reviewer_FAFT · 2025-10-31

**Soundness:** 2
**Presentation:** 1
**Contribution:** 3
**Rating:** 2
**Confidence:** 4

**Summary:**

This paper introduces a tomography algorithm that learns a quantum state prepared by a certain randomly sampled circuit interspersed with local, possible non-unital noise. The main technique leveraged in Pauli Path Truncation, which shows that for a noise strength of O(1), the quantum state generated by such a process is supported on a smaller set of Pauli operator strings. This observation reduces the size of the learning problem considerably and allows for efficient tomography using standard techniques.

**Strengths:**

The core idea of the paper is interesting and pertinent. The authors have used recent results on the decomposition of non-unital channels to make their algorithm applicable to both unital and non-unital channels. The inclusion on non-trivial numerical experiments are a big positive.

**Weaknesses:**

The main issue with the paper is how overstated the results are in the title, introduction and abstract.
The algorithm and the proofs only work for noise at the single qubit level and also require that the circuit be drawn from the specific 2-qubit random family. Neither of these important caveats are obvious from the the title and abstract, which severely oversells the results. I strongly urge the authors to change the title, abstract and introduction to accurately reflect the scope of their results. This will greatly improve the scientific quality of the paper and make it suitable for an interdisciplinary venue like ICLR.

For instance, the abstract says that the method works for "...any noisy process or state from measurement data under both unital
and non-unital local noise..". The method in fact only works for a process defined by (4), the noise is not just local but single qubit and the circuits have to sampled at random from a 2-qubit ensemble. The results here will not work even if the noise locally correlates nearest neighbor qubits.

Even in other parts of the paper this distinction is not clear: For instance in Problem 1, the authors state the error in learning a quantum state prepared by the circuit $\mathcal{C}$ as defined in (4). But (4) is a randomly sampled, noisy circuit. So is the error in Problem 1 defined for all possible circuits that can be sample from (4) or is it an average statement?

The noise model is referred to as ``local'' in many places. It should be called ``single-qubit".

Similarly, later Lemmas and theorems have to be stated more precisely to make sure that the readers understand that the the circuit $\mathcal{C}$ is chosen randomly from a family and if the results in these statements are expected results or high probablity results w.r.t to this randomness.

**Questions:**

0. Do the tomography algorithms explained in the paper require prior knowledge of the noise strength $\gamma$? If we do not know $\gamma$ how do we fix largest weight of Pauli strings to include in the tomography procedure?

1. In theorem 1, could the authors explain why the post-processing complexity does not seem to depend on n? Naively, one would think that there should at least be a linear dependence on $n$ as the algorithm has to parse measurement outcomes of length $n$ for an $n$ qubit system?

2. How the decrease of error with the system size the experiment in Fig 2 consistent with the findings of Theorem 1? Since there is no $n$ dependence in the sample complexity, shouldn't the error stay flat with the system size increase?

3. In going from Eq.(27)  to subsequent expressions, how  did the authors
   (a) go from a double sum in (27) to a single sum in the next expression? What happened to the cross terms?
   (b) take the square root into the sum while maintaining equality?

4. For Problem 1 and Theorems 1–2, are guarantees high‑probability over the random circuit distribution in Eq. (4), or uniform over all circuits consistent with Eq. (4)? Please state the probability space clearly where “with nearly unit success probability” in Lemma 2 and Lemma 3 is taken

5. Theorem 2 shows that complexity depends on $|| O ||_F$. What is the value of this quantity for an interesting $k$-local operator? My concern is that even a simple Pauli operators for an $n$ qubit system can have a Frobenius  norm that scales as $exp(n)$. These can introduce undesirable $n-$dependence back into sample complexity.

6. The benchmarks use Trotterized Ising circuits (fixed, structured gates), not random 2‑qubit 2‑design layers. Can you comment on why the method still works there, and which parts of the theory you expect to survive without per‑layer randomness?
7. In Algorithm 1, how is $l'$ chosen without depending on $\gamma$? My confusion comes form the fact that truncation in Pauli paths in Lemma 1, show that we have to keep Pauli matrices with weight $O(1/\gamma)$, but in Algorithm 1, this $\gamma$ dependence seems to be not there.

---

> ### Author Response · Authors · 2025-11-20
> **Addressing "Questions/Weaknesses in Official Review of 9252 by Reviewer  FAFT" (part 1)**
>
> We thank the reviewer for the thorough review and constructive feedback. We have revised the manuscript according to the reviewer's comments, with clarifying text highlighted in blue for convenient inspection. Below, we address the reviewer's comments in detail.
>
> **Comment:** *The title and abstract severely oversell the results by not clearly stating the assumptions of "i.i.d. single-qubit noise" and "random 2-design circuits." The noise model should be called "single-qubit."Similarly, later Lemmas and theorems have to be stated more precisely to make sure that the readers understand that the the circuit $\mathcal{C}$ is chosen randomly from a family and if the results in these statements are expected results or high probablity results w.r.t to this randomness.*
>
> **Response:** We thank the reviewer's suggestion. We initially used the term “Local Noise” following previous arts (Schuster T., et al. PRX, 2025, Aharonov D., et al., 2023), where the noise is uniformly acted on each qubit between quantum layers. We have revised  “Local Noise” to a more precise term \textbf{“i.i.d. single-qubit noise”} throughout the manuscript.
>
> For all our theorems and lemmas, we emphasize that  $\mathcal{C}_i$ is a layer of two-qubit Haar random quantum gates. and the algorithm succeeds with a probability arbitrarily close to unity.
>
> Reference:
>
> Schuster, T., Yin, C., Gao, X.et al. A polynomial-time classical algorithm for noisy quantum circuits. Phys. Rev. X 15, 041018 (2025).
>
> Aharonov D., Gao X., Landau Z., et al., A Polynomial-Time Classical Algorithm for Noisy Random Circuit Sampling. in Proceedings of the 55th Annual ACM Symposium on Theory of Computing(STOC) 945–957 (ACM, Orlando FL USA, 2023).
>
> **Comment:** *Do the tomography algorithms require prior knowledge of the noise strength $\gamma$? If we do not know $\gamma$ how do we fix the largest weight of Pauli strings $l'$?*
>
> **Response:** The learning algorithm does not require the exact value of $\gamma$ during training; it only requires the assumption that $\gamma$ is a constant-level parameter independent of the system size. The referee may be concerned about the choice of the truncation-length threshold $l'$ which is fundamentally determined by the target accuracy $\epsilon$ and strength $\gamma$. However, similar to related works on shallow-circuit learning (Huang et al., STOC 2024) and constant-time Hamiltonian learning (Haah et al., Nat. Phys. 2024), the sample-complexity upper bounds also depend on other problem parameters (e.g., circuit depth $d$ n Huang et al. and time $t$ in in Haah et al.). This dependence does not affect the implementation of our algorithm provided these parameters remain constant. Therefore, the scaling of the truncation length $l'$ is asymptotically determined by the target accuracy $\epsilon$, which is the dominant factor.
>
> We emphasize that, although our algorithm does not require knowledge of $\gamma$, its value can be estimated experimentally using standard benchmarking protocols: the single-qubit noise strength $\gamma$can be estimated via interleaved randomized benchmarking (RB) (Magesan et al.PRL, 2012), and the overall circuit error rate (which is related to $\gamma$) can be monitored through standard RB (Magesan et al.PRL, 2011).
>
> Reference:
>
> Huang, H.-Y.; Liu, Y.; Broughton, M.; Kim, I.; Anshu, A.; Landau, Z.; McClean, J. R. Learning Shallow Quantum Circuits. In Proceedings of the 56th Annual ACM Symposium on Theory of Computing(STOC); 2024; pp 1343–1351.
>
> Haah, J.; Kothari, R.; Tang, E. Learning Quantum Hamiltonians from High-Temperature Gibbs States and Real-Time Evolutions. Nat. Phys. 2024, 20 (6), 1027–1031.
>
> Magesan, E.; Gambetta, J. M.; Johnson, B. R.; Ryan, C. A.; Chow, J. M.; Merkel, S. T.; Da Silva, M. P.; Keefe, G. A.; Rothwell, M. B.; Ohki, T. A.; Ketchen, M. B.; Steffen, M. Efficient Measurement of Quantum Gate Error by Interleaved Randomized Benchmarking. Phys. Rev. Lett. 2012, 109 (8), 080505.
>
>
> Magesan, E.; Gambetta, J. M.; Emerson, J. Scalable and Robust Randomized Benchmarking of Quantum Processes. Phys. Rev. Lett. 2011, 106 (18), 180504.

---

> ### Author Response · Authors · 2025-11-20
> **Addressing "Questions/Weaknesses in Official Review of 9252 by Reviewer FAFT" (part 2)**
>
> **Comment:** *How the decrease of error with the system size the experiment in Fig 2 consistent with the findings of Theorem 1? Since there is no dependence in the sample complexity, shouldn't the error stay flat with the system size increase?*
>
> **Response:** The observed decrease in error $\epsilon$ with increasing system size $n$ (Figures 2a and 2c) is entirely consistent with the asymptotic nature of Theorem 1-2. The theorems guarantee that the sample complexity required to achieve a fixed accuracy $\epsilon$ is $n$-independent only once the asymptotic regime is reached; it does not mandate that the error must be strictly flat for small systems.
>
> The experimental trend visually validates this convergence process. For smaller system sizes, the decrease in error reflects that the average-case behavior, where noise effectively confines the dynamics to a low-weight Pauli subspace, quickly becomes dominant. Crucially, the graph shows that the error rapidly transitions from a decreasing trend to a flat plateau once $n$ is sufficiently large. This plateau confirms the central theoretical finding: the error magnitude ($\epsilon$) for a fixed sample size stabilizes, demonstrating that the learning performance has become $n$-independent as the predicted asymptotic regime is reached.
>
> **Comment:** In going from Eq.(27) to subsequent expressions, how did the authors (a) go from a double sum to a single sum? (b) take the square root into the sum while maintaining equality?
>
> **Response:** We appreciate the reviewer's pointing out these critical technical oversights regarding the norm conversion and sum simplification. We acknowledge that the original derivation was not mathematically rigorous. The error has been fully corrected, and **the revised mathematical analysis is detailed in Appendix C.2.**
>
> **Comment:** *For Problem 1 and Theorems 1–2, are guarantees high-probability over the random circuit distribution in Eq. (4), or uniform over all circuits consistent with Eq.(4)? Please state the probability space clearly where "with nearly unit success probability" in Lemma 2 and Lemma 3 is taken*
>
> **Response:** Here, we formally state the graph-theoretic definitions of circuit
> \begin{equation}
>   \mathcal C = \mathcal C_{1}\circ\mathcal E^{\otimes n}\circ\mathcal C_{2}\circ\mathcal E^{\otimes n}\circ\cdots\circ\mathcal E^{\otimes n}\circ\mathcal C_{d}
> \end{equation}
>  studied in the manuscript.
>
> *Definition 1 [Architecture, restatement of (Haferkamp. Nat. Phys. 2020)]
> An architecture is a directed acyclic graph that contains $R\in Z>0$ vertices (gates). Two edges (qubits) enter each vertex, and two edges exit. Two typical examples are listed below:*
>
>  * *A brickwork is the architecture of any circuit formed as follows. Apply a string of two-qubit gates: $U_{1, 2}\otimes U_{3, 4}\otimes \cdots \otimes U_{n-1,n}. $Then apply a staggered string of gates. Perform this pair of steps $T$ times in total, using possibly different gates each time.*
>  * *A staircase is the architecture of any circuit which apples a stepwise string of two-qubit gates: $U_{n,n-1}U_{n-2,n-1}\cdots U_{2, 1}$. Repeat this process $T$ times, using possibly different gates each time.*
> Here, the quantum circuit layer $\mathcal{C}_i$ may adopt any architecture, and \emph{\textbf{we note that our learning algorithm can be applied to any geometrical architecture,}} and thus covers a large class of noisy quantum circuits, especially for those used in NISQ algorithms
>
> *Definition 2 [Random Quantum Circuit,  restatement of (Haferkamp. Nat. Phys. 2020)]
> Let $G$ denote an arbitrary architecture. A probability distribution can be induced over the architecture-$G$ circuits as follows: for each vertex in $G$, draw a gate Haar-randomly from $SU(4)$. Then contract the unitaries along the edges of $G$. Each circuit so constructed is called a random quantum circuit.*
>
> Based on the Def.1 and 2, the definition of the random noisy quantum is provided below.
>
> *Definition 3 [Random noisy quantum circuit]
> Let $\widetilde G$ denote an arbitrary architecture.A probability distribution can be induced over the architecture-$\widetilde G$ circuits as follows: for each vertex in $\widetilde G$, draw a gate Haar-randomly from $SU(4)$ and an i.i.d single-qubit noisy channel. Then contract the unitaries along the edges of $\widetilde G$. Each circuit so constructed is called a random noisy quantum circuit.*
>
> **We have already attached the definitions above in Appendix B.** The guarantee is a **high-probability bound ($\geq 1-\delta$）** over random circuit ensemble defined in Def.3.
>
> reference:
>
> Haferkamp, J., Faist, P., Kothakonda, N. B. T., Eisert, J. and  Yunger Halpern, N. Linear growth of quantum circuit complexity. Nat. Phys. 18, 528–532 (2022).

---

> ### Author Response · Authors · 2025-11-20
> **Addressing "Questions/Weaknesses in Official Review of 9252 by Reviewer FAFT" (part 3)**
>
> **Comment:** *Theorem 2 shows that complexity depends on $||O||_F$. What is the value of this quantity for an interesting $k$-local operator? My concern is that even a simple Pauli operators for an $n$ qubit system can have a Frobenius norm that scales as $exp(n)$.*
>
> **Response:** There is no exponential dependency here. The concern regarding exponential scaling is resolved by relying on the definition of $O$ as a sparsely supported operator in the Pauli basis, that is
> \begin{align}
>    O=\sum _ {Q\in\{I,X,Y,Z\}^{\otimes n}, |Q|=\mathcal{O}(1)}{\rm Tr}[OQ]Q/2^n,
> \end{align}
> which is widely used in NISQ algorithms, such as QML, VQE and QAOA [McClean.et al.New J. Phys. 2016; Cerezo. et al.Nat Commun, 2021]. The normalized Frobenius norm $\||O\|| _ F$ is the sum of the squared coefficients of the orthogonal Pauli terms in the decomposition, which is $\||O\|| _ F=\sqrt{\sum_{Q\in\{I,X,Y,Z\}^{\otimes n}, |Q|=\mathcal{O}(1)}{\rm Tr}^2[OQ]/2^n}$. Since the number of terms in the decomposition is polynomial in $n$, and the norm of each individual Pauli string is constant, the resulting complexity dependence of $\||O\|| _ F$ is at most polynomial in $n$, and does not scale exponentially. This ensures our overall complexity remains feasible.
>
> Reference:
>
> McClean, J. R.; Romero, J.; Babbush, R.; Aspuru-Guzik, A. The Theory of Variational Hybrid Quantum-Classical Algorithms. New J. Phys. 2016, 18 (2), 023023.
>
> Cerezo M., Sone, A., Volkoff, T. et al. Cost function dependent barren plateaus in shallow parametrized quantum circuits. Nat Commun 12, 1791 (2021).
>
> **Comment:** *The benchmarks use Trotterized Ising circuits (fixed, structured gates), not random 2-qubit 2-design layers. Can you comment on why the method still works there, and which parts of the theory you expect to survive without per-layer randomness?*
>
> **Response:** We thank the reviewer for this question. In the manuscript we rigorously show that random quantum processes subject to i.i.d. single-qubit noise can be learned efficiently by our algorithm. Although the theory does not cover every worst-case scenario, our numerical results confirm that the algorithm remains effective and accurate when applied to noisy Hamiltonian dynamics.
>
> In our simulations we decompose the Ising-model dynamics into single- and two-qubit gates with fixed rotation angles. This setup can be viewed as an instance of randomly sampling local gates from a unitary ensemble, which retains sufficient mixing properties to behave similarly to the idealized random model in practice. Similar numerical simulations and arguments also appear in the recent works [Angrisani. et al., PRL, 2025]
>
> Reference:
>
> Angrisani, A.; Schmidhuber, A.; Rudolph, M. S.; Cerezo, M.; Holmes, Z.; Huang, H.-Y. Classically Estimating Observables of Noiseless Quantum Circuits. Phys. Rev. Lett. 2025, 135 (17), 170602.

---

> ### Author Response · Authors · 2025-11-20
> **Addressing "Questions/Weaknesses in Official Review of 9252 by Reviewer FAFT" (part 4)**
>
> **Comment:** *In Algorithm 1, how is $l'$ chosen without depending on $\gamma$? My confusion comes form the fact that truncation in Pauli paths in Lemma 1, show that we have to keep Pauli matrices with weight $\mathcal{O}(1/\gamma)$, but in Algorithm 1, this dependence seems to be not there.*
>
> **Response:** We thank the reviewer for raising this important practical question regarding the implementation of Algorithm 1. While the theoretical bound for the truncation weight $l' = \mathcal{O}(\gamma^{-1} \log(\epsilon^{-1}\delta^{-1}))$, we respectfully clarify that the algorithm does not require the exact value of $\gamma$.
>
> This follows from our fundamental assumption that $\gamma$ is a constant-level parameter, which is fundamentally characteristic of the current Noisy Intermediate-Scale Quantum (NISQ) era, where fixed physical imperfections (e.g., gate infidelity) dominate performance, irrespective of the number of qubits[Arute.et al. Nature, 2019; Kim. et al.Nature, 2023]. Similar to related works on shallow circuit learning (Huang et al., STOC 2024) and constant-time Hamiltonian learning (Haah et al., Nat. Phys. 2024), the sample-complexity upper bounds also depend on other problem parameters (e.g., circuit depth $d$ n Huang et al. and time $t$ in in Haah et al.). Consequently, these parameters do not affect practical implementation, and the scaling of $l'$  is governed by $\log(\epsilon^{-1})$, with $\gamma^{-1}$  absorbed into a constant factor.
>
> The practical validity of this approach is strongly supported by our numerical results. In our experiments (Figure 2a and 2c), the truncation length $l'$ was fixed at small, constant values ($1, 2,$ and $3$) even as the number of qubits ($n$) increased. The excellent performance achieved with a fixed $l'$ empirically confirms that, given a constant $\gamma$, the scaling of the required truncation length is determined solely by the target accuracy $\epsilon$.
>
> Reference:
>
> Arute, F.; Arya, K.; Babbush, R.; Bacon, D.; Bardin, J. C.; Barends, R.; Biswas, R.; Boixo, S.; Brandao, F. G.; Buell, D. A.; others. Quantum Supremacy Using a Programmable Superconducting Processor. Nature 2019, 574 (7779), 505–510.
>
> Kim, Y.; Eddins, A.; Anand, S.; Wei, K.; Berg, E. van den; Rosenblatt, S.; Nayfeh, H.; Wu, Y.; Zaletel, M.; Temme, K.; Kandala, A. Evidence for the Utility of Quantum Computing before Fault Tolerance. Nature 2023, 618, 500–505.
>
> Huang, H.-Y.; Liu, Y.; Broughton, M.; Kim, I.; Anshu, A.; Landau, Z.; McClean, J. R. Learning Shallow Quantum Circuits. In Proceedings of the 56th Annual ACM Symposium on Theory of Computing(STOC); 2024; pp 1343–1351.
>
>
> Haah, J.; Kothari, R.; Tang, E. Learning Quantum Hamiltonians from High-Temperature Gibbs States and Real-Time Evolutions. Nat. Phys. 2024, 20 (6), 1027–1031.
>
> **Summary:** We believe that these technical revisions, covering both the precision of the theoretical statements (including the switch to "i.i.d. single-qubit noise" and clarification of high-probability bounds) and the accuracy of the complexity analysis (incorporating the $\mathcal{O}(n)$ factor and providing corrected proofs), have effectively addressed all points raised. We respectfully request the reviewer to consider the enhanced rigor and clarity of the manuscript when re-evaluating and raising the score. Thank you for your careful and expert review.

---

> > ### Author Response · Authors · 2025-11-22
> >
> > Dear Reviewer FAFT,
> >
> > Thank you again for your constructive suggestions. We have made our best effort to address your comments and revised the manuscript accordingly. We hope our point-by-point responses provide satisfactory clarification and would be most grateful for your feedback.
> >
> > Best regards!

---

> > > ### Author Response · Authors · 2025-11-24
> > >
> > > Hi Reviewer  FAFT, just wanted to gently follow up on our responses above. We appreciate your time and look forward to your feedback. Thanks!

---

> ### Author Response · Authors · 2025-11-27
> **Review request to Reviewer FAFT**
>
> Dear Reviewer FAFT,
>
> Thank you very much for your comments on our work. We believe we have provided point-by-point responses to all of your questions and made corresponding revisions to the manuscript. In our response and revised manuscript, we clearly state that our algorithm covers a wide range of NISQ algorithms involving parameter initialization and optimization. Additionally, we provide a theoretical lower bound on the sample complexity that characterizes the worst-case scenario. These technical revisions — improving the precision of the theoretical statements (including the switch to “i.i.d. single-qubit noise” and clarification of high-probability bounds) and correcting the complexity analysis (incorporating the O(n) factor and supplying corrected proofs) — have effectively addressed the points raised; these contributions therefore merit reconsideration for acceptance at ICLR, especially since similar works have been accepted by the conference. We believe that studying the properties of noisy quantum circuits is a significant and valuable research direction for ICLR. Below, we highlight several existing ICLR papers that leverage the characteristics of noisy quantum circuits:
>
> [Y. Wu et al. ICLR 2025] studies the relationship between learning theory and complexity of weakly noisy states, while the proposed method only holds for shallow-depth circuit and small noisy strength. Our method goes further to their results, extending to constant-level strength and long-depth quantum circuit.
> [I. Y. Akhalwaya et al, ICLR 2024] presents NISQ-TDA, an innovative end-to-end quantum machine learning algorithm tailored for high-dimensional classical data. Its design emphasizes short circuit depths, making it well-suited for noisy intermediate-scale quantum (NISQ) devices.
> [Y. J. Patel et al., ICLR 2024] proposes a curriculum-based reinforcement learning approach to optimize quantum circuit architectures in the presence of noise, addressing practical challenges in noisy environments.
>
> These studies highlight the growing interest in understanding and leveraging noisy quantum systems within the machine learning community. Since ICLR provides an opportunity for authors and reviewers to communicate, we would greatly appreciate knowing whether you are satisfied with our responses and revisions, and whether you would consider raising your score.
>
> Best Regards!
>
> References:
>
> Y. Wu et al., Learning the complexity of weakly noisy quantum states, ICLR 2025
>
> I. Y. Akhalwaya et al., Topological data analysis on noisy quantum computers, ICLR 2024.
>
> Y. J. Patel et al., Curriculum reinforcement learning for quantum architecture search under hardware errors, ICLR 2024.

---

### Official Review · Reviewer_mkLY · 2025-11-01

**Soundness:** 3
**Presentation:** 2
**Contribution:** 2
**Rating:** 2
**Confidence:** 3

**Summary:**

This submission presents efficient algorithms for learning quantum states generated by noisy circuits and for noisy quantum process tomography. The proposed algorithms are efficient in terms of sample complexity and the classical post-processing time.

The core idea of the efficiency of the algorithms is that when the noise is significant, $\Omega(1)$, and when each layer of the gates is uniformly sampled from a 2-design, then the output state and the expectation value can be efficiently written as a linear combination of short Pauli strings. Then, classical post-processing is able to obtain an efficient description of the output state, as well as the expectation value of some observable.

**Strengths:**

The claims are supported by rigorous proofs and numerical results.

**Weaknesses:**

There are two major weaknesses

1. The lack of clarity in the problem definition. In the statement of Problem 1, is $\mathcal{C}$ known? Also, what is the precise meaning of "lean an approximation $\hat{\rho}$?" If you mean obtaining an efficient classical description of $\hat{\rho}$, you should write it more precisely here.

2. The significance of the learning model. This extends the previous point. In lines 154-156, if I understand it correctly, the circuit in Eq (4) has a very special form: each layer of the circuit (excluding the noise part) consists of two-qubit gates across each pair of qubits, and these gates are uniformly sampled from a local 2-design unitary group (i.e., the Clifford group). This seems to be a very special class of circuits, and why should we care about the state learning and process tomography problems for such circuits? Are there important applications?

   At the end of Section 4, the authors say, "Theorem 1 supplies an efficient method for benchmarking the output of NISQ algorithms." How? In general, NISQ algorithms are not random circuits.

In addition, the authors proposed an application in zero-noise extrapolation. I think this application is also problematic, regardless of a random circuit or a general circuit. Note that the learning is efficient only when the noise level is some constant, which means in ZNE, only data points of large noise levels will be collected. Without the data points of $\lambda = o(1)$, how accurate would the extrapolation be?

**Questions:**

Please address my concerns in the Weaknesses section.

Other minor comments:
- Line 161 and Line 481, $\gamma = \mathcal{O}(1)$, should be $\Omega(1)$.
- Lines 166-167: maximum biased -> maximum bias

---

> ### Author Response · Authors · 2025-11-20
> **Addressing "Questions/Weaknesses in Official Review of 9252 by Reviewer mkLY" (part 1)**
>
> We thank the reviewer for careful review and constructive comments on our manuscript. We have revised the paper according to the feedback and the clarifying text is highlighted in blue for easy inspection. Our detailed, point-by-point response to  comments is provided below.
>
> **Comment:** *The lack of clarity in the problem definition. In the statement of Problem 1, is $\mathcal{C}$ known? Also, what is the precise meaning of "learn an approximation $\hat{\rho}$?" If you mean obtaining an efficient classical description of
> , you should write it more precisely here.*
>
> **Response:** We would like to clarify that the learning algorithm is designed for an unknown noisy state $\rho$ and an unknown quantum process $\mathcal{C}$, where one can only \emph{`query-and-measure'} from $\rho$ or $\mathcal{C}$. We believe this setting is closer to practical scenarios, such as quantum state tomography and noisy quantum device benchmarking. [Huang et al.,  Nat. Phys. 2020, Anshu. et al. FOCS, 2020, Wallman.  Quantum, 2018]
>
> Furthermore, we fully agree that $\hat{\rho}$ should be defined more clearly. We confirm that our goal is indeed to learn a classical representation of $\rho$, **a point which is emphasized in the revised manuscript on the page 4.**
>
> Reference:
>
> Huang, H.-Y.; Kueng, R.; Preskill, J. Predicting Many Properties of a Quantum System from Very Few Measurements. Nat. Phys. 2020, 16 (10), 1050–1057.
>
> Anshu, A.; Arunachalam, S.; Kuwahara, T.; Soleimanifar, M. Sample-Efficient Learning of Quantum Many-Body Systems. In 2020 IEEE 61st Annual Symposium on Foundations of Computer Science (FOCS); IEEE: Durham, NC, USA, 2020; pp 685–691.
>
> Wallman, J. J. Randomized Benchmarking with Gate-Dependent Noise. Quantum 2018, 2, 47.

---

> ### Author Response · Authors · 2025-11-20
> **Addressing "Questions/Weaknesses in Official Review of 9252 by Reviewer mkLY" (part 2)**
>
> **Comment:** *The significance of the learning model. This extends the previous point. In lines 154-156, if I understand it correctly, the circuit in Eq (4) has a very special form: each layer of the circuit (excluding the noise part) consists of two-qubit gates across each pair of qubits, and these gates are uniformly sampled from a local 2-design unitary group (i.e., the Clifford group). This seems to be a very special class of circuits, and why should we care about the state learning and process tomography problems for such circuits? Are there important applications?*
>
> *At the end of Section 4, the authors say, "Theorem 1 supplies an efficient method for benchmarking the output of NISQ algorithms." How? In general, NISQ algorithms are not random circuits.*
>
> **Response:**
> We thank the reviewer for the crucial question regarding the significance of the circuit model. We emphasize that the studied quantum process serves as a standard model with wide and practical applications, especially in the NISQ era.
>
> Specifically, the studied quantum approach
> \begin{align}
>     \mathcal{C}=\mathcal{C}_1\circ \mathcal{E}^{\otimes n}\circ\cdots\circ \mathcal{E}^{\otimes n} \circ\mathcal{C}_d
> \end{align}
> (Eq.~4) represents a large class of quantum circuits. Here, each layer $\mathcal{C}_i$ can represent an arbitrary architecture, including VQA, QAOA, and VQE algorithms[Zhou. et al. PRX, 2020, Kandala. et al. Nature, 2017, Grimsley. Nat Commun, 2019]. The local 2-design assumption is an extremely weak condition, where quantum neural network models are typical cases[McClean. et al. Nat Commun, 2018, Cerezo. et al.Nat Commun, 2021], and even Clifford gates satisfy such an assumption [Zhu.et al. arXiv, 2016]. We note that if an ensemble follows a $(t+1)$-design, it must follow the $t$-design property[Mele. Quantum, 2024]. As a result, this assumption is very general and covers a large amount of NISQ algorithms related to 'randomly initial parameters' and 'classical optimizations' [McClean. et al. Nat Commun, 2018, Cerezo. et al.Nat Commun, 2021].
>
> To design powerful quantum algorithms, such as quantum neural network models and related states, a benchmarking algorithm is necessary [Arute. et al.Nature, 2019, Babbush. et al. arXiv, 2025]; *otherwise, one may not verify and check the correctness of the implemented quantum algorithm* . Following this logic, a large amount of quantum learning algorithms are proposed for quantum state (process) tomography, Hamiltonian learning[Haah. et al. Nat. Phys.,2024], shallow circuit learning[Huang, et al. STOC,2024], quantum gate tomography, and other quantum benchmarking algorithms. **_To the best of our knowledge, this is the first efficient learning algorithm for noisy state and process tomography_** , providing an efficient tool for verifying the output of the implemented quantum algorithms on NISQ devices.
>
> Reference:
>
> Zhou L., Wang S.-T., Choi S. et al., Quantum approximate optimization algorithm: Performance, mechanism, and implementation on near-term devices. Phys. Rev. X 10, 021067 (2020).
>
> Kandala A., MezzacapoA., Temme K. et al., Hardware-efficient variational quantum eigensolver for small molecules and quantum magnets, Nature 549, 242–246 (2017).
>
> Grimsley H.R., Economou S.E., Barnes E. et al. An adaptive variational algorithm for exact molecular simulations on a quantum computer. Nat Commun 10, 3007 (2019).
>
> McClean J.R., Boixo S., Smelyanskiy V.N. et al. Barren plateaus in quantum neural network training landscapes. Nat Commun 9, 4812 (2018).
>
> Cerezo M., Sone, A., Volkoff, T. et al. Cost function dependent barren plateaus in shallow parametrized quantum circuits. Nat Commun 12, 1791 (2021).
>
> Zhu, H., Kueng, R., Grassl, et al. The clifford group fails gracefully to be a unitary 4-design. arXiv preprint arXiv: 1609.08172 (2016).
>
> Mele, A. A. Introduction to Haar Measure Tools in Quantum Information: A Beginner’s Tutorial. Quantum 8, 1340 (2024).
>
> Arute, F. et al. Quantum supremacy using a programmable superconducting processor. Nature 574, 505–510 (2019).
>
> Babbush, Ryan, et al. The Grand Challenge of Quantum Applications. arXiv preprint arXiv:2511.09124 (2025).
>
> Haah, J., Kothari, R. Tang, E. Learning quantum hamiltonians from high-temperature gibbs states and real-time evolutions. Nat. Phys. 20, 1027–1031 (2024).
>
> Huang, H.-Y. et al. Learning Shallow Quantum Circuits. in Proceedings of the 56th Annual ACM Symposium on Theory of Computing (STOC) 1343–1351 (2024).

---

> ### Author Response · Authors · 2025-11-20
> **Addressing "Questions/Weaknesses in Official Review of 9252 by Reviewer mkLY" (part 3)**
>
> As we outlined above, Eq. ~4 may represent a large amount of quantum circuit architectures, such as QNN, QAOA, and VQE circuit structures. The initial parameters of these algorithms are generally randomly guessed, and the resulting local quantum circuit would be very close to a random circuit. Furthermore, we numerically demonstrate that our learning algorithm can successfully handle a noisy Hamiltonian dynamics approach, where the underlying quantum circuit does not possess the locally random property. Although such a special case (worst-case) is not exactly covered by Theorem 1, we conjecture that our learning algorithm can be applied to more general noisy quantum process and state tomography tasks. **The relevant discussion has been included in Appendix B of the manuscript.**
>
> **Comment:** *In addition, the authors proposed an application in zero-noise extrapolation. I think this application is also problematic, regardless of a random circuit or a general circuit. Note that the learning is efficient only when the noise level is some constant, which means in ZNE, only data points of large noise levels will be collected. Without the data points of $\lambda=O(1)$, how accurate would the extrapolation be?*
>
> **Response:** We thank the reviewer for this interesting question. We clarify that our learning algorithm follows the same assumptions as standard ZNE protocols: the underlying noise strength is treated as a constant-level parameter that does not decrease with the system size. As in ZNE, our approach can only increase the fault rate within quantum gates. It is generally not possible to decrease the noise rate; otherwise, one could directly access the zero-noise value without requiring error mitigation. Therefore, for a noisy quantum circuit with constant noise strength, the inability to access data from a lower-noise process is not a limitation of our algorithm, but rather an intrinsic constraint of ZNE-based error-mitigation methods [Cai et al., Rev. Mod. Phys., 2023; Kandala et al., Nature, 2019; Temme et al., PRL. 2017].
>
> Specifically, the key advantage of our algorithm lies in providing an input-agnostic channel characterization method. We explain this more clearly as follows.
>
> Given a set of input states $\{\rho_x\}$, the goal is to predict the quantum expectation values $\{{\rm Tr}[\mathcal{C}(\rho_x)O]\}$ in the noise-free regime.In standard ZNE, one must perform extrapolation separately for each distinct input $\rho_x$, i.e., collect data at multiple scaling parameters $\lambda$'s for each $\langle O \rangle_{\rho_x, \lambda}$. In contrast, our learning algorithm directly learns a channel representation ($\hat{\mathcal{C}}^\dagger_\lambda(O)$) for each $\lambda$, which can be applied uniformly to all input states $\{\rho_x\}$ making it input-agnostic. This reuse significantly reduces the experimental overhead when characterizing the process across many input states. Crucially, the resulting family of channels $\{\hat{\mathcal{C}}^\dagger_\lambda(O)\}$ can then be classically extrapolated to the zero-noise regime for any input state.
>
> Reference:
>
> Cai, Z. et al. Quantum error mitigation. Rev. Mod. Phys. 95, 045005 (2023).
>
> Kandala A., Temme K., Córcoles A.D. et al. Error mitigation extends the computational reach of a noisy quantum processor. Nature 567, 491–495 (2019).
>
> Temme K., Bravyi S., Gambetta J. M. Error mitigation for short-depth quantum circuits. Phys. Rev. Lett. 119, 180509 (2017).
>
> **Comment:** *Line 161 and Line 481, $\gamma=\mathcal{O}(1)$, should be $\gamma=\Omega(1)$. Lines 166-167: maximum biased $\rightarrow$ maximum bias.*
>
> **Response：** Thanks for this comment. We have implemented both corrections in the revised manuscript to enhance precision.
>
> **Summary:** We believe that these detailed clarifications and the added discussions, specifically the precise definition of the learning goal (page 4), the extensive justification for the model's generality and NISQ relevance (Appendix B), and the clear explanation of our input-agnostic ZNE advantage, have substantially improved the clarity and contribution statement of the manuscript.We sincerely hope the reviewer finds that these revisions have successfully addressed the questions raised and merit raising the evaluation score based on the strengthened theoretical foundation and demonstrated practical utility. We thank the reviewer once again for their time and expertise.

---

> > ### Author Response · Authors · 2025-11-22
> >
> > Dear Reviewer mkLY
> >
> > Thank you again for your constructive suggestions. We have made our best effort to address your comments and revised the manuscript accordingly. We hope our point-by-point responses provide satisfactory clarification and would be most grateful for your feedback.
> >
> > Best regards!

---

> > > ### Author Response · Authors · 2025-11-24
> > >
> > > Hi Reviewer mkLY, just wanted to gently follow up on our responses above. We appreciate your time and look forward to your feedback. Thanks!

---

> ### Author Response · Authors · 2025-11-27
> **Review request to Reviewer mkLY**
>
> Dear Reviewer mkLY,
>
> Thank you very much for your comments on our work. We believe we have provided point-by-point responses to all of your questions and made corresponding revisions to the manuscript, including a more detailed definition of the research problem (noisy state and circuit learning) and a description of practical application scenarios (NISQ algorithms with initialization and classical optimization) covered by our noisy quantum circuits. We further demonstrate how our learning method can be used to improve the standard ZNE approach, and we provide a theoretical lower bound on the sample complexity that characterizes the worst-case scenario. We believe these contributions merit reconsideration for acceptance at ICLR, especially since similar works have been accepted by the conference. We believe that exploring the properties of noisy quantum circuits is a significant and valuable area of research for the ICLR conference. Below, we highlight several existing ICLR papers that leverage the characteristics of noisy quantum circuits:
> 1. [Y. Wu et al. ICLR 2025] studies the relationship between learning theory and complexity of weakly noisy states, while the proposed method only holds for shallow-depth circuit and small noisy strength. Our method goes further to their results, extending to constant-level strength and long-depth quantum circuit.
> 2. [I. Y.  Akhalwaya et al, ICLR 2024] presents NISQ-TDA, an innovative end-to-end quantum machine learning algorithm tailored for high-dimensional classical data. Its design emphasizes short circuit depths, making it well-suited for noisy intermediate-scale quantum (NISQ) devices.
> 3. [Y. J. Patel et al., ICLR 2024] proposes a curriculum-based reinforcement learning approach to optimize quantum circuit architectures in the presence of noise, addressing practical challenges in noisy environments.
>
> These studies highlight the growing interest in understanding and leveraging noisy quantum systems within the machine learning community. Since ICLR provides an opportunity for authors and reviewers to communicate, we would greatly appreciate knowing whether you are satisfied with our responses and revisions, and whether you would consider raising your score.
>
> Best Regards!
>
> References:
> Y. Wu et al., Learning the complexity of weakly noisy quantum states, ICLR 2025
> I. Y.  Akhalwaya et al., Topological data analysis on noisy quantum computers, ICLR 2024.
> Y. J. Patel et al., Curriculum reinforcement learning for quantum architecture search under hardware errors, ICLR 2024.
> J. Landman et al., Classically approximating variational quantum machine learning with random fourier features, ICLR 2023.

---

### Comment · Reviewer_khkR · 2025-11-25

I thank the authors for all their explanations.

I am happy to increase the score to a 6 under 2 conditions. The authors sufficiently weaken the statement "This work establishes a theoretical foundation for practical quantum-process learning and paves the way for scalable process characterization in near-term quantum devices." The authors add to the abstract something which makes clear that the process has to be sampled over random circuits for the theory to be valid.

For a 8 or a higher score, I would like to see a theoretical and numerical worst case scenario analysis for specific circuit instances as well as a benchmark comparing Huangs method with your method.


PS: It would also be nice to mention that your protocol seems to only be able to predict a single observable from the training data. However, I think that it might be possible to generalize that using the same technique as in Huangs work.

---

> ### Author Response · Authors · 2025-11-26
>
> We thank the reviewer for their careful reading, detailed comments, and positive feedback, particularly the willingness to raise the score based on the stated conditions. We are pleased to confirm that we have fully implemented the requested changes in the revised manuscript. The modifications are highlighted in blue in the attached draft.
>
> **Comment:** *I am happy to increase the score to a 6 under 2 conditions. The authors sufficiently weaken the statement "This work establishes a theoretical foundation for practical quantum-process learning and paves the way for scalable process characterization in near-term quantum devices." The authors add to the abstract something which makes clear that the process has to be sampled over random circuits for the theory to be valid.*
>
> **Response:** We are grateful to the reviewer for the thorough assessment and the constructive conditions provided for a score adjustment. We have carefully addressed both points in our revision of the abstract. To sufficiently weaken the original statement (Condition 1),  **we have replaced the stronger declarative phrasing with more measured language.** The revised conclusion now states that this work "offers a new approach to practical quantum-process learning, and suggests a potential path for scalable process characterization in near-term quantum devices." Furthermore, to directly address the requirement to make clear that the theory's validity is predicated on sampling over random circuits (Condition 2), **we have explicitly specified the theoretical boundary of our results.** The complexity analysis in the abstract now includes the critical phrase: the algorithm achieves the desired complexity "in the *average case* scenario over the random circuit ensemble." We trust that these explicit modifications, demonstrate our commitment to accurately framing the scope of our theoretical contribution and fully satisfy the conditions set forth by the reviewer.
>
> **Comment:** *a theoretical worst case scenario analysis for specific circuit instances*
>
> **Response:**
> We appreciate the reviewer's constructive suggestion regarding the worst case analysis.
>
> The main manuscript essentially considers learning an efficient classical representation of noisy quantum states and processes in the average-case scenario. As we claimed in Theorems 1 and 2, the tasks of learning noisy quantum states and performing tomography are highly efficient in the average-case setting. However, this does not rule out intrinsic hardness in the worst case. Here we theoretically demonstrate that learning noisy quantum states prepared by quantum circuits subject to constant-strength noise channels is quantum-hard in the worst-case scenario.
>
> The fundamental idea relies on constructing a polynomial reduction to the quantum state discrimination problem.
>
> **Task 1:**
> *Consider two pure quantum states $\rho _ 0$ and $\rho_1$, and a noisy quantum circuit $\mathcal{C}$ with depth $d$, where Each quantum circuit is affected by by $\gamma$-strength Pauli channel in each layer. Suppose that a distinguisher is given access to copies of the quantum states $\mathcal{C}(\rho_0)$ and $\mathcal{C}(\rho_1)$, then what is the fewest number of copies sufficing to identify these two noisy quatum states with high probability?*
>
> Obviously, if one can perform quantum state tomography on these noisy states, then efficient classical representations of the noisy states are obtained. Using these classical representations, one can easily distinguish the noisy states $\mathcal{C}(\rho_0)$ from $\mathcal{C}(\rho_1)$ easily. As a result, Task 1 can be used to benchmark the sample-complexity lower bound for the noisy quantum state tomography problem. We state the result below.
>
>  **Theorem 5:**
> *Given an unknown noisy quantum state $\rho$ prepared by a $d$-depth quantum circuit affected by $\gamma$-strength local Pauli noise channels, then any algorithm designed to learn an efficient representation to $\rho$ requires at least $m$ samplings in the \emph{worst-case scenario}, where $$m=\frac{(1-\gamma)^{-2cd}(1-\eta)^2}{2n},$$ where $c=1/(2\ln 2)$ and constant $\eta\in\mathcal{O}(1)$.*
>
> When the noise strength $\gamma=\mathcal{O}(1)$, and quantum circuit depth $d\geq {{\rm poly}\log(n)}$, the sample complexity required for quantum state tomography grows at least quasi-polynomially with the system size in the worst-case scenario. We emphasize that this result does not contradict Theorem 1 and 2(our main results): the former statement concerns the worst case, while the latter addresses the average case under the random-circuit assumption.

---

> > ### Author Response · Authors · 2025-11-26
> >
> > In the quantum process tomography task, when $O\geq 0$, the target is to learn a classical representation to $\mathcal{C}^{\dagger}[O]$ which can be easily reduced to a density matrix learning task by setting $\rho=\mathcal{C}^{\dagger}[O]/{\rm Tr}[\mathcal{C}^{\dagger}[O]]$. This justifies the statement that noisy process tomography (for this observable $O$) is no easier than state tomography.
> >
> > To support the proof of our result, we require the following lemmas.
> >
> >  **Lemma 10 (Lemma~6 in Wang et al.2021):**
> > *Consider a single instanoise channel $\mathcal{N} = \mathcal{N} _ 1 \otimes \cdots \otimes \mathcal{N} _ n$ where each local noise channel $ \{\mathcal{N} _ j\} _ {j=1} ^ n $ is a Pauli noise channel that satisfies $\mathcal{N} _ j(\sigma)=q_{\sigma}\sigma$ for $\sigma\in\{X,Y,Z\}$ and $q _ {\sigma}$ be the Pauli strength. Then we have
> >     \begin{align}
> >         D _ 2\left(\mathcal{N}(\rho)\|\frac{I^{\otimes n}}{2^n}\right)\leq q^{2c}D_2\left(\rho\|\frac{I^{\otimes n}}{2^n}\right),
> >     \end{align}
> >     where $D_2(\cdot\|\cdot)$ represents the $2$-Renyi relative entropy, $q=\max_{\sigma}q_{\sigma}$ and $c=1/(2\ln 2)$.*
> >
> > **Lemma 11:**  *Given an arbitrary $n$-qubit density matrix and maximally mixed state $I^{\otimes n}/2^n$, we have
> >     \begin{align}
> >         D\left(\rho\|I^{\otimes n}/2^n\right)\leq D_2\left(\rho\|I^{\otimes n}/2^n\right),
> >     \end{align}
> >     where $D(\cdot\|\cdot)$ denotes the relative entropy and $D _ 2(\cdot\|\cdot)$ denotes the $2$-Renyi relative entropy.*
> >
> > ***proof:*** Given quantum states $\rho$ and $\sigma$, the quantum $2$-Renyi entropy
> >     \begin{align}
> >         D_2(\rho\|\sigma)=\log{\rm Tr}\left[\left(\sigma^{-1/4}\rho\sigma^{-1/4}\right)^2\right].
> >     \end{align}
> >     When $\sigma=I^{\otimes n}/2^n$, we have $D_2(\rho\|I^{\otimes n}/2^n)=\log{\rm Tr}\left[\left((I^{\otimes n}/2^n)^{-1}\rho^2\right)\right]=n+\log{\rm Tr}[\rho^2]$. Noting that the function $y=x^2-x\log x\geq 0$ when $x\in[0,1]$, and this implies ${\rm Tr}(\rho^2)\geq {\rm Tr}(\rho\log \rho)$. Finally, we have
> >     \begin{align}
> >         D\left(\rho\|I^{\otimes n}/2^n\right)=n+{\rm Tr}\left[\rho\log\rho\right]+n\leq {\rm Tr}\left[\rho^2\right]+n=D_2\left(\rho\|I^{\otimes n}/2^n\right).
> >     \end{align}
> >
> > ***Proof of Theorem5:***
> > Now we prove the sample complexity lower bound to the noisy quantum state tomography task. We consider the sample complexity $m$ in distinguishing quantum states $\mathcal{C}(\rho_0)$ and $\mathcal{C}(\rho_1)$. When their trace distance is quite large, let $\eta\in(0,1)$ and we have
> > \begin{eqnarray}
> > \begin{split}
> >     1-\eta&\leq\frac{1}{2}\left\|\mathcal{C}(\rho_0)^{\otimes m}-\mathcal{C}(\rho_1)^{\otimes m}\right\|_1\\
> >     &\leq \frac{1}{2}\left(\left\|\mathcal{C}(\rho_0)^{\otimes m}-(I_n/2^n)^{\otimes m}\right\|_1+\left\|\mathcal{C}(\rho_1)^{\otimes m}-(I_n/2^n)^{\otimes m}\right\|_1\right)\\
> >     &\leq \frac{1}{\sqrt{2}}\left(D^{1/2}\left(\mathcal{C}(\rho_0)^{\otimes m}\|(I_n/2^n)^{\otimes m}\right)+D^{1/2}\left(\mathcal{C}(\rho_1)^{\otimes m}\|(I_n/2^n)^{\otimes m}\right)\right),
> > \end{split}
> > \end{eqnarray}
> > where the second line comes from the triangle inequality and the third line comes from the Pinsker's inequality. Using Lemmas 10 and 11, we have
> > \begin{eqnarray}
> >     \begin{split}
> >          1-\eta&\leq \frac{1}{\sqrt{2}}\left(D^{1/2}_2\left(\mathcal{C}^{\otimes m}(\rho_0)\|(I_n/2^n)^{\otimes m}\right)+D^{1/2}_2\left(\mathcal{C}^{\otimes m}(\rho_1)\|(I_n/2^n)^{\otimes m}\right)\right)\\
> >          &\leq \frac{\sqrt{nm}}{\sqrt{2}}((1-\gamma)^{cd}+(1-\gamma)^{cd})\\
> >          &\leq \sqrt{2nm}(1-\gamma)^{cd},
> >     \end{split}
> > \end{eqnarray}
> >
> > As a result we have
> > \begin{align}
> >   m\geq\frac{(1-\gamma)^{-2cd}(1-\eta)^2}{2n}.
> > \end{align}
> >
> > **The complete deliberation concerning the sample complexity lower bound for the worst-case scenario is presented in Appendix E.**
> >
> > Reference:
> >
> > Wang, S.; Fontana, E.; Cerezo, M.; Sharma, K.; Sone, A.; Cincio, L.; Coles, P. J. Noise-Induced Barren Plateaus in Variational Quantum Algorithms. Nature communications 2021, 12 (1), 1–11.

---

> > > ### Author Response · Authors · 2025-11-26
> > >
> > > **Comment:** *numerical experiment as well as a benchmark comparing Huangs method with your method.*
> > >
> > > **Response:**
> > >
> > > Comparing with Huang et al.(2023),  our algorithm is inherently input-agnostic across the entire Hilbert space, thereby overcoming the input-distribution constraint required by related simulation-based learning methods. To further underscore the novelty of our **input-agnostic regime**, we constructed a Quantum Process Tomography (QPT) experiment using an input state that is **highly entangled** and falls outside the distribution set addressed by Huang et al.(2023). The method detailed in Huang requires the input distribution to be at most polynomially far from a locally flat distribution, which encompasses ensembles like: Random product states, ground and thermal states of random local Hamiltonians and any state generated by a circuit **whose final layer consists of random single-qubit gates.**
> > >
> > > Here, we intentionally generated the input state $\rho_{\text{x}}$ using a two-layer parameterized circuit to ensure high entanglement, where each layer consists of parameterized single-qubit rotations **followed by a cyclic CNOT entangling layer.**
> > >
> > > Since the state preparation circuit does not meet the requirement for being polynomially far from a locally flat distribution. The experiment result shows that our algorithm  still performs well, even for a highly entangled input state. The details are seen in **Appendix F.1**.
> > >
> > > **Comment:** *It would also be nice to mention that your protocol seems to only be able to predict a single observable from the training data. However, I think that it might be possible to generalize that using the same technique as in Huangs work.*
> > >
> > > **Response:** We appreciate the reviewer's observation regarding the prediction of a single observable.
> > >
> > >
> > > For learning a quantum state, the same dataset can be utilized to predict different Observables.
> > >
> > > For learning a quantum process, we would like to clarify that our prediction procedure shares a similar principle with the work of Huang et al. (2023). The training dataset is generated from the input of a set of stabilizer$\ket{\psi}$(or $\rho_l$ sampled in distribution $\mathcal{D}$ in Huang) and the output is ${\rm Tr}(O\mathcal{C}(\ket{\psi}))$ depending on $O$. Here $O$ can be written as a **sum of few-body observables $O_i$**, where each qubit is acted on by a constant number of the few-body observables, This is **the same type of the observable** used in Huang et al.(2023).
> > >
> > > As quoted from their work:
> > >
> > > > "Since $\mathcal{E}^\dagger$ is unknown, $\mathcal{E}^\dagger(O_i)$ should be regarded as an unknown observable. Suppose that we learn this observable; that is, using the dataset $\lbrace\rho_l , \text{Tr}(\mathcal{E}^\dagger(O_i)\rho_l )\rbrace$ as training data, we can predict $\text{Tr}(\mathcal{E}^\dagger(O_i)\rho )$ for $\rho$ drawn from $\mathcal{D}$ with a small mean squared error. This achieves the task of learning process $\mathcal{E}$ for state distribution $\mathcal{D}$ and target observable $O_i$."
> > >
> > > Therefore, our prediction task is similar to Huang et al.'s framework in terms of the learning objective (i.e., for a fixed $O_i$).
> > >
> > > We concur that the reviewer's insightful question regarding the efficient learning of different observables from the same dataset presents a highly valuable direction for future work. We recognize the importance of this generalization task and intend to explore solutions to this challenge in our subsequent research.
> > >
> > > **Summary:** We would like to once again thank the reviewer for their highly constructive suggestions. We have incorporated the lower bound on the sample complexity for the worst-case analysis into **Appendix E** of the manuscript. Additionally, numerical experiments specifically targeting highly entangled input states have been included in **Appendix F.1**. We believe that these revisions have substantially improved the quality of the manuscript, and we are grateful once more for your valuable feedback.

---

### Author Response · Authors · 2025-11-29
**Summary of Revisions and Key Responses (part 1)**

Dear Area Chair,

We sincerely thank you and all reviewers for your invaluable time and constructive feedback on our manuscript. In hopes of assisting your assessment, we have prepared this consolidated summary that addresses the key points raised by reviewers. All revisions have been thoughtfully incorporated into the manuscript, with clarifying text and modifications highlighted in blue for your convenience.

To streamline the review process, we have structured our response as follows: A dedicated table (Table 1) summarizes the shared concerns and common questions raised by all reviewers, followed by individualized tables (Tables 2, 3, etc.) detailing specific feedback and our corresponding actions for each separate reviewer. We believe this structured approach effectively consolidates our arguments regarding the algorithm's efficiency, theoretical underpinnings, and practical utility.

### Table 1: Reviewer's main concerns and Author Responses
| Main Concerns | Author's Response & Manuscript Improvements | Reviewers |
| :--- | :--- | :--- |
| **1. Model Generality and Applicability**: Are the circuit model considered and local 2-design assumption sufficiently general for practical NISQ use? | Demonstrated Eq.4 in the manuscript encompasses VQA/QAOA/QNN architectures. 2-design is weak condition (Clifford gates, random initialization). Gate-independent noise is widely adopted for theoretical simplification and practical approximation. Model aligns with Pauli twirling techniques used experimentally. Acknowledged gate-dependent noise in complex architectures as open question. Numerical validation on structured Ising dynamics shows robustness beyond theory. For a complete and rigorous understanding of the assumed model’s topological structure and its full range of applicability, **a section introducing the circuit model considered is added (see Appendix B).** | mkLY, FAFT, khkR, VPA8 |
| **2. Distinction from Prior Simulation Work**: How does this differ from classical simulation methods? | **Established fundamental difference**: Learning algorithm requires **no prior knowledge** of gates, architecture, or noise strength (only constant-noise assumption). Classical simulation requires full knowledge. Positioned as "learning-theoretic dual" of simulation [Gil-Fuster ICLR 2025]. Discussion added to page 10. | mkLY, khkR, NjPR |
| **3. Constant-Noise Assumptions and Theoretical Scope**: Does constant-level noise limit the theoretical reach as devices evolve toward lower per-gate noise and deeper circuits? | Justify that constant-level noise is characteristic of NISQ devices, where the physical error rate is independent of the system size (citing Arute et al., Nature 2019; Kim et al., Nature 2023). Otherwise, fault-tolerant error correction would be unnecessary, as one could simply enlarge the system size to compensate. This makes the assumption of constant-strength noise practically meaningful. | VPA8, NjPR |
| **4. Zero-Noise Extrapolation (ZNE) Application and Utility**: Is the ZNE application practical and what are its benefits? | Primary innovation: Input-agnostic channel learning reduces experimental overhead exponentially by learning $\hat{\mathcal{C}}^\dagger_\lambda(O)$ once per $\lambda$ and reusing across all input states. Achieved minimal final error of 0.0222 in zero-noise prediction (Section 6.3, Fig. 3). Recognized ZNE's intrinsic worst-case hardness but emphasized practical overhead reduction. Protocol follows standard ZNE assumptions with benefit of reusable channel representation. | mkLY, NjPR |

---

> ### Author Response · Authors · 2025-11-29
> **Summary of Revisions and Key Responses (part 2)**
>
> ### Table 2: Addressing Reviewer mkLY's Concerns: Key Manuscript Revisions
> | Reviewer mkLY's Specific Concern | Author's Response & Solution |
> | :--- | :--- |
> | **1. Unclear Problem Definition**: Is $\mathcal{C}$ known? What does "learn $\hat{\rho}$" mean? | Clarified: Both $\rho$ and $\mathcal{C}$ are **unknown** (only query-and-measure access). The goal of problem 1 is to learn the **classical representation of $\rho$**, emphasized on **page 4**. |
> ---
>
>
> ### Table 3: Addressing Reviewer FAFT's Concerns: Key Manuscript Revisions
>
> | Reviewer FAFT's Specific Concern | Author's Response & Manuscript Improvements |
> | :--- | :--- |
> | **1. Overselling & Imprecision**: The assumption should be clarified more; probability space undefined. | **Global terminology correction**: Replaced "Local noise" with "**i.i.d. single-qubit noise**" throughout. **Established rigorous probabilistic framework**: Formalized definitions of circuit architecture and random noisy circuit ensemble in Appendix B. **Explicitly clarified guarantees**: Stated that results hold with high probability ($\geq 1-\delta$) over this ensemble. |
> | **2. Noise Strength Knowledge**: How to choose truncation length $l'$ without prior knowledge of $\gamma$? | **Constant-level assumption**: Only assumes $\gamma$ is a \textbf{constant independent of system size}. Similar to how related works treat other parameters as constants (circuit depth $d$ in Huang et al. STOC 2024; evolution time $t$ in Haah et al. Nat. Phys. 2024), the scaling of $l'$ is asymptotically dominated by target accuracy $\epsilon$. **Numerical validation**: Fig. 2 confirms fixed, constant $l'$ works effectively across system sizes. (If needed, $\gamma$ can be estimated via standard randomized benchmarking.) |
> | **3. Theory-Experiment Mismatch**: Fig. 2 shows error decreasing with $n$, seemingly contradicting Theorem 1's $n$-independent sample complexity. | **Clarified asymptotic interpretation**: Explained that the decrease for small $n$ is a **transient finite-size effect**. **Highlighted visual validation**: The observed **error plateau at large $n$** experimentally confirms the predicted $n$-independence in the asymptotic regime. |
> | **4. Mathematical Rigor**: Derivation in Eq. (27) contained errors regarding sum reduction and square root manipulation. | **Acknowledged and fully corrected**: Identified the oversight and provided a complete, rigorous re-derivation of the mathematical analysis in **Appendix C.2**. |
> | **5. Frobenius Norm Scaling**: Theorem 2's dependence on $\|O\|_F$ could imply exponential scaling for Pauli operators. | **Proved polynomial scaling**: Clarified that for $k$-local operators, $\|O\|_F$ involves a polynomial number of Pauli terms, resulting in at most **polynomial scaling in $n$**. |
> ---
>
> ### Table 4: Addressing Reviewer khkR's Concerns: Revisions \& Score-Adjustment Conditions
> | Reviewer khkR's Specific Concern | Author's Response & Manuscript Improvements |
> | :--- | :--- |
> | **1. Comparison with Huang et al. (2023a)** | **Highlighted fundamental difference**: Our sparsity originates from noise via Pauli-path integration (holds for all input states), while Huang's efficiency requires input distribution constraint (polynomially far from locally flat). **Established input-agnostic advantage**: Our method works across entire Hilbert space without input restrictions, overcoming Huang's key limitation. Clarified that both methods handle few-body observables similarly; distinction is in input-state generality. Updated Appendix A to clarify the comparison. |
> | **Score-Adjustment Conditions (Second Rebuttal)** | |
> | **Cond. 1 (Score $\uparrow$ to 6)**: Weaken claim "establishes theoretical foundation..." | **Implemented**: Revised abstract to "offers a new approach... suggests a potential path..." using measured language. |
> | **Cond. 2 (Score $\uparrow$ to 6)**: Clarify random circuit sampling requirement in abstract | **Implemented**: Added explicit phrase "in the *average case* scenario over the random circuit ensemble" to abstract. |
> | **Add. Req. (Score $\uparrow$ to 8+)**: Theoretical worst-case analysis | **Delivered**: Proved the sample complexity lower bound for the worst-case scenario. Full analysis in **Appendix E**. |
> | **Add. Req. (Score $\uparrow$ to 8+)**: Benchmark comparison with Huang's method | **Delivered**: QPT experiment using **highly entangled input state** (outside Huang's valid distribution). Our algorithm performs well; results in **Appendix F.1**. |
> ---

---

> > ### Author Response · Authors · 2025-11-29
> > **Summary of Revisions and Key Responses (part 3)**
> >
> > ### Table 5: Addressing Reviewer VPA8's Concerns: Revisions Clarifying Scope
> > | Reviewer VPA8's Specific Concern | Author's Response & Manuscript Improvements |
> > | :--- | :--- |
> > | **1. Adaptive Sparsity Techniques** | **Acknowledged future direction**: Theoretical bound $l' \le \log(1/\epsilon)$ already guarantees efficiency. **Hypothesized extension**: LASSO could further compress terms but lacks rigorous theoretical guarantee. **Added to open problems**: Included this suggestion on page 10 for future investigation. |
> > | **Reviewer's Final Assessment: Score Maintained at 6** |
> > | **Positive Outcome** | **Clarifications successful**: Reviewer acknowledged that replies "clarify many aspects" and that the model is "well-motivated and widely used in the NISQ regime." Reviewer found theoretical guarantees "non-trivial and valuable." |
> > | **Reason for Not Raising Score** | **Future direction identified**: Need analysis beyond constant noise (e.g., $\gamma=1/\text{poly}(n)$) as hardware improves. Reviewer recognized this was already in discussion section but maintained extension is crucial for "substantially broaden the theory's impact" before score increase. |
> > ---
> > ### Table 6: Addressing Reviewer NjPR's Concerns: Revisions \& Second Assessment
> > | Reviewer NjPR's  Specific Concern | Author's Response & Manuscript Improvements |
> > | :--- | :--- |
> > | **1. Problem Triviality**: Efficiency seems from simple high-noise problem, not algorithm design. | **Established nontrivial advance**: Efficiency stems from exploiting compact representation enabled by **constant-noise assumption** (where "constant" means $\Omega(1)$ **independent of system size**, not necessarily large intensity). This framework handles both unital and non-unital single-qubit noise, contrasting sharply with prior works: (1) Classical simulation required full circuit/noise knowledge, while learning algorithm requires **no prior knowledge** of gates, architecture, or noise strength (only constant-noise assumption); (2) [Huang et al. PRX Quatum, 2023] was constrained to the input distributions, whereas our algorithm places **no restrictions** on the input distribution during the prediction phase.; (3) Prior works are limited to unitary-dual techniques on restricted circuits (constant-depth [Huang STOC 2024], constant-time dynamics [Wu arXiv 2025]) or weak O(1/n) noise for quantum state analysis [Wu ICLR 2025], neither approach supporting quantum process tomography under constant-strength noise. **To the best of our knowledge, this is the first efficient solution to the noisy state and process tomography problem.** |
> > | **2. ZNE Implementation**: Separate runs needed for each $\lambda$? | **Clarified protocol**: Follows same assumptions as standard ZNE. Key benefit is **input-agnostic reuse** after single run per $\lambda$, significantly reducing experimental overhead. |
> > | **Reviewer's Second Assessment: Score Maintained** | |
> > | **Outcome**: Reviewer retained score after second rebuttal. | **Partial acceptance**: Reviewer agreed tomography is hard and special properties needed; acknowledged ZNE utility but concluded protocol "only addresses constant level noise" without "substantial material for acceptance" and doesn't solve "substantial problem in ZNE." |
> > | **Authors' Second Defense** | **Emphasized practical significance**: Prior works limited to either (i) special circuit structures (constant-depth [Huang STOC 2024], constant-time dynamics [Wu arXiv 2025]) via non-generalizable techniques, or (ii) very weak $O(1/n)$ noise for quantum state analysis [Wu ICLR 2025] that doesn't support process tomography. Our work addresses the **open problem of quantum process tomography under realistic constant-strength noise** (may be small but doesn't diminish with system size). Represents progression from idealized regimes to practically meaningful scenarios. |
> > ---
> > We sincerely thank the Area Chair and all reviewers for their invaluable time and detailed feedback.
> >
> > We believe that through this comprehensive and structured rebuttal, we have effectively clarified the theoretical scope of our method and established the rigor of our algorithm. Our work proposes a provably efficient quantum learning algorithm that handles both unital and non-unital single-qubit noisy channels, extending previous art from restricted input distributions to arbitrary input quantum states.
> >
> > We hope that these detailed revisions and arguments sufficiently demonstrate the academic merit of this work, and we respectfully request the Area Chair to grant our submission favorable consideration.
> >
> > Best Regards!

---

### Meta-Review · Area_Chair_P5Dc · 2025-12-28

**Summary:**

This paper proposes an average-case efficient algorithm that can learn noisy quantum states and quantum channels with noise at constant level strength. The idea is to reduce the learning problem to estimation of an observable as a polynomial number of linear combinations of low-degree Pauli operators. The authors prove sample and time complexity bounds of their algorithm and show performance for an Ising model and zero-noise extrapolation (ZNE) application. Compared to the prior work, the proposed algorithm has the capacity to handle both unital and non-unital noise as well as arbitrary input noise.

The paper received constructive comments, where several reviewers raised the concern on the limitation of the constant-level noise assumption and questioned the practical relevance of the ZNE application. Most reviewers shared common concerns on the working circuit model with random local 2-design layers under which the complexity bounds derived in this paper hold.

**Reviewer Concerns:**

In the rebuttal, the authors have made extensive effort to clarify their contributions. After discussion, reviewers acknowledged the difficulty of learning through tomography; but majority tended to maintain their scores. Given the theory only addresses constant-level noise and works with noisy quantum processes with random local 2-design layers, it is unclear how this paper can advance the learning problem for noisy quantum state and process from measurement data, and there seems to be plenty of room for the authors to improve the current form accordingly for another publishing venue.

**Reviewer Scores:**

5 reviewers submitted their comments and scores (2/6/4/4/2) with confidence (3/2/3/4/4), with average score 3.6 and average confidence 3.2.

Reviewer NjPR gave concrete reasons (constant level noise + questioning applicability to ZNE) that they will retain the current score (4) with confidence 4.

Reviewer VPA8 recognized some merits of this work, but concerned the extendibility beyond constant noise level.  They will maintain their score (6) with confidence 2.

Reivewer khkR stated two conditions for possibility of increasing their score from 2 to higher scores. The authors correspondingly weakened their claim in the revised paper and added a Theorem 5 in Appendix E for worst-case sample complexity lower bound. But it seems that the numeric analysis was not present and unclear how to benchmark comparing with Huang’s method.

There was no discussion for Reviewers mKLY and FAFT.

---

### Decision · Program_Chairs · 2026-01-26

Reject